# Vulnerability of amphibians to global warming

Patrice Pottier[1,2 ✉], Michael R. Kearney[3], Nicholas C. Wu[4], Alex R. Gunderson[5], Julie E. Rej[5], A. Nayelli Rivera-Villanueva[6,7], Pietro Pollo[1], Samantha Burke[1], Szymon M. Drobniak[1,8,10] & Shinichi Nakagawa[1,9,10]

Amphibians are the most threatened vertebrates, yet their resilience to rising temperatures remains poorly understood[1,2]. This is primarily because knowledge of thermal tolerance is taxonomically and geographically biased[3], compromising global climate vulnerability assessments. Here we used a phylogenetically informed data-imputation approach to predict the heat tolerance of 60% of amphibian species and assessed their vulnerability to daily temperature variations in thermal refugia. We found that 104 out of 5,203 species (2%) are currently exposed to overheating events in shaded terrestrial conditions. Despite accounting for heat-tolerance plasticity, a 4 °C global temperature increase would create a step change in impact severity, pushing 7.5% of species beyond their physiological limits. In the Southern Hemisphere, tropical species encounter disproportionally more overheating events, while non-tropical species are more susceptible in the Northern Hemisphere. These findings challenge evidence for a general latitudinal gradient in overheating risk[4–6] and underscore the importance of considering climatic variability in vulnerability assessments. We provide conservative estimates assuming access to cool shaded microenvironments. Thus, the impacts of global warming will probably exceed our projections. Our microclimate-explicit analyses demonstrate that vegetation and water bodies are critical in buffering amphibians during heat waves. Immediate action is needed to preserve and manage these microhabitat features.

Climate change has pervasive impacts on biodiversity, yet the extent and consequences of this environmental crisis vary spatially and taxonomically[7,8]. For ectothermic species, such as amphibians, the link between climate warming and body temperature is clear, with immediate effects on physiological processes[9]. Over 40% of amphibian species are currently listed as threatened, and additional pressures due to escalating thermal extremes may further increase their extinction risk[2,10]. It is therefore vital to assess the resilience of amphibians to climate change to prioritize where and how conservation actions are taken.

Accurate assessments of resilience to climate change require adequate data on thermal tolerance and environmental exposure[5,6,11]. However, the most exhaustive dataset on amphibian heat-tolerance limits only covers 7.5% of known species and is geographically biased towards temperate regions[3] (Fig. 1). This discrepancy is problematic, considering the high species richness in the tropics and the mounting evidence that tropical ectotherms are most susceptible to rising temperatures[4–6,12,13]. Such sampling biases call into question the reliability of inferences in undersampled areas and have implications for conservation strategies. Given the rapid pace of climate change and the finite resources available for research, acquiring sufficient empirical data to fill these knowledge gaps within a realistic timeframe is increasingly untenable[14,15]. Thus, alternative methods to identify the populations and areas most susceptible to thermal stress are critically needed in a rapidly warming climate.

Climate vulnerability assessments also require environmental data with high spatial and temporal resolution, particularly because extreme heat is more likely to trigger overheating events than increased mean temperatures[16–18]. When heat-tolerance limits are known, cutting-edge approaches in biophysical ecology enable fine-scale vulnerability assessments that account for morphology, behaviour and microhabitat setting in both historical and future climate projections[19,20]. While broadly applicable, biophysically informed analyses are particularly relevant for amphibians, whose body temperatures depend on evaporative heat loss and whose microhabitat use spans terrestrial, aquatic and arboreal environments. As microenvironmental features are essential for behavioural thermoregulation[21,22], modelling microhabitats enables assessments of the effectiveness of different thermal refugia in buffering the impacts of extreme heat events.

[1]Evolution & Ecology Research Centre, School of Biological, Earth and Environmental Sciences, University of New South Wales, Sydney, New South Wales, Australia. [2]Division of Ecology and Evolution, Research School of Biology, The Australian National University, Canberra, Australian Capital Territory, Australia. [3]School of BioSciences, The University of Melbourne, Melbourne, Victoria, Australia. [4]Hawkesbury Institute for the Environment, Western Sydney University, Richmond, New South Wales, Australia. [5]Department of Ecology and Evolutionary Biology, Tulane University, New Orleans, LA, USA. [6]Centro Interdisciplinario de Investigación para el Desarrollo Integral Regional Unidad Durango (CIIDIR), Instituto Politécnico Nacional, Durango, Mexico. [7]Laboratorio de Biología de la Conservación y Desarrollo Sostenible de la Facultad de Ciencias Biológicas, Universidad Autónoma de Nuevo León, Monterrey, Mexico. [8]Institute of Environmental Sciences, Faculty of Biology, Jagiellonian University, Kraków, Poland. [9]Department of Biological Sciences, University of Alberta, Edmonton, Alberta, Canada. [10]These authors jointly supervised this work: Szymon M. Drobniak, Shinichi Nakagawa. ✉e-mail: p.pottier@unsw.edu.au

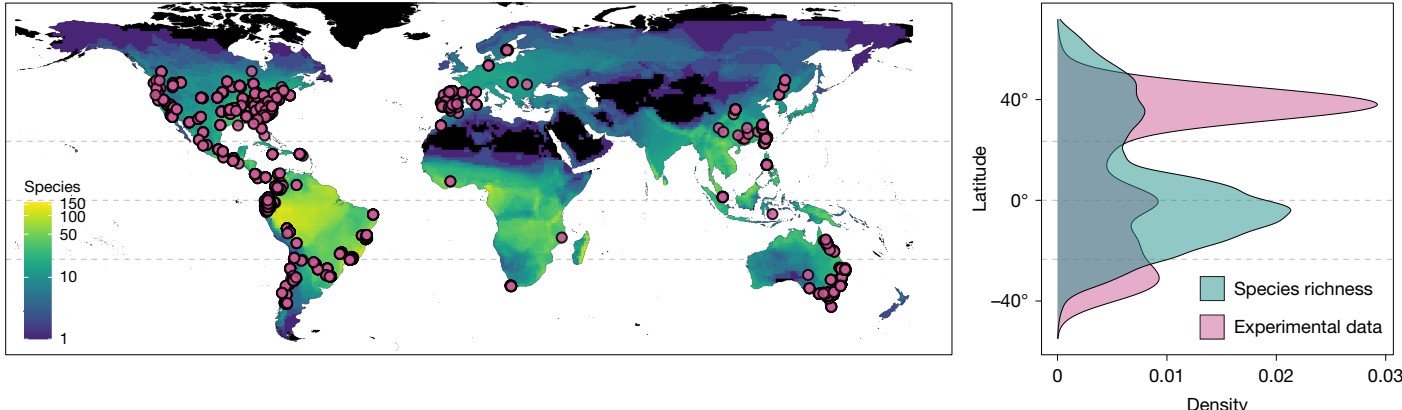

**Fig. 1 | Contrast between the geographical locations at which experimental data were collected and patterns in species richness.** The pink points denote experimental data (*n* = 587 species), and the colour gradients refer to species richness calculated in 1° × 1° grid cells in the imputed data (*n* = 5,203 species).

The density plots on the right represent the distribution of experimental data (pink) and the number of species inhabiting these areas (blue) across latitudes. The black shading indicates areas with no data. The dashed lines represent the equator and tropics.

Here we assess the global vulnerability of amphibians to extreme heat events in different climatic scenarios and thermal refugia (Extended Data Fig. 1). By integrating predicted thermal limits for 60% of amphibian species with daily operative body temperatures, our study offers a comprehensive evaluation of the impact of heat extremes on the physiological viability of amphibians in nature.

## Thermal limits and environmental exposure

We first developed an approach to predict standardized thermal limits for 5,203 amphibian species using data imputation based on phylogenetic niche clustering (Pagel's $\lambda$ = 0.95 (95% credible interval 0.91–0.98)) and known correlations between critical thermal limits ($CT_{max}$) and other variables (*n* = 2,661 estimates measured in 524 species; Methods). Our phylogenetic model-based imputation approach has expanded our understanding of amphibian thermal tolerance by generating testable predictions for 4,679 unstudied species, particularly in biodiversity hotspots (Figs. 1 and 2). We confirmed that our imputation approach was probably accurate and unbiased by demonstrating a strong congruence between experimental and imputed data in cross-validations (experimental mean ± s.d. = 36.19 ± 2.67; imputed mean ± s.d. = 35.93 ± 2.54; *n* = 375; *r* = 0.86; Extended Data Fig. 2a,b), although, as expected, the uncertainty in imputed predictions was higher in understudied clades (Extended Data Fig. 2c).

We next integrated predicted thermal limits with daily maximum operative body temperature fluctuations estimated from biophysical models to evaluate the sensitivity of amphibians to extreme heat events in terrestrial, aquatic and arboreal microhabitats (Methods and Extended Data Fig. 1). Operative body temperatures are the steady-state body temperatures that organisms would achieve in a given microenvironment, which can diverge from ambient air temperatures due to, for example, radiative and evaporative heat-exchange processes[19,20]. For each microhabitat, we modelled daily operative body temperatures during the warmest quarters of 2006–2015 and across the distribution range of each species (Methods). We also used projected future climate data from TerraClimate[23] to generate projections assuming 2 °C or 4 °C of global warming above pre-industrial levels. These temperatures are within the range projected by the end of the century under low and intermediate/high greenhouse gas emission scenarios, respectively[24]. Notably, recent historical $CO_2$ emissions most closely align with high warming scenarios[25] (that is, 4.3 °C of predicted warming by 2100). All microenvironmental projections assumed access to 85% of shade and that amphibians had access to sufficient water to avoid desiccation in thermal refugia (Methods).

We estimated the vulnerability of amphibians by estimating daily differences between predicted thermal limits and maximum hourly operative body temperatures (Methods and Extended Data Fig. 1). We also adjusted daily thermal limits to assume that species were, on any given day, acclimatized to local mean weekly operative body temperatures, effectively accounting for plasticity throughout species' distribution ranges (Methods). In total, we predicted vulnerability metrics for 203,853 local species occurrences (individual species in 1° × 1° grid cells) in terrestrial conditions (5,177 species), 204,808 local species occurrences in water bodies (5,203 species); and 56,210 local species occurrences (1,771 species) in aboveground vegetation, for each warming scenario. The number of species examined in arboreal conditions was lower to reflect morphological adaptations required for climbing in aboveground vegetation. These estimates were then grouped into assemblages (all species occurring in 1° × 1° grid cells), tallying 14,090 and 14,091 assemblages for terrestrial and aquatic species and 6,614 assemblages for arboreal species, respectively.

## Vulnerability to historical and future heat

We first calculated thermal safety margins (TSMs, sensu[6]) as the weighted mean difference between the heat-tolerance limits ($CT_{max}$) and the maximum daily body temperatures of the warmest quarters of 2006–2015 for each local species occurrence. TSMs averaged from long-term climatology are routinely used in climate vulnerability analyses[26–28]. We found evidence for a decline in TSM towards mid to low latitudes in all microhabitats, a pattern maintained across warming scenarios (Fig. 3 and Extended Data Fig. 3). However, warming substantially reduced TSMs at all latitudes (Fig. 3), probably reflecting the contrast between weak plastic responses in $CT_{max}$ across latitudes[11,15] and large variation in environmental temperatures (Extended Data Fig. 3). Across all conditions simulated, TSMs are always positive, even in the highest warming scenario (Fig. 3 and Extended Data Fig. 3). The mean TSM is lower for terrestrial (mean (95% credible intervals); current, 11.69 (8.86–14.43); +4 °C, 9.41 (6.53–12.09)) and arboreal conditions (current, 12.23 (9.40–14.96); +4 °C, 10.07 (7.23–12.80)) than for water bodies (current, 13.60 (10.71–16.28); +4 °C, 11.68 (8.80–14.36); Fig. 3 and Supplementary Table 1).

Because extreme heat events are more likely to trigger overheating events than mean temperatures[5,6,11], we also calculated the binary probability (0/1) that operative body temperatures exceeded $CT_{max}$ for at least one day across the warmest quarters of 2006–2015 (that is, overheating risk). Overall, overheating risk is low, although numerous species are predicted to face overheating events locally (Fig. 4 and Supplementary Table 2). In terrestrial conditions, we predict that 104

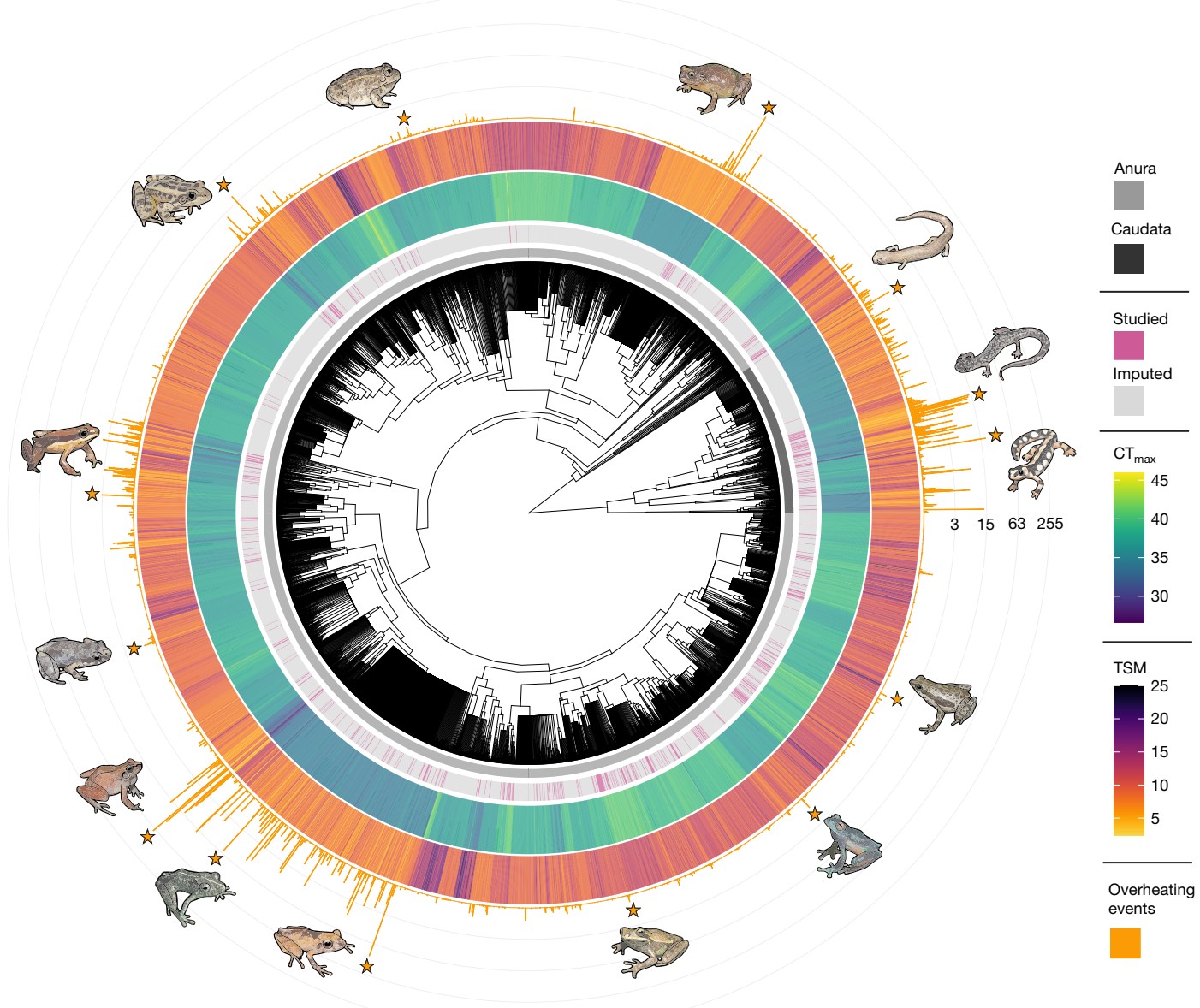

**Fig. 2 | Phylogenetic coverage and taxonomic variation in climate vulnerability.** Heat-tolerance limits ($CT_{max}$; inner heat map), TSM (outer heat map), and the number of overheating events (days) averaged across each species' distribution range (histograms) ($n = 5,177$ species). The pink bars refer to species for which there was previous knowledge ($n = 521$), and the grey bars refer to entirely imputed species ($n = 4,656$). This figure was constructed assuming ground-level microclimates occurring under 4 °C of global warming above pre-industrial levels. Phylogeny is based on the consensus of 10,000 trees sampled from a posterior distribution (described previously[60]). Highlighted species starting from the right side, anti-clockwise: *Neurergus kaiseri*, *Plethodon kiamichi*, *Bolitoglossa altamazonica*, *Cophixalus aenigma*, *Tomaptera cryptotis*, *Lithobates palustris*, *Allobates subfolionidificans*, *Phyzelaphryne miriamae*, *Barycholos ternetzi*, *Pristimantis carvalhoi*, *Pristimantis ockendeni*, *Boana curupi*, *Teratohyla adenocheira* and *Atelopus spumarius*.

species (836 local species occurrences from 253 assemblages) are likely to experience overheating events in current microclimates (Figs. 4 and 5). However, under 4 °C of warming, 391 species (4,248 local species occurrences from 1,328 assemblages) are expected to overheat, which represents nearly a fourfold increase relative to current conditions (Figs. 4 and 5 and Supplementary Tables 2 and 3). The number of species predicted to overheat in each grid cell also increases with warming; each assemblage comprises up to 18 vulnerable species in current climates (mean (95% confidence intervals) = 3.19 (0.60–6.88) species) and up to 37 vulnerable species with 4 °C of global warming (3.08 (0.62–6.56); Fig. 4 and Supplementary Table 3). Moreover, the proportion of species predicted to experience overheating events in each assemblage varies geographically and between warming scenarios (Extended Data Fig. 5 and Supplementary Table 4). The proportion of species at risk is high in some areas with high species richness (such as Northern Australia, Southeastern United States) and not linearly predicted by latitude (Extended Data Fig. 5).

In current conditions for species that can shelter in trees (arboreal), 74 assemblages (comprising 1–6 species; 1.93 (95% confidence interval 0.05–5.05) species) are predicted to overheat, while 285 assemblages (comprising 1–11 species; 2.51 (0.31–5.69) species) are predicted to overheat assuming 4 °C of global warming (Fig. 4 and Supplementary Table 3). While the overheating risk is lower in arboreal conditions, considerably fewer species were examined than in terrestrial conditions (1,771 versus 5,177 species). In fact, comparing the responses of arboreal species in different microhabitats revealed that occupying aboveground vegetation is only partially beneficial (Extended Data Fig. 4). In current climates, up to 15 arboreal species (320 local species

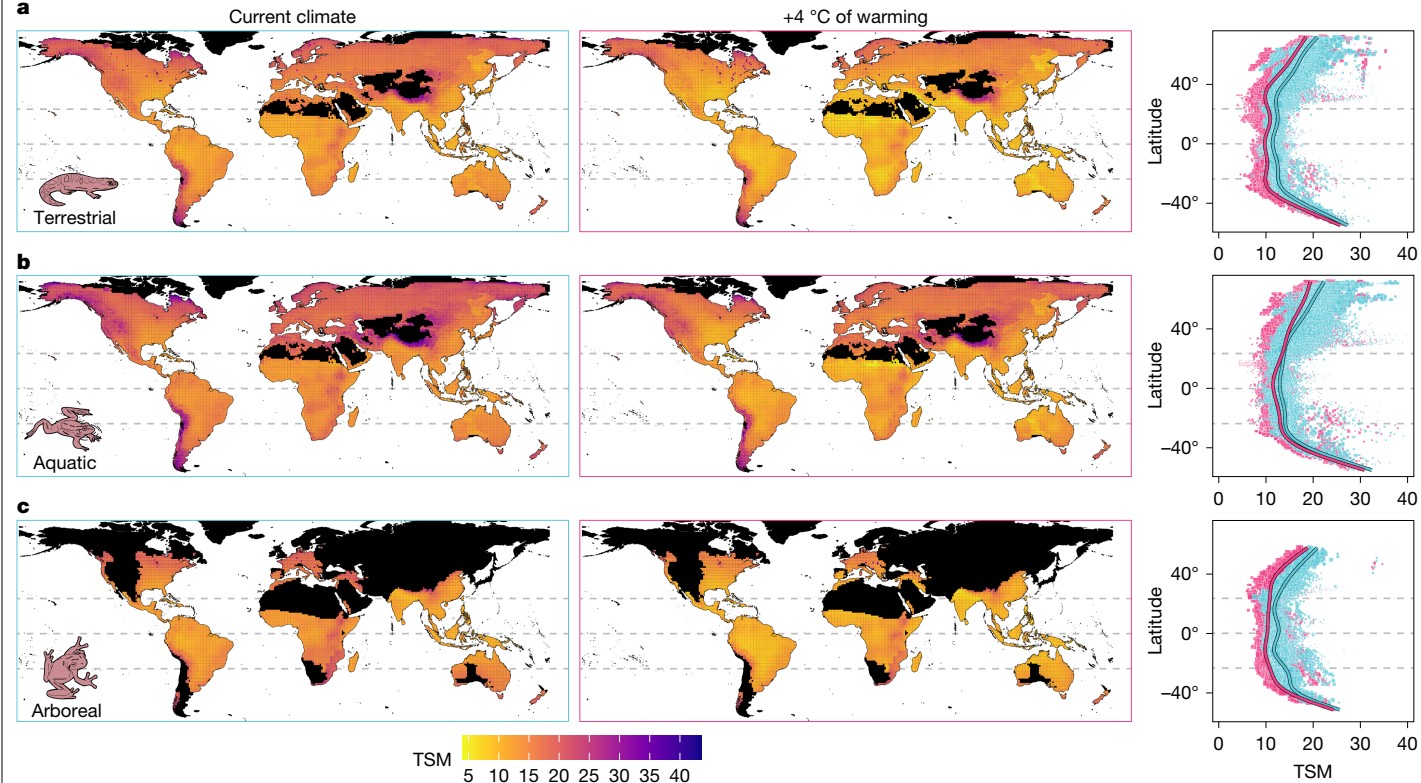

**Fig. 3 | Assemblage-level patterns in the TSM for amphibians. a–c**, The assemblage-level patterns in the TSM for amphibians in the terrestrial (**a**), aquatic (**b**) or arboreal (**c**) microhabitats. TSMs were calculated as the weighted mean difference between $CT_{max}$ and the predicted operative body temperature in full shade during the warmest quarters of 2006–2015 in each assemblage (1° grid cell; $n = 14,090$ for terrestrial species; $n = 14,091$ for aquatic species; $n = 6,614$ for arboreal species). Black colour depicts areas with no data. Right, the mean latitudinal patterns in TSM in current climates (blue) or assuming 4 °C of global warming above pre-industrial levels (pink), as predicted from generalized additive mixed models. Point estimates are scaled by precision (1/s.e.), with smaller points indicating greater uncertainty. The dashed lines represent the Equator and Tropics.

occurrences) are predicted to experience overheating events in terrestrial conditions, whereas 13 arboreal species (152 local species occurrences) are predicted to overheat in aboveground vegetation (Extended Data Fig. 4). Furthermore, under 4 °C of warming, 83 arboreal species (1,137 local species occurrences) are predicted to overheat in terrestrial conditions, while retreating to aboveground vegetation reduces the number of species exposed to overheating events by only 32.5% (56 species, 748 local species occurrences) (Extended Data Fig. 4). Contrary to terrestrial and arboreal conditions, no amphibian populations are predicted to overheat in water bodies in current or intermediate climate warming scenarios owing to the thermal buffering properties of water. However, assuming 4 °C of climate warming, we predict that 11 species (56 local species occurrences from 48 assemblages) will exceed their physiological limits in aquatic microhabitats (Fig. 4).

Finally, we quantified the number of days (out of 910 simulated days across the warmest quarters of 2006–2015) that each species was predicted to locally exceed their plasticity-adjusted $CT_{max}$. This metric fully integrates the frequency at which amphibians are predicted to experience temperatures beyond their thermal limits. For current climates, we found that species rarely experience overheating events in shaded terrestrial conditions (overall mean overheating days (95% confidence intervals) = 0.01 (0.01–0.08); mean among overheating species = 2.15 (0.24–5.26) days; but these figures increase considerably with global warming (Fig. 5 and Supplementary Table 2). Under 4 °C of warming, species are predicted to overheat on as many as 207.18 (182.39–231.97) days, representing up to 22.8% of the warmest days of the year (overall mean = 0.15 (0.05–0.46) days; mean among overheating species = 6.75 (3.14–11.38) days; Fig. 5 and Supplementary Table 2). This is noticeably more than what is predicted under 2 °C of warming (overall mean = 0.02

(0.01–0.13) days; mean among overheating species = 2.58 (0.41–5.86) days; Fig. 5 and Supplementary Table 2). In aboveground vegetation, the frequency of overheating events is lower, as expected. Under current climates, arboreal species are predicted to overheat on up to 5.65 (1.00–10.29) days in total (overall mean = 0.01 (0.01–0.04) days; mean among overheating species = 1.62 (0.03–4.43) days; Fig. 5 and Supplementary Table 2). Under 4 °C of warming, arboreal species are predicted to overheat on up to 76.17 (59.79–92.54) days (overall mean = 0.08 (0.01–0.23) days; mean among overheating species = 5.08 (1.81–9.39) days; Fig. 5 and Supplementary Table 2). Arboreal species retreating to aboveground vegetation are predicted to experience fewer overheating events than those in terrestrial conditions (Extended Data Fig. 4). Notably, we found that species predicted to overheat locally have TSMs well above zero, although some are living particularly close to their heat-tolerance limits during the warmest months in both terrestrial (mean (95% credible intervals); current, 8.20 (6.91–9.98), range = 3.02–12.19; +4 °C, 6.30 (5.02–8.09), range = 0.97–11.27) and aboveground conditions (current, 8.71 (7.20–10.28), range = 3.70–9.76; +4 °C, 6.73 (5.44–8.48), range = 1.75–8.70; Fig. 5c,d). Finally, we found a strong nonlinear negative association between the number of overheating events and the TSM, with stark contrasts between warming scenarios (Fig. 5c,d and Supplementary Table 5). In particular, overheating days increase rapidly as TSMs fall below 5 °C (Fig. 5c,d).

## The mounting impacts of global warming

Quantifying the resilience of biodiversity to a changing climate is one of the most pressing challenges for contemporary science[7,8]. Here we show that over a hundred species may already experience hourly

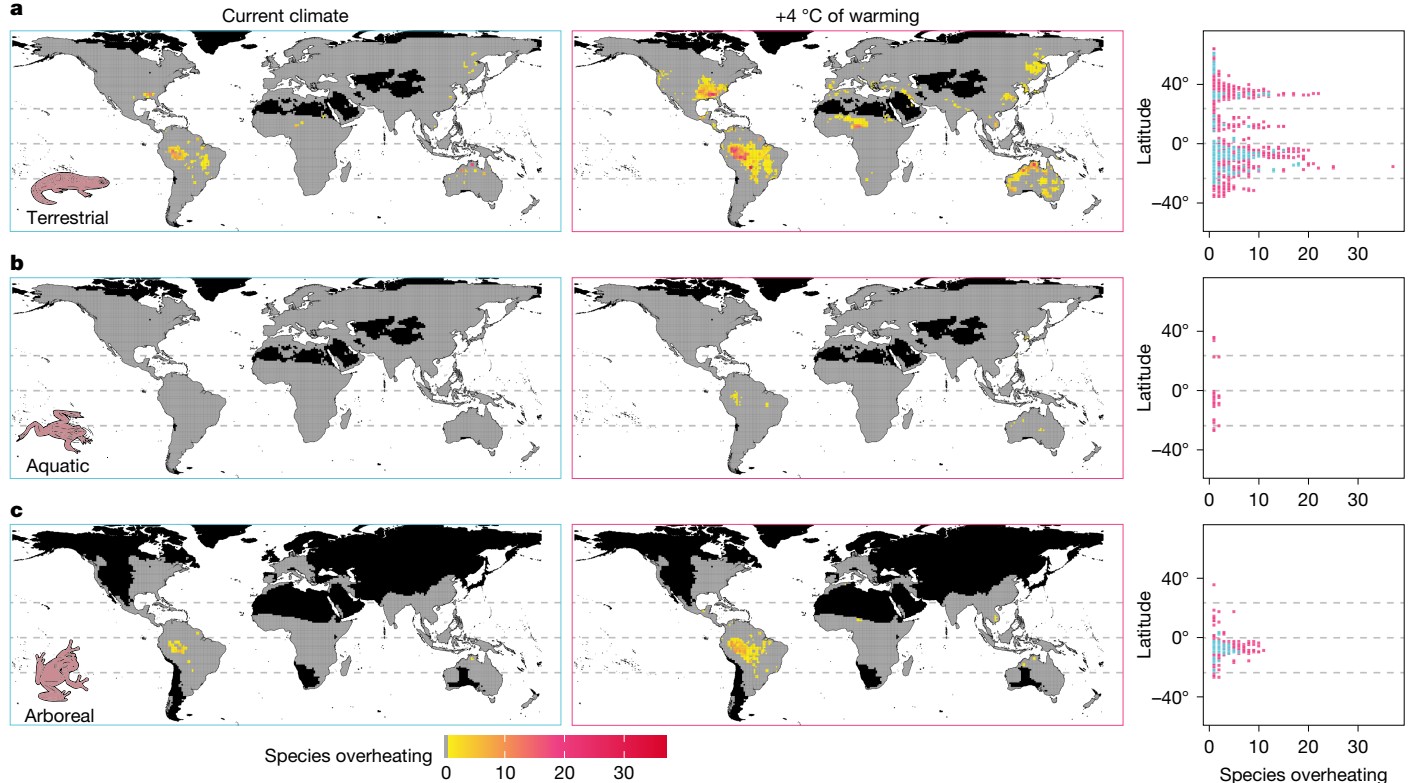

**Fig. 4 | The number of species predicted to experience overheating events.** The number of species predicted to experience overheating events in terrestrial (**a**), aquatic (**b**) and arboreal (**c**) microhabitats. The number of species overheating was assessed as the sum of species overheating for at least one day in the period surveyed (the warmest quarters of 2006–2015) in each assemblage (1° grid cell; $n = 14,090$ (terrestrial species), $n = 14,091$ (aquatic species), $n = 6,614$ (arboreal species)). The black shading indicates areas with no data, and grey shading shows assemblages without species at risk of overheating. Right, latitudinal patterns in the number of species predicted to overheat in current climates (blue) or assuming 4 °C of global warming above pre-industrial levels (pink). The dashed lines represent the Equator and Tropics.

temperatures that would probably result in death over minutes or hours of exposure in thermal refugia. This pattern is only predicted to worsen (Figs. 4 and 5). Assuming 4 °C of global warming, the number of species and assemblages exposed to overheating events would be four to five times higher than currently, totalling 391 out of 5,203 species studied (7.5%; Figs. 4 and 5).

We also found marked disparities in overheating risk between the 2 °C and 4 °C warming projections (Fig. 5 and Supplementary Table 1), which are anticipated by the end of the century under low and high greenhouse gas emission scenarios, respectively[24]. The more extreme warming scenario considerably increased the number overheating events experienced by amphibian populations (Fig. 5), highlighting the escalating and abrupt impacts of global warming[7,29]. Such an increase is attributable to the contrast between the rapid pace at which temperatures are increasing and the low ability of amphibians to acclimatize to new thermal environments through plasticity (Extended Data Fig. 3; species-level acclimatization response ratio ± s.d. = 0.134 ± 0.008). Our study clearly demonstrates, as others have suggested[18,27,30,31], that physiological plasticity is not a sufficient mechanism to buffer many populations from the impacts of rapidly rising temperatures.

## Extreme heat events drive vulnerability

We found large spatial heterogeneity in the vulnerability of amphibians. In tropical areas, most vulnerable species are concentrated in South America and Australia, whereas fewer species are impacted in the African and Asian tropics (Fig. 4). Tropical species also experience disproportionately more overheating events in the Southern Hemisphere, while non-tropical species are more susceptible in the Northern Hemisphere (Fig. 5). Furthermore, the proportion of species experiencing overheating events in each assemblage was not predicted by latitude (Extended Data Fig. 5). Thus, our findings are inconsistent with the expectation of a general latitudinal gradient in overheating risk based on TSMs[4–6,13]. In fact, the overheating risk does not increase linearly with TSM (Fig. 5c,d), and species with seemingly comparable TSMs can have markedly different probabilities of overheating due to varying exposure to daily temperature fluctuations (Fig. 5c,d). Thus, TSMs alone hide critical tipping points for thermal stress (Fig. 5c,d).

Our study questions the reliability of TSMs and other climate vulnerability metrics when averaged across large time scales (for example, using the maximum temperature of the warmest quarter) for detecting species most vulnerable to thermal extremes. It also challenges the general notion that low-latitude species are uniformly most vulnerable to warming[4–6,13], revealing a far more nuanced pattern of climate vulnerability across latitudes. While the reliability of TSM-based assessments has been questioned in previous studies[11], our work further emphasizes the need to consider natural climatic variability and extreme hourly temperatures[4,16–18] when evaluating the vulnerability of ectotherms to global warming. Considering alternative metrics, such as the number of predicted overheating events, may prove particularly useful in identifying the most vulnerable species and populations.

## The vital yet limited role of thermal retreats

Our study highlights the critical yet sometimes insufficient role that thermal retreats have in buffering the impacts of warming on amphibians. Most amphibian species are predicted not to experience overheating events in full shade (Fig. 4), and the availability of water bodies

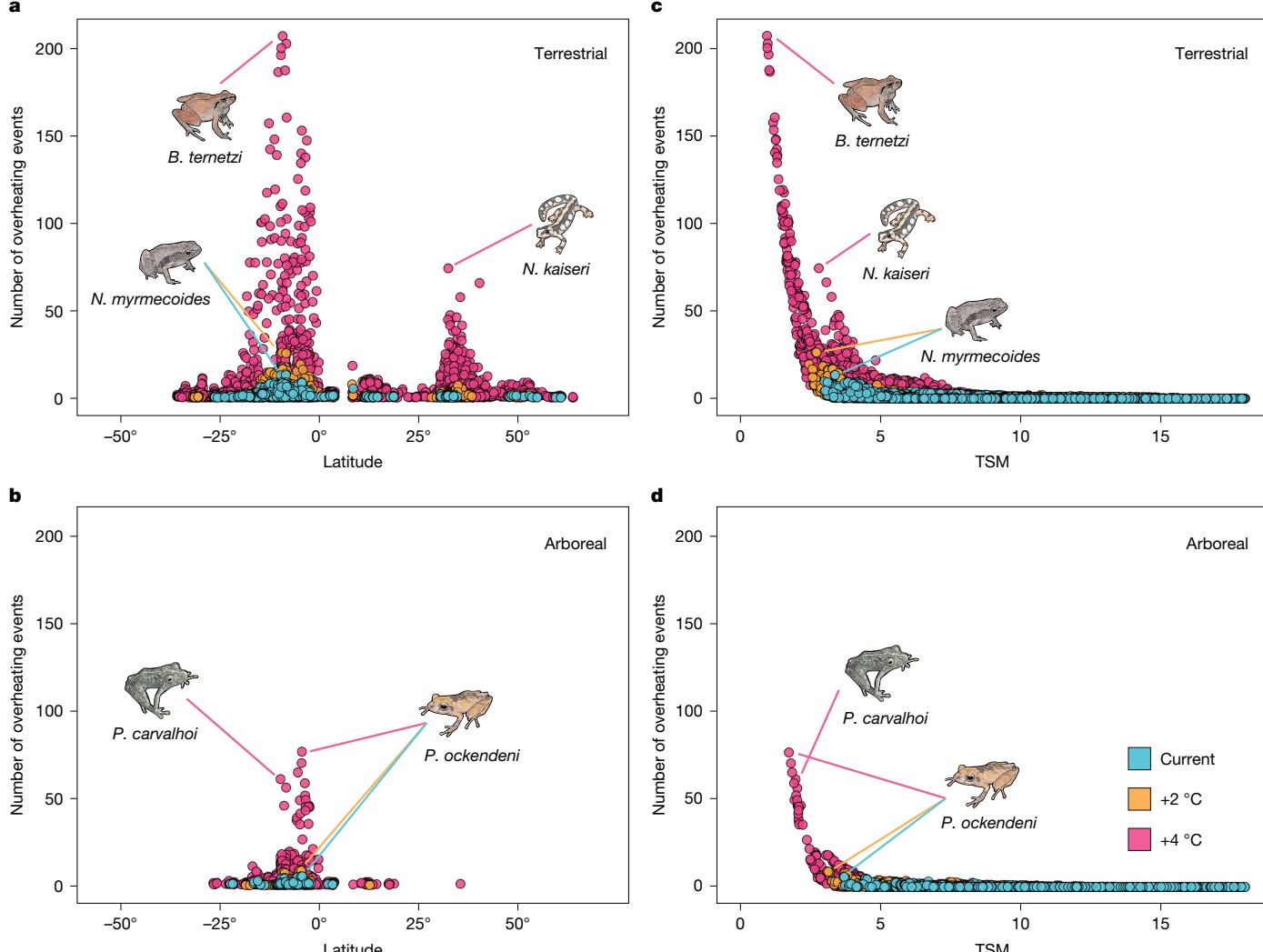

**Fig. 5 | Latitudinal variation in the number of overheating events as a function of latitude and TSM.** Latitudinal variation in the number of overheating events in terrestrial (**a**,**c**) and arboreal (**b**,**d**) microhabitats as a function of latitude (**a**,**b**) and TSM (**c**,**d**). The number of overheating events (days) was calculated based on the mean probability that daily maximum temperatures exceeded the $CT_{max}$ during the warmest quarters of 2006–2015 for each species in each grid cell (that is, local species occurrences; $n = 203,853$ (terrestrial species); $n = 204,808$ (aquatic species); $n = 56,210$ (aquatic species)). The blue points depict the number of overheating events in current microclimates, while the orange and pink points depict the number of overheating events assuming 2 °C and 4 °C of global warming above pre-industrial levels, respectively. For clarity, only the species predicted to experience at least one overheating event are depicted across latitudes (**a**,**b**). Highlighted species are as follows: *Neurergus kaiseri*, *Noblella myrmecoides*, *Barycholos ternetzi*, *Pristimantis carvalhoi* and *Pristimantis ockendeni*.

allows nearly all amphibians to maintain their body temperatures below critical levels, apart from 11 species in the most extreme warming scenario investigated. This is attributable to the higher specific heat capacity of water relative to air, delaying rapid temperature rises and affording a more stable environment during heat waves[32]. Our findings add to the growing evidence that finding access to cooler microhabitats is the main strategy that amphibians and other ectotherms can use to maintain sublethal body temperatures[6,21,33].

However, it is crucial to emphasize that vegetated terrestrial conditions in full shade offer inadequate protection to 7.5% of species, and many arboreal species predicted to overheat at ground level face similar risks in aboveground vegetation (Figs. 4 and 5 and Extended Data Fig. 4). In fact, although reducing the frequency of overheating events (Extended Data Fig. 4), access to shaded aboveground vegetation reduces the number of vulnerable species by only 32.5%. Moreover, although burrows offer cooler microclimates (Extended Data Fig. 9), the ability to use underground spaces is not universal among amphibians and can greatly restrict activity, reproduction and foraging opportunities.

## Warming impacts may exceed projections

Our predictions are largely conservative, and probably overestimate the resilience of amphibians to global warming in two main ways. First, we assume that microhabitats such as shaded ground-level substrates, aboveground vegetation and water bodies are available throughout a species' range, and that amphibians can maintain wet skin. These assumptions will often be violated as habitats are degraded. Deforestation and urbanization are diminishing vital shaded areas[34,35], while increased frequencies of droughts will cause water bodies to evaporate[36,37]. These changes compromise not only habitat integrity but also local humidity levels—key for effective thermoregulation[38,39]. Consequently, amphibians will probably experience higher body temperatures and desiccation stress events than our models predict due to inconsistent access to cooler microhabitats[40], particularly in degraded systems.

Second, ectotherms can experience deleterious effects from heat stress before reaching their heat-tolerance limits. Prolonged exposure

to sublethal temperatures can lead to altered activity windows[41,42], disruptions to phenology[43,44], reduced reproductive fitness (fertility and fecundity)[28,45,46] and death[47,48]. Although comprehensive data on thermal incapacitation times and fertility impacts are sparse in amphibians, integrating both the duration and intensity of thermal stress[48–50] will probably point to more extreme vulnerability estimates. This represents a vital avenue for future research, albeit one requiring a large collection of empirical data.

Alternatively, species that can retreat underground during heat events are likely to experience fewer overheating events than our models predict (Extended Data Fig. 9), and prolonged exposure to high temperatures in the permissive range (sensu[47]) can enhance performance and fitness, thereby reducing the impacts of extreme heat on natural populations. Moreover, some species may adapt to changing temperatures. However, evidence for slow rates of evolution and physiological constraints on thermal tolerance[51,52] challenges the likelihood of local adaptation to occur in rapidly warming climates.

## The power of data imputation

Our imputation approach has generated testable predictions of the thermal limits of 5,203 species, expanding the scope of previous research[3] (Fig. 2). We also addressed geographical biases by generating predictions in undersampled but ecologically critical regions of Africa, Asia and South America (Fig. 2). We found that these understudied regions frequently contain species exhibiting the highest susceptibility to extreme heat events (Figs. 1, 4 and 5), with 74% (288 out of 391) of vulnerable species remaining unstudied. Targeted research efforts in these vulnerability hotspots are instrumental in validating our model predictions and advancing our understanding of amphibian thermal physiology to inform their conservation. Although undeniable logistical and financial challenges exist in accessing some of these remote locations, collaboration with local scientists could expedite data collection and result in timely conservation measures. Exemplary initiatives to sample numerous species in South America[22,53,54] are promising steps in this direction, and we hope that our findings will catalyse research activity in these regions.

## Amphibian biodiversity in a warming world

Our study highlights the dire consequences of global warming on amphibians. Yet it is crucial to differentiate between global extinction and local extirpations—the latter being confined extinctions within specific geographical areas. Most species will not experience overheating events throughout their entire range, and these overheating events may not occur simultaneously. Thus, most species are likely to experience only local extirpation due to overheating, according to our models. Nevertheless, local extirpations carry their own sets of ecological repercussions, such as reshuffling community compositions and eroding genetic and ecological diversity[55,56].

Some amphibian populations may also undergo range shifts, permanently or transiently relocating to habitats with more hospitable weather patterns[57]. However, this is only possible if suitable habitats are available for establishment. Given the low dispersal rates of some amphibians and their common reliance on water bodies for reproduction and thermoregulation, opportunities for range shifts are likely to be rare for many species. Identifying which species at high risk of overheating are simultaneously predicted to have a limited ability to extend their range is an interesting route for research. Moreover, we stress that amphibians living close to their physiological limits for extended times at the warm edge of their distribution are likely to experience heat stress that could hamper activity, foraging opportunities and reproductive success, adding layers of complexity to their survival challenges and potentially leading to population declines[41,47].

Overall, our study contributes to the evidence that climate change is a mounting threat to amphibians[2,10] and emphasizes the importance of limiting global temperature rises below 2 °C to minimize the risk of overheating to amphibian populations. A 4 °C temperature increase would not only increase these risks but also create a step change in impact severity (Fig. 5c). The mechanistic basis of our species- and habitat-specific predictions also leads to clear management priorities. Particularly, our analyses revealed the critical importance of preserving dense vegetation cover and water bodies. These microhabitats provide conditions with cooler and more stable temperatures and increase the potential for amphibians and other ectothermic species to disperse to more suitable microhabitats. Establishing protected areas and undertaking habitat restoration initiatives may support amphibians in a changing climate and buffer additional anthropogenic threats, in turn mitigating amphibian population declines[2,10,58]. These actions are critical for the amphibians at risk and the ecosystems that they support[59] in a planet undergoing perilous climatic changes.

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

## Methods

### Reporting

We report author contributions using the CRediT (Contributor Roles Taxonomy) statement[61] and MeRIT (Method Reporting with Initials for Transparency) guidelines[62]. We also crafted the study title, abstract and keywords to maximize indexing in search engines and databases[63]. All analyses were performed using R statistical software[64] (v.4.3.0), and most computations used the computational cluster Katana supported by Research Technology Services at UNSW Sydney. Maps, phylogenetic trees and data visualizations were generated using the R packages rnaturalearthhires[65], ggtree[66] and ggplot2[67].

### Amphibian heat-tolerance limits

We used the most comprehensive compilation of amphibian heat-tolerance limits[3] for our analyses (Extended Data Fig. 1). In brief, these data were collated by systematically reviewing the literature in five databases and seven languages, comprising 3,095 heat-tolerance limits from 616 amphibian species. To facilitate the comparability and analysis of heat-tolerance limits, we included only data matching four specific criteria. First, we only included heat-tolerance limits measured using a dynamic methodology (that is, the temperature at which animals lose their motor coordination when exposed to ramping temperatures, critical thermal maximum $CT_{max}$[68]) because it was the most used and comparable metric. Second, we selected only data for which the laboratory acclimatization temperature, or the field temperature during the month of capture, was recorded. Third, we included only data from species listed in the phylogeny from a previous report[60]. Fourth, we included only species for which their geographical range was reported in the International Union for the Conservation of Nature Red List[69] (accessed in January 2023).

These criteria were chosen to perform phylogenetically, climatically and spatially informed analyses. In total, we selected 2,661 heat-tolerance limits estimates with metadata for 524 amphibian species (mean = 5.08; range = 1–146 estimates per species; 287 species with more than one estimate). We also complemented this dataset with ecotypic data for each species. Amphibians were grouped into six major ecotypes according to[40] ground-dwelling, fossorial, aquatic, semi-aquatic, stream-dwelling and arboreal. Cave specialists were excluded because they experience unique microclimatic conditions.

### Data-deficient species

Our objective was to assess the thermal tolerance of amphibians globally. However, the data compiled in ref. 3 are geographically and taxonomically biased. We therefore used a data-imputation procedure to infer the thermal tolerance of data-deficient species, totalling 5,203 species at a broad geographical coverage (524 species + 4,679 data-deficient species; ~60% of all described amphibian species, https://amphibiaweb.org; accessed in December 2023). We selected data-deficient species from a species list that matched the phylogeny from ref. 60 (7,238 species), was listed in the IUCN red list[69] along with geographic distribution data (5,792 species) and for which ecotypes were known (6,245 species). We did not consider Caecilians (order Gymnophiona) because, to our knowledge, heat-tolerance limits are unknown for all Caecilian species[3]. Of the 5,792 species for which we had distribution and phylogenetic data, 5,268 were data deficient for $CT_{max}$, of which 4,822 had a known ecotype. After removing Caecilians, we were left with 4,679 species to impute. We also supplemented our dataset with published body mass data retrieved from literature sources or estimated based on length–mass allometries[40,70,71]. We then estimated the geographical coordinates at which all extant species occurred in their IUCN distribution range at a 1° × 1° resolution to use for biophysical modelling (Extended Data Fig. 1).

### Data imputation

We developed a phylogenetic imputation procedure, here named Bayesian augmentation with chained equations (BACE). The BACE procedure combines the powers of Bayesian data augmentation and multiple imputation with chain equations (MICE[72]). In brief, we ran multiple iterative models using MCMCglmm[73] (v.2.34) and supporting functions from the hmi package[74]. In the first cycle, missing data were either taken as the arithmetic mean for continuous predictors, or randomly sampled from existing values for (semi)categorical predictors. Predicted (augmented) values from the models were then extracted from the response variables and used as predictor variables in the next models to predict other response variables. Ultimately, heat-tolerance limits were predicted using augmented data from all predictors. We ran five cycles where the data from one cycle were iteratively used in the next cycle, and estimations converged after the first cycle. Although the proportion of missing data was large (89.9%), imputations based on large amounts of missing data are common[13,75], and although estimate uncertainty increases with the proportion of missing data, as expected, simulation studies have shown estimations remain unbiased[76,77]. However, note that, although our approach took the uncertainty of missing data in the response or variable of interest ($CT_{max}$) into account, we used the most likely values for the predictors. While such an approach could underestimate the uncertainty in the response, point estimates should not be biased. In fact, our cross-validation approach demonstrated the ability of our models to predict back known experimental estimates with reasonable error (experimental mean ± s.d. = 36.19 ± 2.67; imputed mean = 35.93 ± 2.54; $r$ = 0.86; Extended Data Fig. 2).

Heat-tolerance limits were imputed based on the species' acclimatization temperatures, the duration of acclimatization, the ramping rate and end point used in assays, the medium used for measuring heat-tolerance limits (that is, ambient temperatures, water/body temperatures), and the life stage of the animals (adults or larvae) and their ecotype. These variables were correlated with amphibian heat-tolerance limits and were fitted as covariates in Bayesian linear mixed models. We also weighted heat-tolerance estimates based on the inverse of their sampling variance, accounted for phylogenetic non-independence using a correlation matrix of phylogenetic relatedness and fitted random intercepts for species-specific effects and phylogenetic effects, as well as their correlation with acclimatization temperatures (that is, random slopes). In other words, we modelled species-specific slopes (acclimatization response ratio) and partitioned the variance among phylogenetic and non-phylogenetic effects. We imputed data for adult amphibians assuming that they were acclimatized to the median, 5th or 95th percentile operative body temperatures experienced across their geographical range (see the 'Microenvironmental data and biophysical modelling' section) for a duration of 10 days, tested using a ramping rate of 1 °C min⁻¹ in a container filled with water, and for which thermal tolerance end point was recorded as the onset of spasms. These methodological parameters were the median values in the experimental dataset, or the most common values (mode). This enabled standardization of heat-tolerance limits for the comparative analysis[78–80]. In amphibians, the onset of spasms usually occurs after the loss of righting response[78], meaning that our estimates are conservative. Although we did include data from larvae in the training data, we imputed only data for adults to increase the comparability of our estimates.

For both known species and data-deficient species, we generated three ecologically relevant and standardized heat-tolerance estimates, and all analyses were built on these standardized imputed estimates. In total, we generated data for 5,203 species of amphibians (Extended Data Figs. 1 and 2). Notably, our imputed estimates are accompanied by standard errors, which provide estimates of uncertainty in the imputation, and errors were propagated throughout our analyses (see the 'Climate vulnerability analysis' section).

## Microenvironmental data and biophysical modelling

We used the package NicheMapR[81,82] (v.3.2.1) to estimate microenvironmental temperatures and hourly operative body temperatures in current (2006–2015) and projected climatic conditions (2 °C or 4 °C of global warming above pre-industrial levels). Operative body temperatures are the steady-state body temperatures that organisms would achieve in a given microenvironment, which can diverge significantly from ambient air temperatures due to, for example, radiative and evaporative heat exchange processes[19,20,83–88].

For each geographical location, we generated microclimatic temperatures experienced by amphibians on (1) a vegetated ground-level substrate (that is, terrestrial); (2) in aboveground vegetation (that is, arboreal); (3) in a water body (that is, aquatic) (Extended Data Fig. 1). For terrestrial and aquatic species, we simulated microenvironmental temperatures 1 cm above the surface. For arboreal species, we simulated microenvironmental temperatures 2 m aboveground, applied a reduction of 80% in windspeed to account for reduced wind due to vegetation[89,90] and assumed that 90% of the solar radiation was diffused due to canopy cover[78]. All microenvironmental projections were made using 85% shade to simulate animals in thermal refugia, that is, the microhabitats in which animals would retreat during the hottest times of the day. We did not model temperatures in the sun because ectothermic species most likely behaviourally thermoregulate by retreating to thermal refugia during extreme heat events[21]. Our calculations therefore represent conservative estimates of the vulnerability of amphibians to extreme temperature events.

For microclimatic temperature estimates, we used the micro_ncep function from NicheMapR[81] (v.3.2.1), which integrates six-hourly macroclimatic data from the National Center for Environmental Predictions (NCEP). This function also inputs from the microclima package[91] (v.0.1.0) to predict microclimatic temperatures after accounting for variation in radiation, wind speed, altitude, albedo, vegetation and topography. These data are downscaled to an hourly resolution, producing high-resolution microclimatic data. We used projected future monthly climate data from TerraClimate[23] to generate hourly projections assuming 2 °C or 4 °C of global warming above pre-industrial levels. These temperatures are within the range projected by the end of the century under low (Shared Socioeconomic Pathway SSP1–2.6 to SSP2–4.5) and high (SSP3–7.0 to SSP5–8.5) greenhouse gas emission scenarios, respectively[24]. TerraClimate projections use monthly data on precipitation, minimum temperature, maximum temperature, wind speed, vapour pressure deficit, soil moisture and downward surface shortwave radiation. These projections impose monthly climate projections from 23 CMIP5 global circulation models, as described previously[92]. The micro_ncep function then downscales monthly TerraClimate inputs to hourly by imposing a diurnal cycle to the data and imposes TerraClimate offsets onto the climatic data from NCEP. As the TerraClimate data is already bias-corrected, adding future climate projections onto the NCEP data did not require further bias correction. We ran all microclimatic estimations between 2005 and 2015 to match the range of pseudoyears available for TerraClimate future climate projections. We did not use a larger range of historical records and only used climate projections available in TerraClimate (that is, 2 °C and 4 °C) to reduce computational demands.

We then used microclimate estimates to generate hourly operative body temperatures using the ectotherm function in NicheMapR[82]. This modelling system has been extensively validated with field observations[93–95] (Extended Data Fig. 10). We modelled an adult amphibian in the shape of the leopard frog *Lithobates pipiens*, positioned 1 cm aboveground (or 2 m for arboreal species), and assumed that 80% of the skin acted as a free water surface (wet skin). Estimating body-mass-specific operative body temperatures for each grid cell, species and microhabitat was too computationally extensive, given the geographical and taxonomic scale of our study (464,871 local species occurrences). We

therefore ran the ectotherm models using the median body mass of the species assemblage in each geographical coordinate. When body mass was unknown, we ran models assuming a body mass of 8.4 g, the median assemblage-level body mass. Given that most amphibians in our dataset are small (median = 1.4 g, mean = 27.5 g), body temperatures equilibrate quickly with the environment, and operative body temperatures are probably representative of core body temperatures.

To model operative body temperatures in water bodies (for example, ponds or wetlands), we used the container model from NicheMapR. In contrast to previously mentioned calculations predicting steady-state temperatures, this approach accounts for transient temperature changes, capturing lags due to thermal inertia (that is, transient heat budget model[96,97]). For pond simulations, we modelled a container permanently filled with water (12 m width and 1.5 m depth) and decreased direct solar radiation to zero to simulate full shade. This modelling approach serves as a proxy for estimating the body temperature of ectotherms submerged in water bodies such as ponds or wetlands, which was validated with field measurements[39,94]. Ground-level and water temperatures were modelled for all species regardless of their ecotype (apart from paedomorphic salamanders that were only assessed in aquatic environments) because arboreal and terrestrial species may retreat on land or in water occasionally. Temperatures in aboveground vegetation were only estimated for arboreal and semi-arboreal species, as reaching 2 m height in vegetation requires a morphology adapted to climbing. Our biophysical models assume that shaded microhabitats are available to species throughout their range. While this may not hold true, fine-scaled distribution of these microenvironments are not available at global scales. Moreover, assuming that these microenvironments are available serves a functional role; it provides a best-case scenario that is useful for comparative analyses and offers actionable insights for conservation. For example, reduced exposure to overheating events in aquatic relative to terrestrial environments would suggest that preserving ponds and wetlands may be critical in buffering the impacts of climate change on amphibians.

We then estimated, for each geographical coordinate, the maximum daily body temperature and the mean and maximum weekly maximum body temperature experienced in the 7 days before each given day to account for acclimatization responses and to assess climate vulnerability metrics[18] (see the 'Climate vulnerability analyses' section). We only used data for the 91 warmest days (that is, warmest quarter) of each year, as we were interested in the responses of amphibians to extreme heat events[18]. Note that data from the year 2005 was excluded a posteriori as a burn-in to remove the effects of initial conditions on soil temperature, soil moisture and pond calculations. Thus, our analyses are based on 910 days (91 days per year in the range 2006–2015) for each climatic scenario (current climate, 2 °C above pre-industrial levels, 4 °C above pre-industrial levels).

We also used maximum daily body temperatures on terrestrial conditions to calculate the median, 5th percentile and 95th percentile maximum body temperature experienced by each species across their range of distribution. These values were used as acclimatization temperatures in the training data to calibrate the data imputation with ecologically relevant environmental temperatures (see the 'Data imputation' section); while maximizing the range of temperatures used to infer the plasticity of heat-tolerance limits (see the 'Climate vulnerability analysis' section).

## Climate vulnerability analysis

Using the imputed data, we fitted an individual meta-analytic model for each species to estimate the plasticity of imputed heat-tolerance limits ($CT_{max}$) to changes in operative body temperatures using the metafor package[98] (v.4.2-0). $CT_{max}$ was used as the response variable, acclimatization temperature (that is, median, 5th percentile, or 95th percentile daily maximum body temperature experienced by a species across its distribution range) was used as the predictor variable

and imputed estimates were weighted based on their standard error. From these models, we used out-of-sample model predictions (using the predict function) to estimate the $CT_{max}$ of each species in each $1° × 1°$ grid cell across their distribution range in different warming scenarios, based on predicted mean weekly body temperatures. Specifically, we assumed that species were, on any given day, acclimatized to the mean daily body temperature experienced in the 7 days before[18]. Thus, $CT_{max}$ was simulated as a plastic trait that varied daily as animals acclimatize to new environmental conditions (Extended Data Fig. 1). While evidence in small amphibians suggests that the full acclimatization potential is reached within 3–4 days[99–101], other evidence points to some variation after longer periods[102]. We therefore chose 7 days to reflect that some amphibians may require longer to acclimatize. As we used out-of-sample model predictions, we propagated errors from the imputation when estimating the predicted $CT_{max}$ across geographical coordinates. Predicted $CT_{max}$ values and their associated standard errors therefore reflect variation in both the imputation procedure and the estimation of plastic responses. Our approach to accounting for plasticity assumes that plasticity is homogeneous within species and ignores the possible influence of local adaptation. However, given the low variability in plasticity among species (mean acclimatization response ratio ± s.d. = 0.134 ± 0.008; range = 0.049–0.216; $n = 5,203$), the lack of evidence for latitudinal variation in plasticity[27,30,103], the high phylogenetic signal in thermal tolerance (Pagel's $\lambda$ (ref. 104) = 0.95 (95% credible interval 0.91–0.98); see the 'Cross-validation and sensitivity analyses' section) and evidence for slow rates of evolution and physiological constraints on $CT_{max}$[51,52], geographical variation in thermal tolerance and plasticity is unlikely to have a major influence on our results.

We next estimated the vulnerability of amphibians to global warming using three metrics (Extended Data Fig. 1). First, we calculated the difference between $CT_{max}$ and the maximum daily body temperature, that is, the TSM (TSM, sensu[6]). We calculated weighted means and standard errors (sensu[105]) of TSMs across years to estimate the mean difference between $CT_{max}$ and the maximum temperature during the warmest quarters. Using TSM averaged from the maximum temperature of the warmest quarter is common in the literature[26–28]. Second, we calculated the number of days on which the maximum daily operative body temperature exceeded the $CT_{max}$ across the warmest quarters of 2006–2015, that is, the number of overheating events. To propagate the uncertainty, we calculated the mean probability that daily operative body temperatures exceeded the predicted distribution of $CT_{max}$ (using the dnorm function). Note that the standard error (s.d. of estimates) of simulated $CT_{max}$ distributions was restricted to 1 (that is, simulating distributions within ~3 °C of the mean) to avoid inflating overheating probabilities due to large imputation uncertainty (compare with ref. 75; see the 'Cross-validation and sensitivity analyses' section; Extended Data Fig. 8). We then multiplied the mean overheating probability by the total number of simulated days (910) to estimate the number of overheating events and their associated standard error using properties of the binomial distribution. Third, we calculated the binary probability (0/1) that species overheat for at least 1 day across the 910 days surveyed (warmest quarters of 2006–2015). The latter two metrics provide a finer resolution than TSMs, as they capture daily temperature fluctuations and potential overheating events[18].

## Macroecological patterns

The objective of this study was to characterize the vulnerability of amphibians to global warming. We investigated patterns at the level of local species occurrences (presence of a given species in a $1° × 1°$ grid cell based on IUCN data), allowing one to identify specific populations and species that may be more susceptible to heat stress and direct targeted research efforts. We also analysed data at the assemblage level, the species composition within a grid cell. In such cases, we calculated the weighted mean and standard error of TSM (sensu[104]) across species in each grid cell. Assemblage-level analyses allow one to identify areas

containing a higher number of vulnerable species, offering actionable insights for broader-scale conservation initiatives.

We used the gamm4 package[105] to fit generalized additive mixed models against latitude. For local species occurrences, we fitted latitude as a fixed factor, and nested genus and species identity as random terms to account for phylogenetic non-independence. Note that we did not include family as a random term because models failed at estimating higher taxonomic variation. While better methods exist to model phylogenetic patterns, generalized additive linear models do not allow for phylogenetic correlation matrices, and other functions such as brms[106] surpassed our computational time and memory limits. Nevertheless, imputed estimates already reflect variation due to phylogeny (see the 'Data imputation' section), and phylogeny was further modelled when deriving mean estimates in each microhabitat and climatic scenario (see below). We fitted models using the three metrics as response variables independently: the TSM, overheating risk and number of overheating events. The former was modelled using a Gaussian distribution of residuals, overheating risk was modelled using a binomial error structure and the latter using a Poisson error structure. Note that overheating risks were rounded to integer values to fit a Poisson distribution. TSM estimates were weighted by the inverse of their sampling variance to account for the uncertainty in the imputation and predictions across geographical coordinates. We fitted separate models for each climatic scenario (current climate, 2 °C above pre-industrial levels, 4 °C above pre-industrial levels) and microhabitat (terrestrial, aquatic, arboreal).

To investigate the mean TSM in each microhabitat and climatic scenario, we fitted models with the interaction between microhabitat and climatic scenario as a fixed effect using MCMCglmm[73] (v.2.34) and flat, parameter-expanded priors. In these models, we weighted estimates based on the inverse of their sampling variance, species identity was fitted as a random effect and we accounted for phylogenetic non-independence using a variance–covariance matrix of phylogenetic relatedness (calculated from the consensus tree of ref. 60). To investigate the overall overheating risk and the number of overheating events in each condition, we attempted to fit models in MCMCglmm, but these models failed to converge. We therefore fitted Poisson and binomial models using lme4 (v.1.1-33)[107] and nested genus, species and observation as random terms. We used similar Poisson models to investigate the relationship between the number of overheating events and TSMs. While the mean estimates from these simpler models should be unbiased, estimate uncertainty is probably underestimated[108].

We also investigated patterns of climate vulnerability at the assemblage level. We calculated the weighted average of TSM and overheating risk in each 1° grid cell (14,091, 14,090 or 6,614 grid cells for terrestrial, aquatic and arboreal species, respectively), and mapped patterns geographically. Averaging overheating risk effectively returned the proportion of species overheating in each coordinate, and we also calculated the number of species overheating in each grid cell. For assemblage-level models, we fitted Gaussian, binomial or Poisson models as described above, but without taxon-level random effects because these cannot be modelled at the assemblage level. All models were fitted without a contrast structure to estimate mean effects in each microhabitat and climatic scenario, and with two-sided contrasts to draw comparisons with current terrestrial conditions.

## Cross-validation and sensitivity analyses

We assessed the accuracy of the data imputation procedure using a cross-validation approach. Specifically, we removed heat-tolerance estimates for 5% of the species in the experimental data and 5% of the data-deficient species (maintaining the same proportion of missing data) and assessed how well experimental values could be predicted from the models. Of relevance, we only removed data that were comparable to the data that were imputed. That is, data from adult animals that were tested using a ramping rate of 1 °C min⁻¹, and where thermal limits were recorded as the onset of spasms. While we could have trimmed

any data entry in the experimental data, validation of the imputation performance can be achieved only by comparing comparable entries, and imputing data from species tested in unusual settings would naturally result in large errors. In total, we cross-validated experimental estimates for 77 species.

We investigated alternative ways to (1) calculate TSMs; (2) account for acclimatization responses; and (3) control for prediction uncertainty (Extended Data Figs. 6–8). In our study, we projected $CT_{max}$ estimates assuming that animals were acclimatized to the mean weekly temperature experienced prior to each day. We also assessed the climate vulnerability of amphibians assuming they were acclimatized to weekly maximum body temperatures (compare with ref. 18), which reflects more conservative estimates (Extended Data Fig. 7). We also calculated TSMs as the difference between the maximum (or 95th percentile; compare with ref. 4) hourly body temperature experienced by each population and their predicted $CT_{max}$ to investigate the consequences of averaging temperatures when calculating TSMs (Extended Data Fig. 6). To increase the comparability of our estimations with similar studies[4], we also calculated climate vulnerability metrics more conservatively. Specifically, we excluded temperature data falling below the 5th percentile and above the 95th percentile body temperature for each population to mitigate the impact of outliers (Extended Data Fig. 6). However, extreme weather events, which are typically captured by these outlier values, are the very phenomena that are most likely to precipitate mortality events[16,17]. Omitting these outliers could therefore obscure the ecological significance of extreme temperatures, thereby underestimating true overheating risks. To estimate overheating probabilities, we calculated the mean daily probability that operative body temperatures exceeded the predicted distribution of $CT_{max}$ and restricted the s.d. of simulated distributions to 1 (that is, within around 3 °C of the mean) to avoid inflating overheating probability for observations with large uncertainty. We also provided alternative results (Extended Data Fig. 8) where the s.d. of $CT_{max}$ was restricted to the biological range, that is, the s.d. of the distribution of all $CT_{max}$ estimates across species (range = 1.84–2.17). We also provide a sensitivity analysis where overheating risk was positive only when the 95% confidence intervals of predicted overheating days did not overlap with zero (Extended Data Fig. 8).

We also investigated the influence of different parameters of our biophysical models (that is, shade and burrow availability, height in aboveground vegetation, solar radiation, wind speed, pond depth) on predicted vulnerability risks (Extended Data Fig. 9). Specifically, we modelled the responses of the species at highest risk in terrestrial and aquatic conditions, *N. myrmecoides*, in its most vulnerable location (latitude, longitude = −9.5, −69.5). For terrestrial conditions, we modelled the response of amphibians with different body sizes (0.5, 4.28 or 50 grams), and with different levels of exposure to open habitat conditions. Specifically, we modelled an amphibian exposed to 50% of shade to simulate an open habitat lightly covered by vegetation, and inferred temperatures at different soil depths (2.5, 5, 10, 15 or 20 cm underground). For aquatic conditions, we adjusted pond depths to simulate a very shallow pond (50 cm) and compared it to deeper ponds (depth of 1.5 or 3 m). For arboreal conditions, we modelled the responses of *P. ockendeni* in its most vulnerable location (−4.5, −71.5), and adjusted the height in aboveground vegetation (0.5, 2 or 5 m), the percentage of radiation diffused by vegetation (50%, 75% or 90% of radiation diffused) and the percentage of wind speed reduced by vegetation (0%, 50% or 80% of wind speed reduced by vegetation). We did not estimate the influence of these parameters on all species and at all locations owing to the scale of our study, but these results should provide insights into how varying microenvironmental features and biological characteristics may impact our general conclusions. Our results were generally robust to changes in model parameters, although amphibians are likely to experience more overheating events in open habitats[6,41] and shallow ponds, and lower risks in underground conditions[109] (Extended Data Fig. 9).

We also compared our predictions of operative body temperatures with field body temperature measurements. We extracted night-time (18:00–00:30) field body temperatures measured for 11 species of frogs in Mexico (21.48° N, −104.85° W; and 21.45° N, −105.03° W) between June and October of 2013 and 2015 from table 1 of ref. 109. We chose this study because it provided the data and location of body temperature measurements, covered multiple species from different sites and matched our study timeframe (2006–2015). We then compared these estimates with hourly operative body temperatures predicted in shaded terrestrial conditions at the same dates and time windows (Extended Data Fig. 10). We confirmed that predicted operative body temperatures were comparable to field body temperatures measured in some wild frogs (Extended Data Fig. 10), and we invite additional validations with other species in different geographical areas.

Finally, we confirmed the presence of a phylogenetic signal in the experimental dataset by fitting a Bayesian linear mixed model using all complete (no missing data) predictors (that is, acclimatization temperature, end point, acclimatization status, life stage and ecotype) in MCMCglmm. We accounted for phylogenetic non-independence using a correlation matrix of phylogenetic relatedness and fitted random intercepts for non-phylogenetic species effects. The phylogenetic signal (Pagel's $\lambda$[104], which is equivalent to phylogenetic heritability[110,111]), was calculated as the proportion of variance explained by phylogenetic effects relative to the total non-residual variance.

Results from all statistical models and additional data visualizations are available at GitHub (https://p-pottier.github.io/Vulnerability_amphibians_global_warming/).

### Inclusion and ethics statement

This study did not involve researchers who collected the original data. All data used for the analyses were taken from a previous data compilation[3], and the original references on which all analyses were built on are provided[22,33,35,42,53,78,99,100,112–316].

### Reporting summary

Further information on research design is available in the Nature Portfolio Reporting Summary linked to this article.

## Data availability

Raw and processed data are available at GitHub (https://github.com/p-pottier/Vulnerability_amphibians_global_warming), and archived permanently in Zenodo[317]. Note that some intermediate data files were too large to be shared through GitHub, but are available through Jagiellonian University's repository[318]. TerraClimate data (https://www.climatologylab.org/terraclimate.html) and NCEP data (https://psl.noaa.gov/thredds/catalog/Datasets/ncep.reanalysis2/gaussian_grid/catalog.html) are available online.

## Code availability

All code needed to reproduce the analyses is available at GitHub (https://github.com/p-pottier/Vulnerability_amphibians_global_warming) and archived permanently in Zenodo[317].

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

**Acknowledgements** This study was funded by UNSW Scientia Doctoral Scholarships awarded to P. Pottier, S.B. and P. Pollo. S.N. was supported by the Australian Research Council (ARC) Discovery Project (DP210100812). S.M.D. was supported by the ARC Discovery Early Career Award (DE180100202). M.R.K. was supported by the ARC Discovery Project DP200101279. We thank the authors of the original studies who provided the groundwork for our analyses. We thank the Bedegal people, the traditional custodians of the land on which this work was primarily conducted.

**Author contributions** This study was conceptualized by P. Pottier, M.R.K., S.B., S.M.D. and S.N. All data manipulation and analyses were performed by P. Pottier (with conceptual and technical input from S.M.D. and S.N. for the imputation methods and statistical analyses, and M.R.K., A.R.G., J.E.R. and N.C.W. for the biophysical modelling and climate vulnerability analyses). All code was reviewed by N.C.W., A.R.G. and J.E.R. following the recommendations of a previous study[319]. Ecotype information was collected by N.C.W., P. Pollo and A.N.R.-V. P. Pottier, N.C.W. and S.M.D. contributed to data visualization. P. Pottier wrote the initial draft, and all of the authors were involved in the review and editing. P. Pottier oversaw the project administration, and S.M.D. and S.N. were in charge of the supervision.

**Funding** Open access funding provided through UNSW Library.

**Competing interests** The authors declare no competing interests.

**Additional information**
**Correspondence and requests for materials** should be addressed to Patrice Pottier.

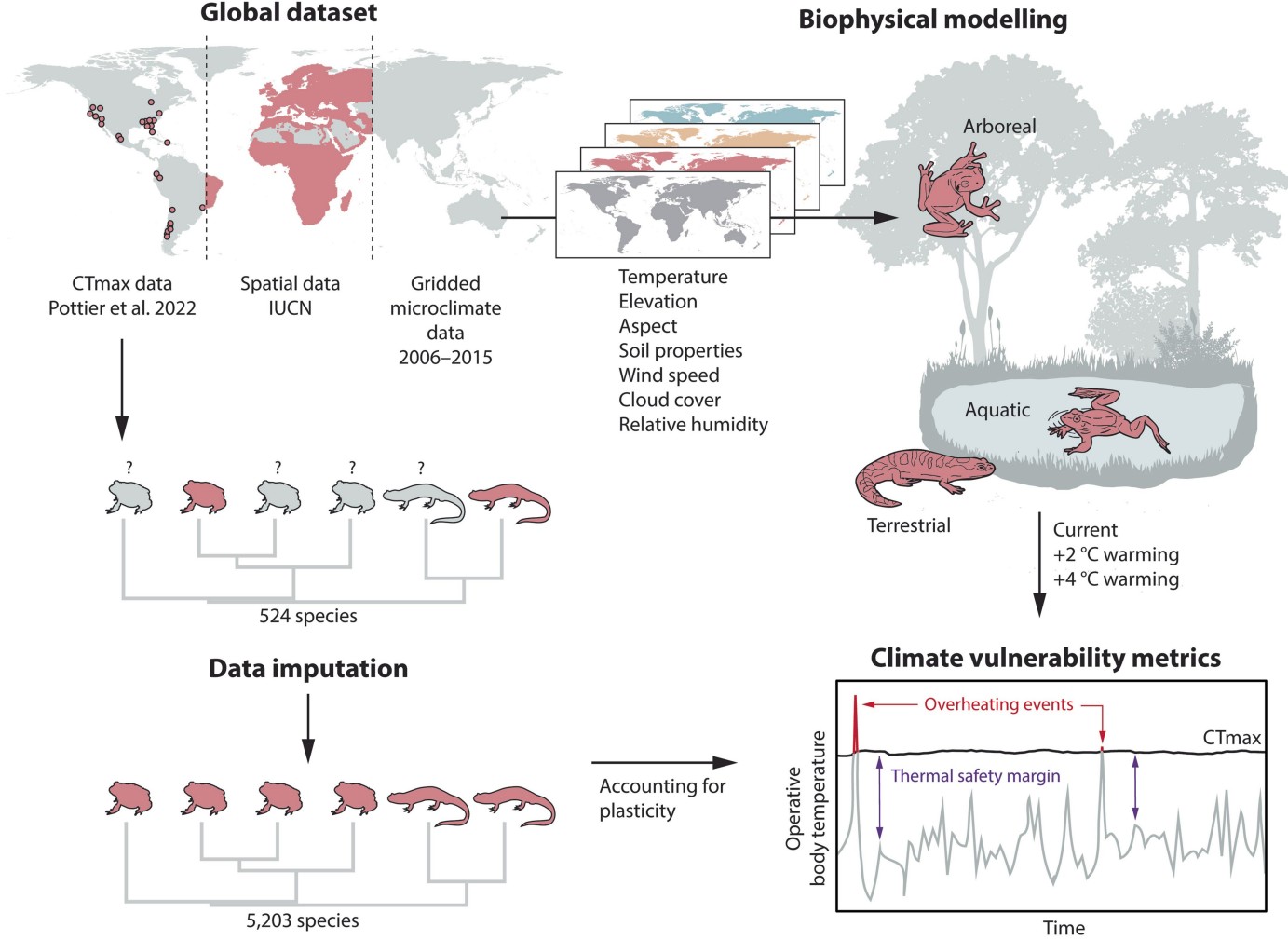

**Extended Data Fig. 1 | Methods used to assess the vulnerability of amphibians to global warming.** Conceptual overview of the methods employed to assess the vulnerability of amphibians to global warming.

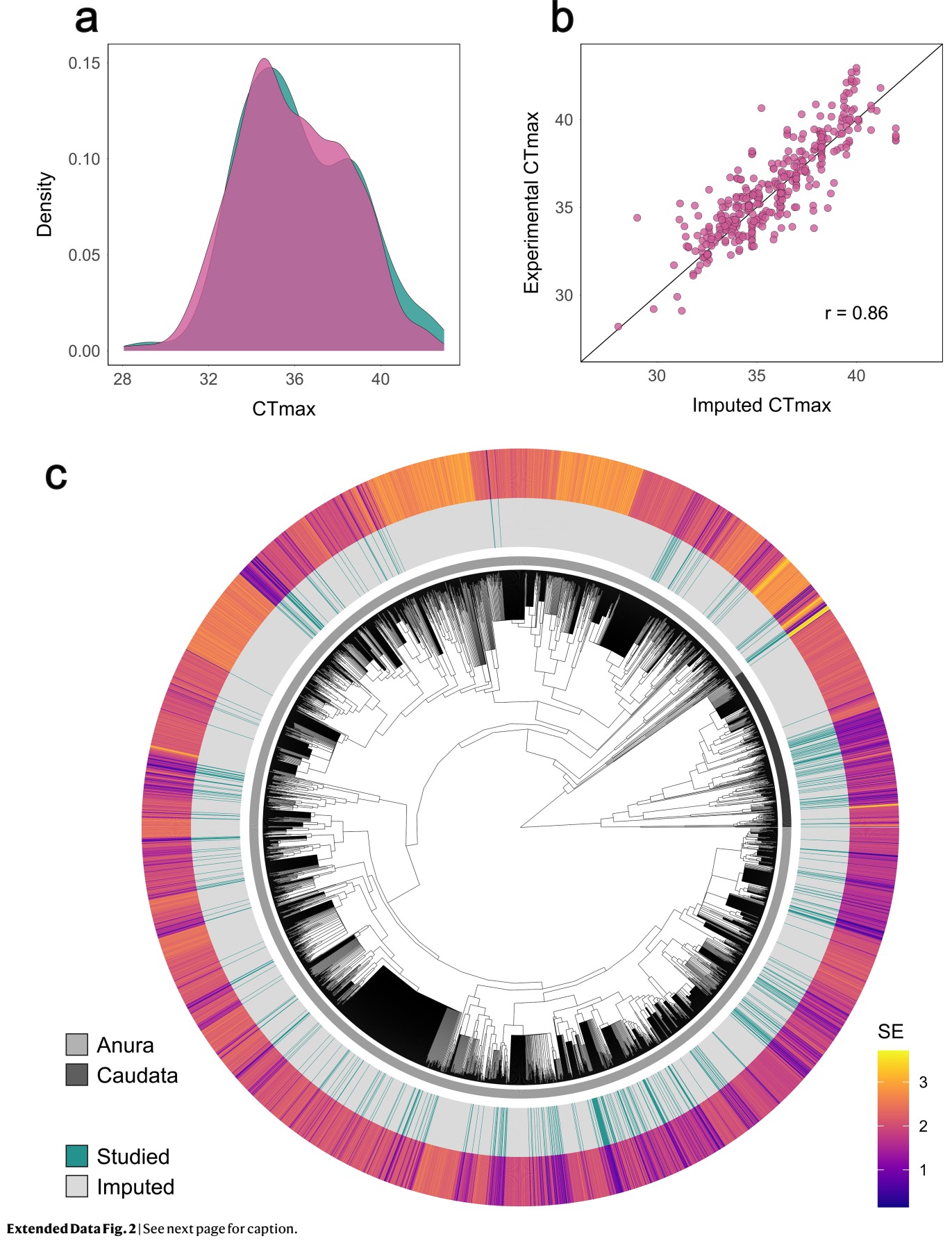

**Extended Data Fig. 2** | See next page for caption.

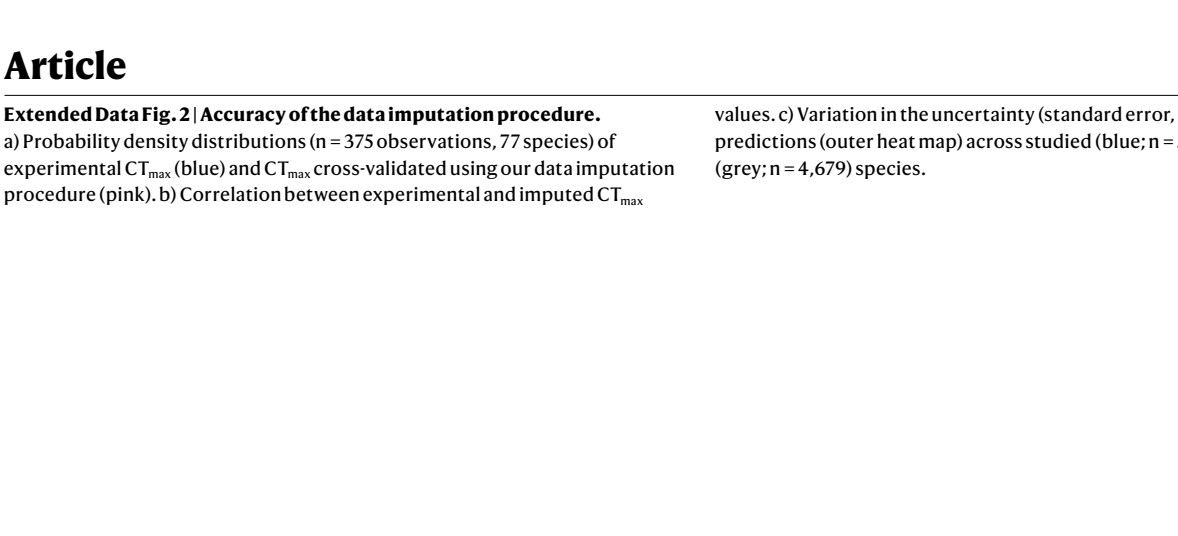

**Extended Data Fig. 2 | Accuracy of the data imputation procedure.**
a) Probability density distributions (n = 375 observations, 77 species) of experimental $CT_{max}$ (blue) and $CT_{max}$ cross-validated using our data imputation procedure (pink). b) Correlation between experimental and imputed $CT_{max}$ values. c) Variation in the uncertainty (standard error, SE) of imputed $CT_{max}$ predictions (outer heat map) across studied (blue; n = 524) and imputed (grey; n = 4,679) species.

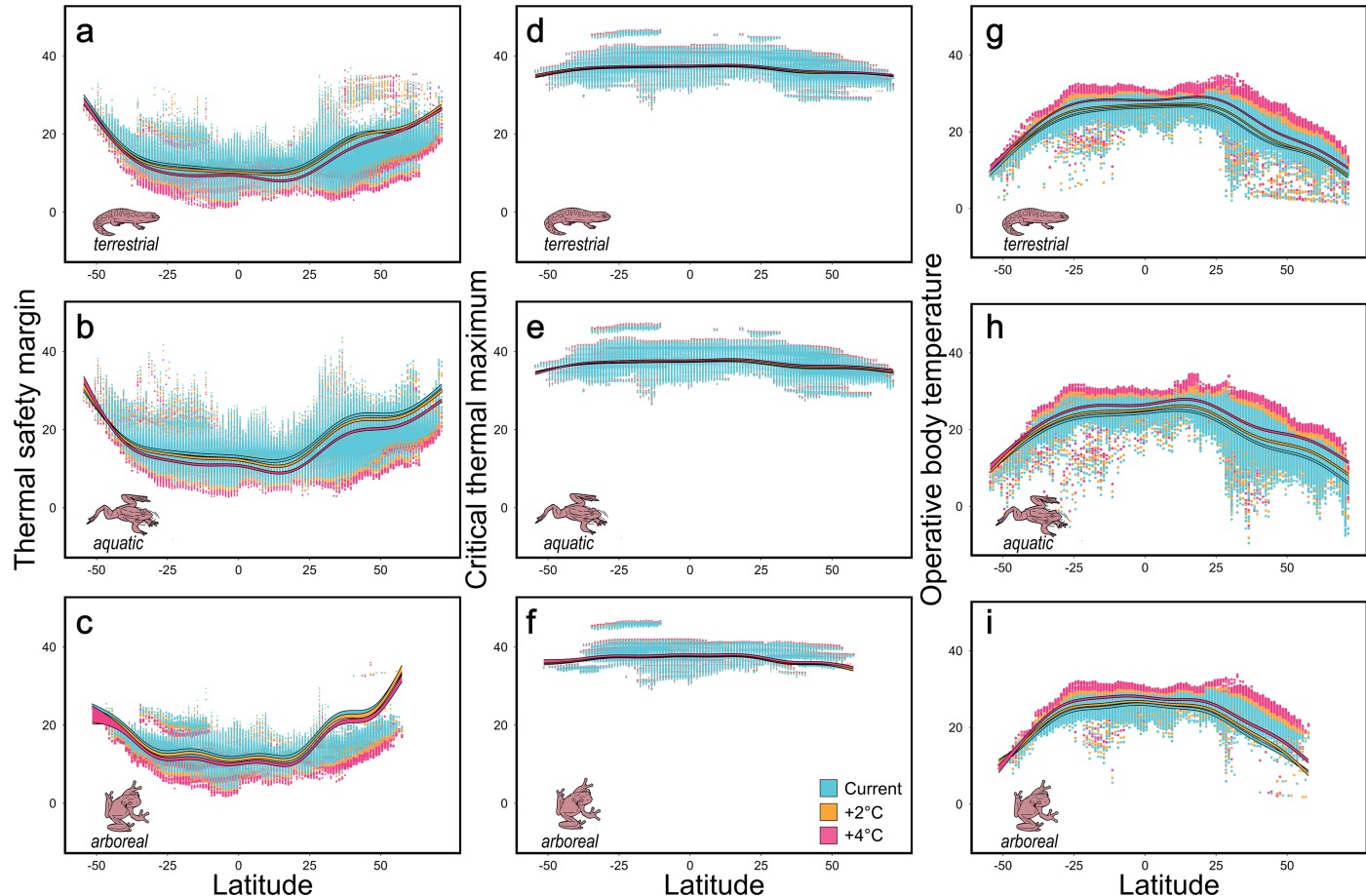

**Extended Data Fig. 3 | Thermal safety margin, critical thermal maximum, and operative body temperatures in different microhabitats and climatic scenarios.** Weighted mean thermal safety margins (TSM; a-c), critical thermal maximum ($CT_{max}$; d-f) and operative body temperatures (g-i) in terrestrial (a,d,g), aquatic (b,e,h) and arboreal (c,f,i) microhabitats are depicted in current microclimates (blue data points), or assuming 2 °C and 4 °C of global warming above pre-industrial levels (orange, and pink data points, respectively) across latitudes, for each local species occurrence (n = 203,853 for terrestrial species; n = 204,808 for aquatic species; n = 56,210 for aquatic species). Lines represent 95% confidence intervals of model predictions from generalized additive mixed models. $CT_{max}$ and TSM estimates are scaled by precision (1/s.e.), with smaller points indicating higher uncertainty. Each point represents a species in a given grid cell.

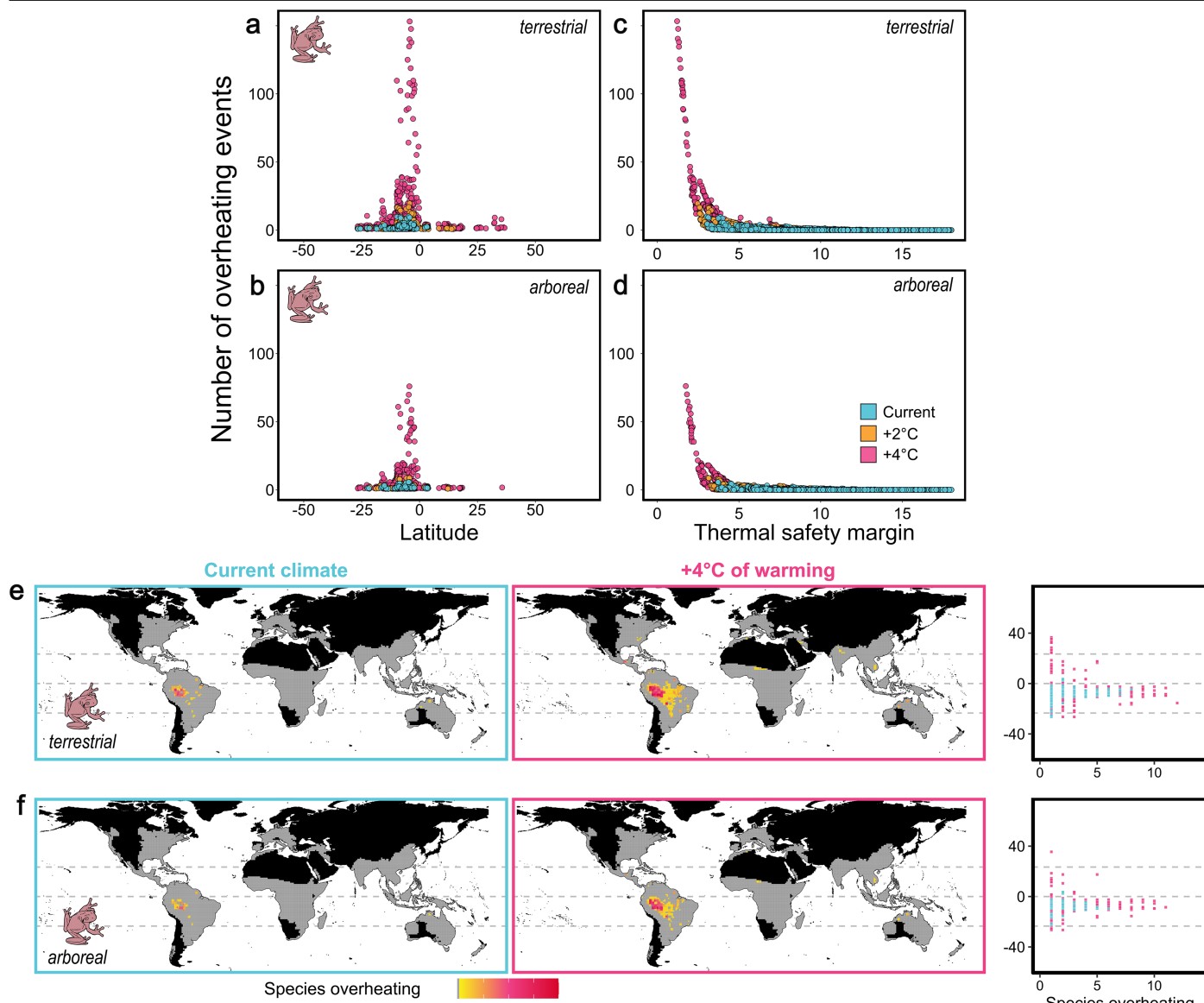

**Extended Data Fig. 4 | Vulnerability of arboreal amphibians in terrestrial and arboreal microhabitats.** Depicted are the number of overheating events experienced by arboreal species across latitudes (a-b) and in relation to thermal safety margins (c-d) in terrestrial (a-c) and arboreal microhabitats (b-d). The number of overheating events were calculated based on the mean probability that daily maximum temperatures exceeded $CT_{max}$ during the warmest quarters of 2006–2015 for each species in each grid cell (i.e., local species occurrence; n = 203,853 for terrestrial species; n = 204,808 for aquatic species; n = 56,210 for aquatic species). Blue points depict the number of overheating events in historical microclimates, while orange and pink points depict the number of overheating events assuming 2 °C and 4 °C of global warming above pre-industrial levels, respectively. In panel a) and b), only the species predicted to overheat for at least one day are displayed. The number of arboreal species predicted to experience overheating events in terrestrial (e) and arboreal (f) microhabitats in each assemblage is also depicted. The number of species overheating was assessed as the sum of species overheating for at least one day in the period surveyed (warmest quarters of 2006–2015) in each assemblage (1-degree grid cell; n = 14,090 for terrestrial species; n = 14,091 for aquatic species; n = 6,614 for arboreal species). Black colour depicts areas with no data, and grey colour assemblages without species at risk. The right panel depicts latitudinal patterns in the number of species predicted to overheat in current climates (blue) or assuming 4 °C of global warming above pre-industrial levels (pink). Dashed lines represent the equator and tropics. Few species (n = 11) were predicted to experience overheating events in water bodies, and hence are not displayed.

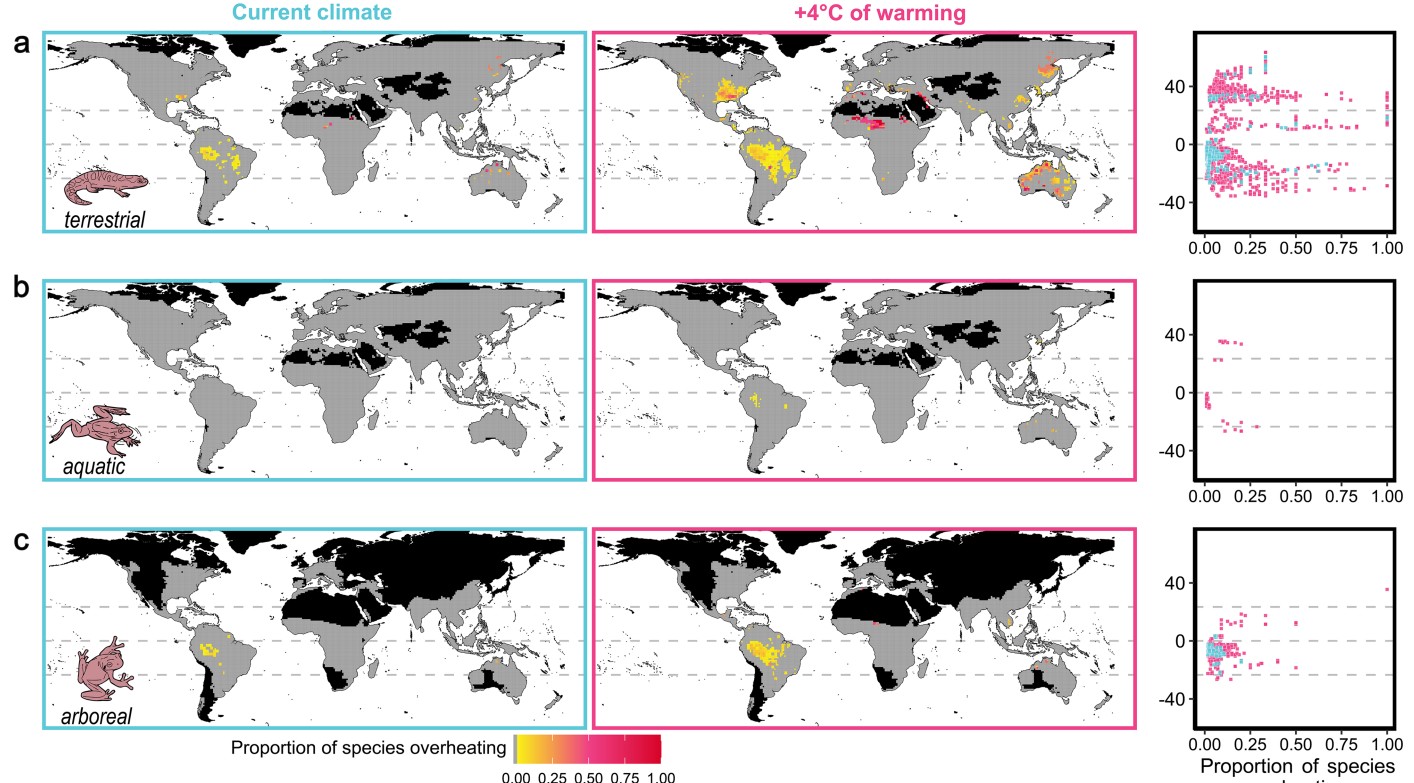

**Extended Data Fig. 5 | Proportion of species predicted to experience overheating events in terrestrial (a), aquatic (b), and arboreal (c) microhabitats.** The proportion of species overheating was assessed as the sum of species overheating for at least one day in the period surveyed (warmest quarters of 2006–2015) divided by the number of species in each assemblage (1-degree grid cell; n = 14,090 for terrestrial species; n = 14,091 for aquatic species; n = 6,614 for arboreal species). Black colour depicts areas with no data, and grey colour assemblages without species at risk. The right panel depicts latitudinal patterns in the proportion of species predicted to overheat in current climates (blue) or assuming 4 °C of global warming above pre-industrial levels (pink). Dashed lines represent the equator and tropics.

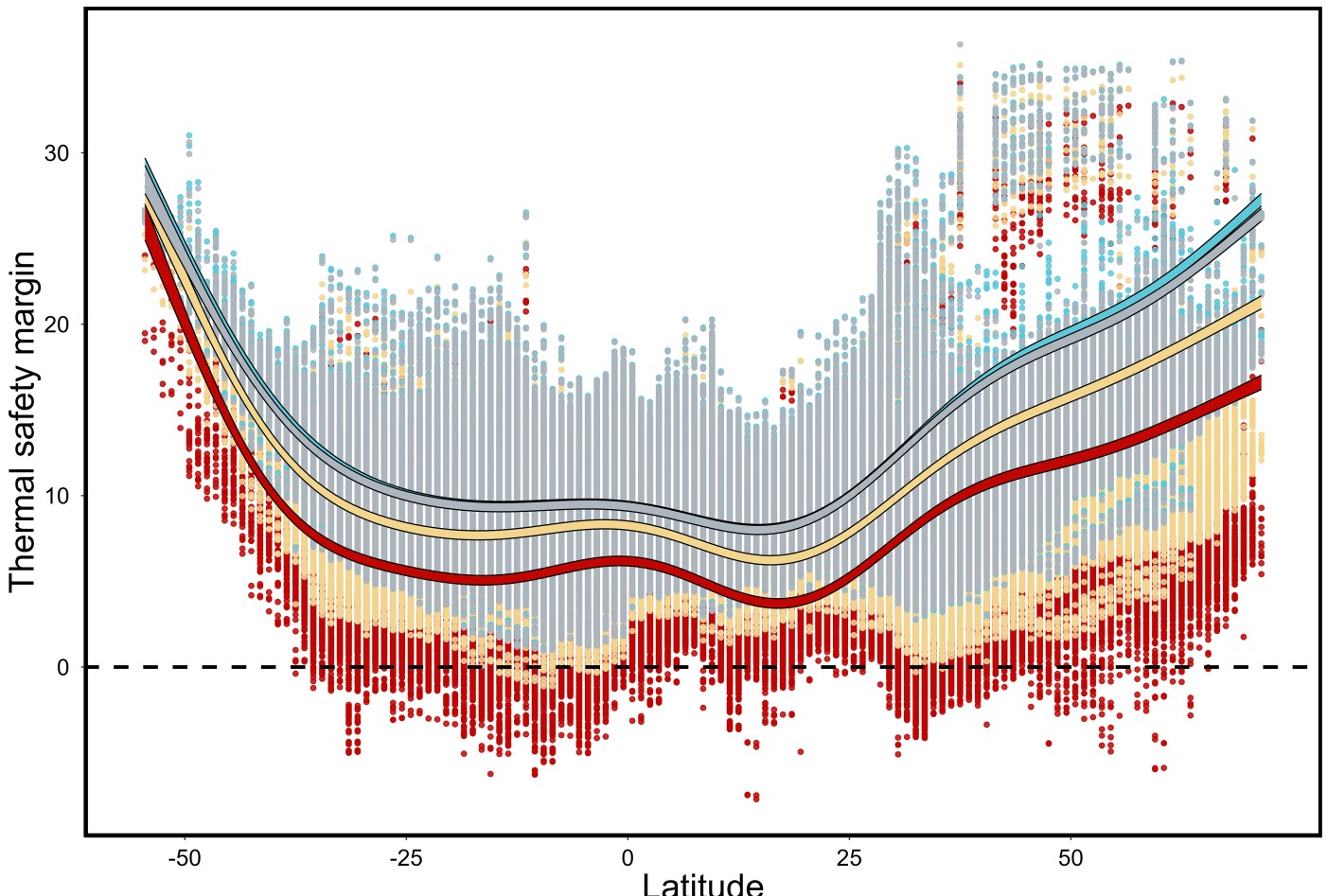

**Extended Data Fig. 6 | Variation in thermal safety margins calculated using different assumptions.** Thermal safety margins (TSM) were calculated as the mean difference between $CT_{max}$ and the predicted operative body temperature in full shade during the warmest quarters of 2006–2015 (grey), as the mean difference between $CT_{max}$ and the predicted operative body temperature in full shade during the warmest quarters of 2006–2015 excluding body temperatures falling outside the 5% and 95% percentile temperatures (blue), as the difference between the 95% percentile operative body temperature and the corresponding $CT_{max}$ (yellow), or as the difference between the maximum operative body temperature and the corresponding $CT_{max}$ (red). Lines represented 95% confidence interval ranges predicted from generalized additive mixed models. This figure was constructed assuming ground-level microclimates occurring under 4 °C of global warming above pre-industrial levels, for each species in each grid cell (i.e., local species occurrences; n = 203,853 for terrestrial species; n = 204,808 for aquatic species; n = 56,210 for aquatic species).

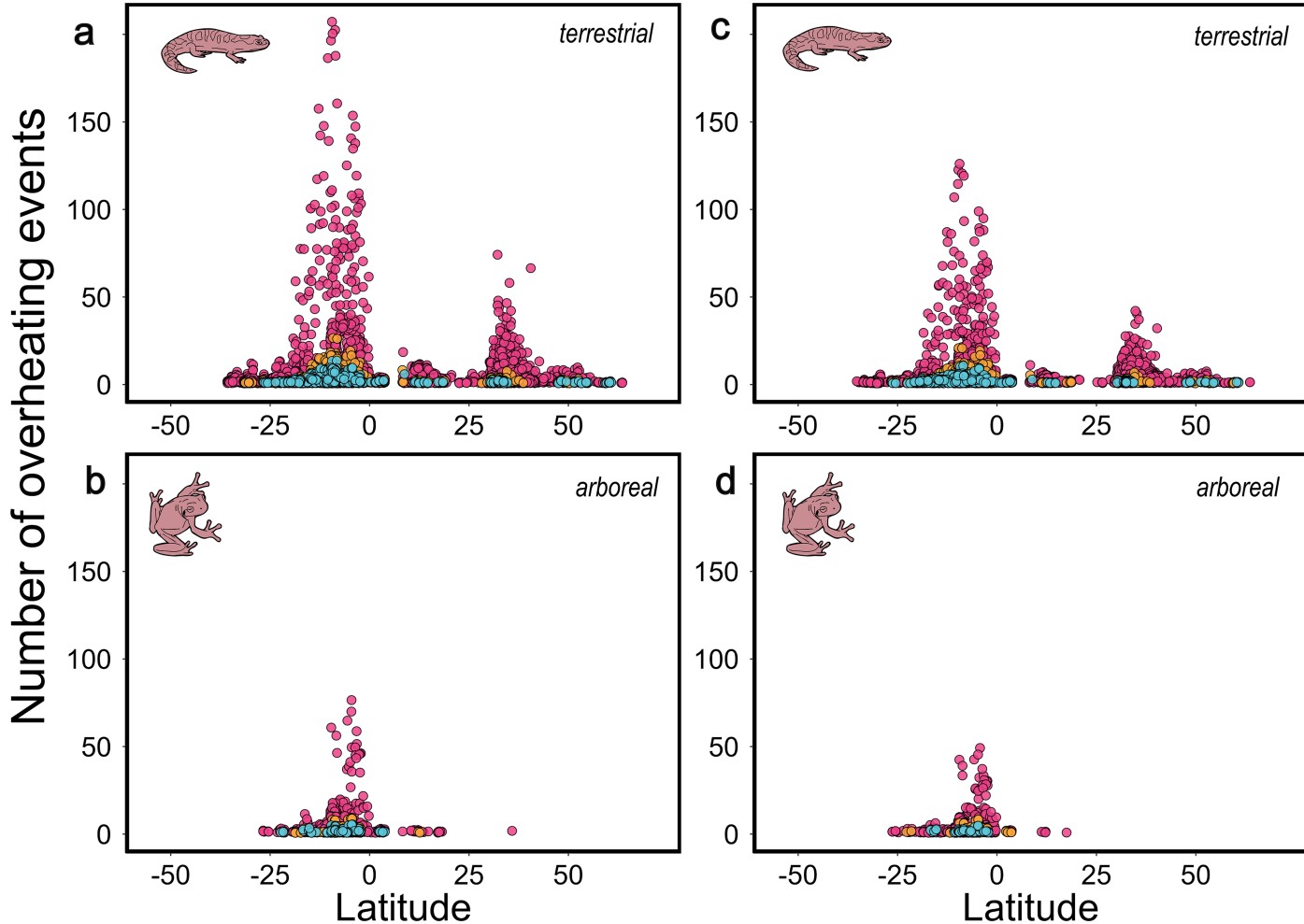

**Extended Data Fig. 7 | Latitudinal variation in the number of overheating events when animals are acclimated to the mean (a,b) or maximum (c,d) weekly body temperature experienced in the seven days prior in terrestrial (a,c) and arboreal (b,d) microhabitats.** The number of overheating events (days) were calculated based on the mean probability that daily maximum temperatures exceeded $CT_{max}$ during the warmest quarters of 2006–2015 for each species in each grid cell (i.e., local species occurrences; n = 203,853 for terrestrial species; n = 204,808 for aquatic species; n = 56,210 for aquatic species). Blue points depict the number of overheating events in historical microclimates, while orange and pink points depict the number of overheating events assuming 2 °C and 4 °C of global warming above pre-industrial levels, respectively. For clarity, only the species predicted to experience overheating events across latitudes are depicted.

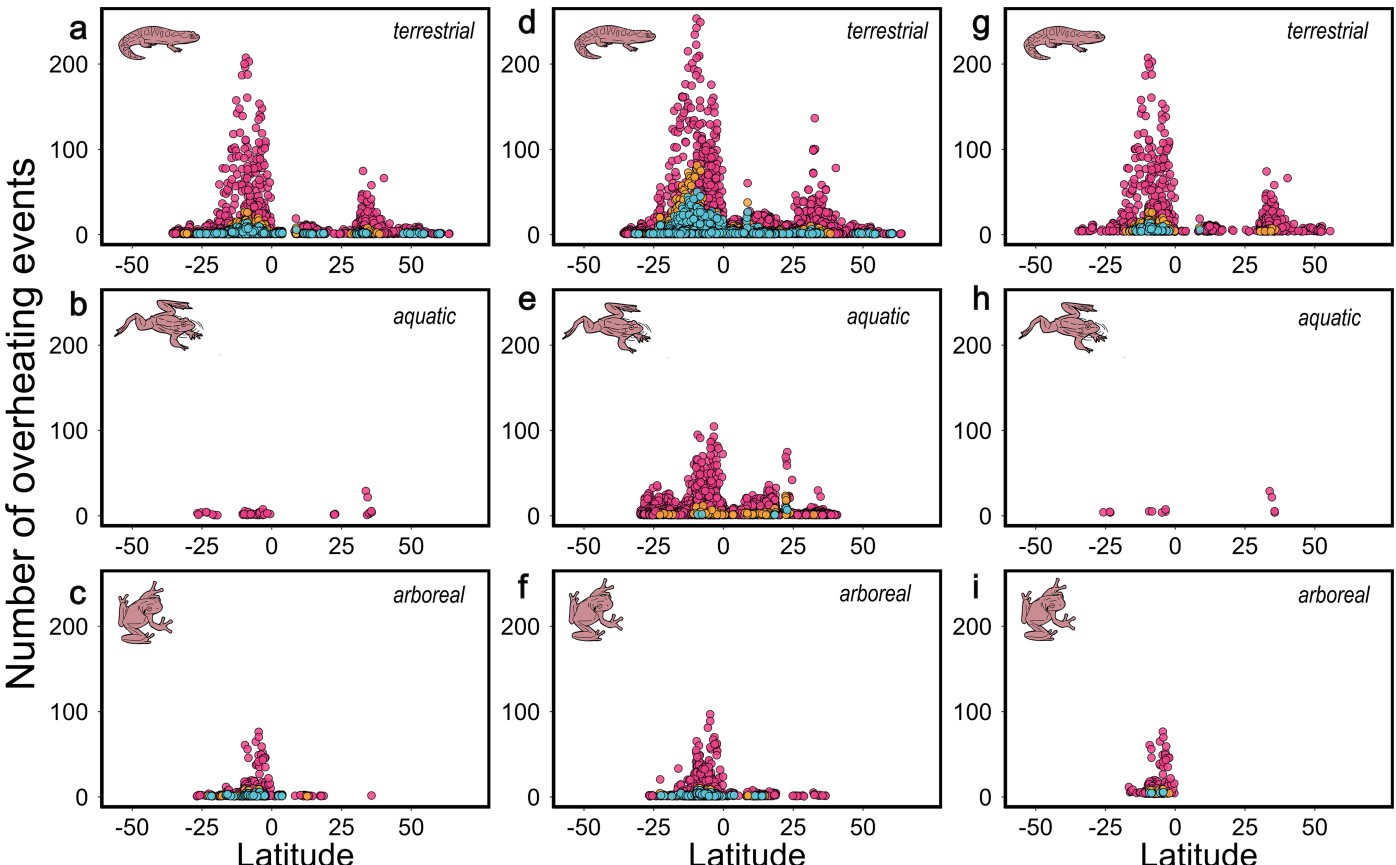

**Extended Data Fig. 8 | Latitudinal variation in the number of overheating events using regular (a,b,c), uncertain (d,e,f), or conservative estimates (g,h,i) in terrestrial (a,d,g), aquatic (b,e,h) and arboreal (c,f,i) microhabitats.** The number of overheating events (days) were calculated based on the mean probability that daily maximum temperatures exceeded CT_max during the warmest quarters of 2006–2015 for each species in each grid cell (i.e., local species occurrences; n = 203,853 for terrestrial species; n = 204,808 for aquatic species; n = 56,210 for aquatic species). Uncertain estimates are those where daily overheating probabilities were calculated based on broad predicted distributions of CT_max (i.e., simulated over the whole *"biological range"*), likely inflating overheating probabilities for observations with large uncertainty. Conservative estimates are those when overheating risk was considered only when the 95% confidence intervals of the predicted number of overheating events did not overlap with zero (e,f). Blue points depict the number of overheating events in historical microclimates, while orange and pink points depict the number of overheating events assuming 2 °C and 4 °C of global warming above pre-industrial levels, respectively. For clarity, only the species predicted to experience overheating events across latitudes are depicted.

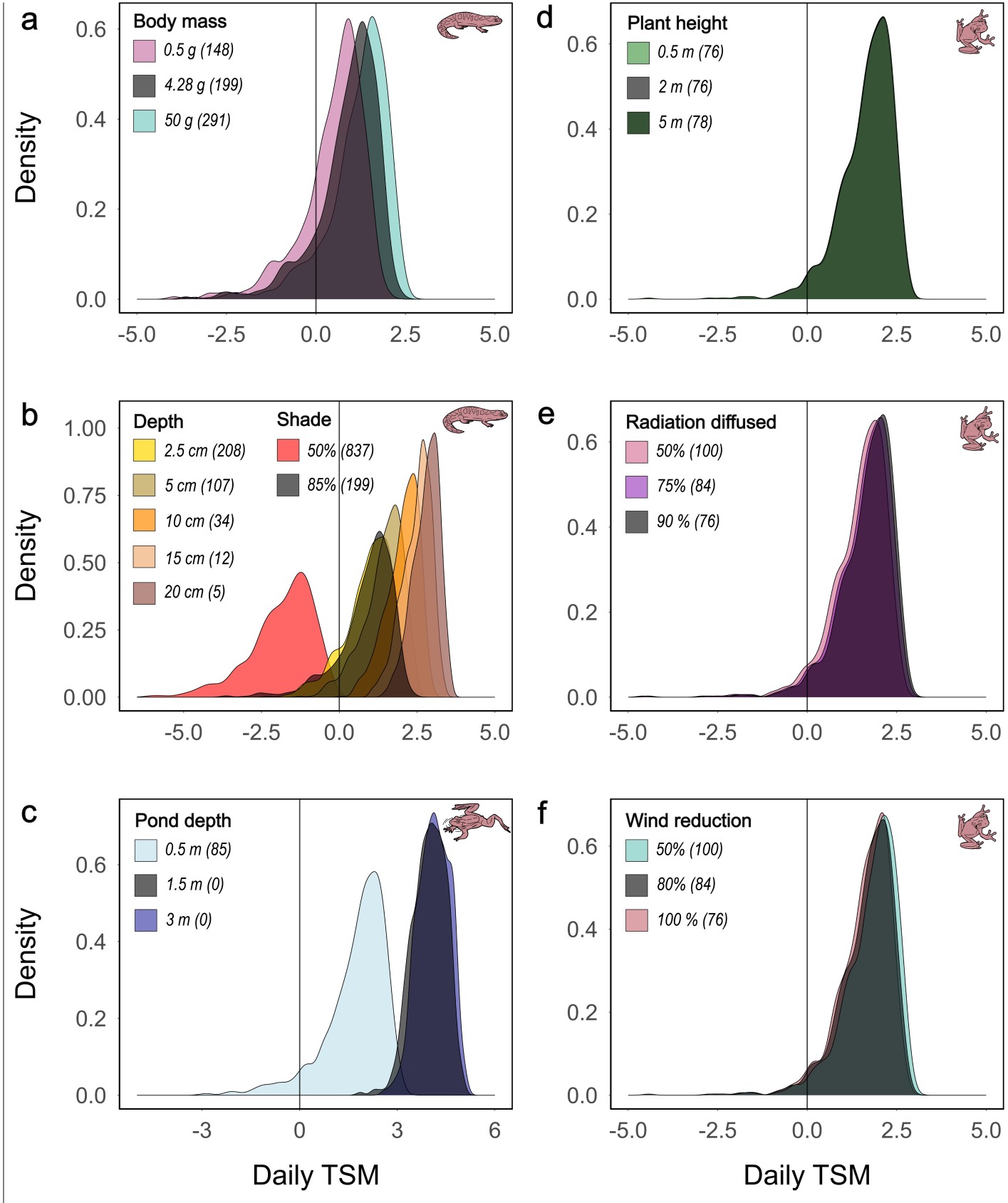

**Extended Data Fig. 9** | See next page for caption.

**Extended Data Fig. 9 | Influence of biophysical model parameters on the estimation of terrestrial (a,b), aquatic (c), and arboreal (d,e,f) thermal safety margins.** Depicted is the variation in daily thermal safety margins (TSM) as density distributions according to body mass (a), shade availability and soil depth (b), pond depth (c), height of the animal in above-ground vegetation (d), percentage of solar radiation diffused by vegetation (e), and percentage of wind reduced by vegetation (f). All simulations were performed assuming 4 °C of global warming above pre-industrial levels in a specific grid cell (latitude, longitude = −9.5, −69.5; where the highest number of overheating events was predicted), for the most vulnerable species (*Noblella myrmecoides* in terrestrial and aquatic microhabitats, *Pristimantis ockendeni* in arboreal microhabitats). Negative daily TSMs were recorded as overheating events, and conditions depicted in dark grey reflect the results presented in the manuscript. The number of predicted overheating events is indicated in brackets for each condition (n = 910 days).

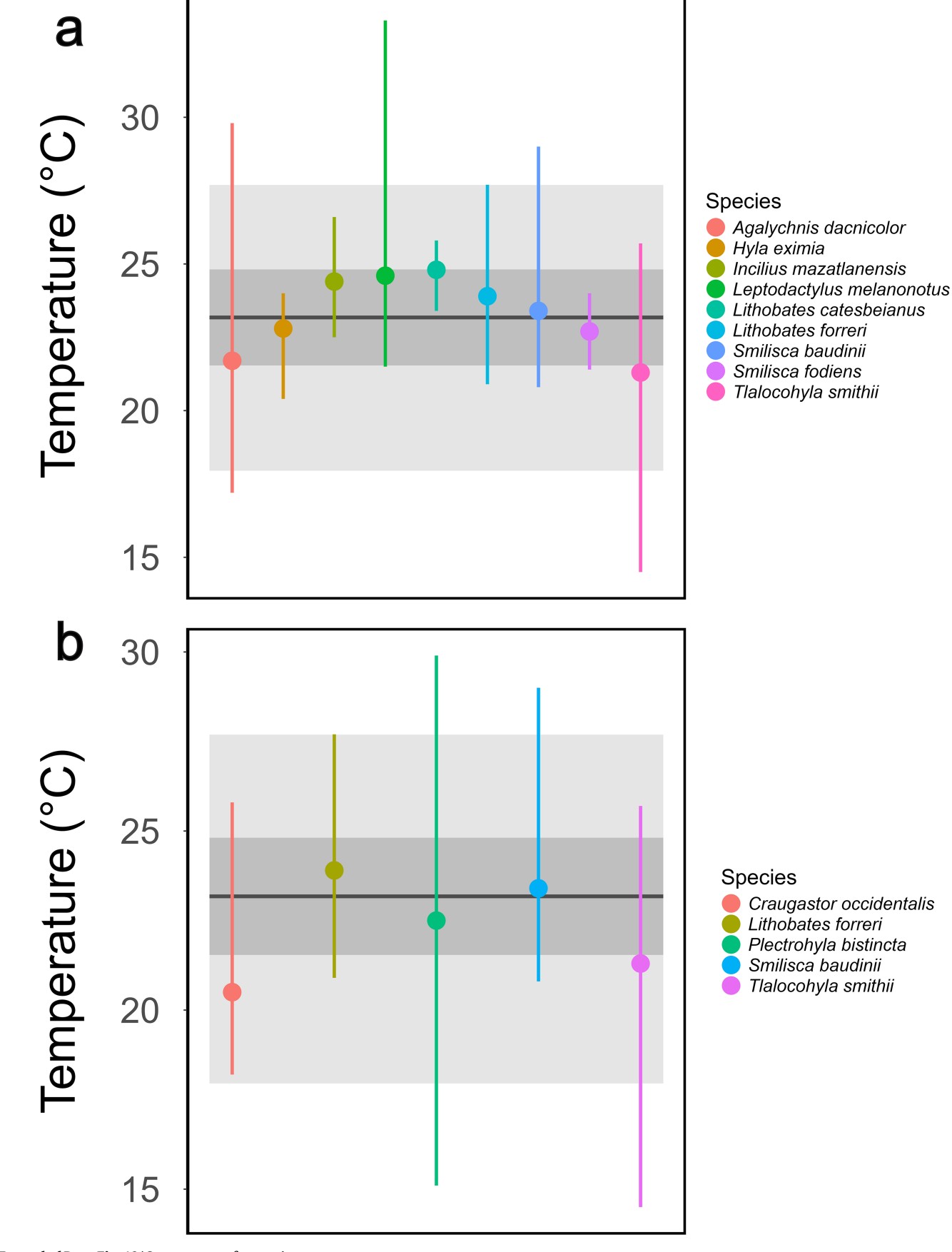

**Extended Data Fig. 10** | See next page for caption.

**Extended Data Fig. 10 | Validation of operative body temperature estimations.** Terrestrial operative body temperatures estimated from biophysical models were compared to field body temperatures recorded around Tepic (21.48° N, −104.85° W; n = 11 species; panel a) and El Cuarenteño (21.45° N, −105.03° W; n = 5 species; panel b) between June and October of 2013/2015, for 11 species of frogs[5]. The mean hourly operative body temperatures predicted from our models for the same date and time windows (18:00 – 01:00) are represented by the black horizontal line, along with their standard deviation (dark grey box), and range (light grey box). The mean (point) and range (bars) of field body temperatures recorded for each species are presented in colour. Note that our analyses were based on the maximum daily temperature recorded at each site during the warmest quarters of 2006–2015, which may not match the times and dates at which field body temperatures were recorded. Nevertheless, congruence between night-time predicted and field body temperatures suggests our models are likely to capture true biological variation in operative body temperatures throughout the day.

# Reporting Summary

## Statistics

For all statistical analyses, confirm that the following items are present in the figure legend, table legend, main text, or Methods section.

| n/a | Confirmed | |
|---|---|---|
| ☐ | ☒ | The exact sample size (*n*) for each experimental group/condition, given as a discrete number and unit of measurement |
| ☒ | ☐ | A statement on whether measurements were taken from distinct samples or whether the same sample was measured repeatedly |
| ☐ | ☒ | The statistical test(s) used AND whether they are one- or two-sided<br>*Only common tests should be described solely by name; describe more complex techniques in the Methods section.* |
| ☐ | ☒ | A description of all covariates tested |
| ☐ | ☒ | A description of any assumptions or corrections, such as tests of normality and adjustment for multiple comparisons |
| ☐ | ☒ | A full description of the statistical parameters including central tendency (e.g. means) or other basic estimates (e.g. regression coefficient) AND variation (e.g. standard deviation) or associated estimates of uncertainty (e.g. confidence intervals) |
| ☐ | ☒ | For null hypothesis testing, the test statistic (e.g. *F*, *t*, *r*) with confidence intervals, effect sizes, degrees of freedom and *P* value noted<br>*Give P values as exact values whenever suitable.* |
| ☐ | ☒ | For Bayesian analysis, information on the choice of priors and Markov chain Monte Carlo settings |
| ☐ | ☒ | For hierarchical and complex designs, identification of the appropriate level for tests and full reporting of outcomes |
| ☒ | ☐ | Estimates of effect sizes (e.g. Cohen's *d*, Pearson's *r*), indicating how they were calculated |

*Our web collection on statistics for biologists contains articles on many of the points above.*

## Software and code

Policy information about availability of computer code

| Data collection | Software and code was used to download data from the National Centre of Environmental Predictions (NCEP), using the curl package (version 5.0.0). Other data were not collected using software or code. |
|---|---|

| Data analysis | All code is available at https://github.com/p-pottier/Vulnerability_amphibians_global_warming (accessible with the webpage: https://p-pottier.github.io/Vulnerability_amphibians_global_warming/).<br><br>All data analyses were performed using R statistical software (version 4.3.0). The following packages were used for processing and/or analysing the data: RNCEP (version 1.0.10), terra (version 1.7-46), emmeans (version 1.7.3), optimx (version 2023-10.21), ggeffects (version 1.2.2), cowplot (version 1.1.1), lwgeom (version 0.2-13), ggspatial (version 1.1.8), metafor (version 4.2-0), numDeriv (version 2016.8-1.1), metadat (version 1.2-0), rnaturalearthhires (version 0.2.1), rnaturalearthdata (version 0.1.0), rnaturalearth (version 0.3.3), futile.logger (version 1.4.3), future.apply (version 1.10.0), furrr (version 0.3.1), future (version 1.33.0), rlang (version 1.1.1), gamm4 (version 0.2-6), lme4 (version 1.1-33), mgcv (version 1.8-40), nlme (version 3.1-157), MCMCglmm (version 2.34), coda (version 0.19-4), Matrix (version 1.5-4), microclima (version 0.1.0), NicheMapR (version 3.3.2), RNetCDF (version 2.6-2), data.table (version 1.14.8), sf (version 1.0-14), zoo (version 1.8-12), curl (version 5.0.0), abind (version 1.4-5), doParallel (version 1.0.17), iterators (version 1.0.14), foreach (version 1.5.2), rgdal (version 1.6-7), taxize (version 0.9.100), rredlist (version 0.7.1), letsR (version 4.0), rgeos (version 0.6-2), rasterSp (version 0.0.1), raster (version 3.6-23), sp (version 2.0-0), ggbeeswarm (version 0.7.2), ggExtra (version 0.10.0), here (version 1.0.1), ggstatsplot (version 0.11.1), ggdist (version 3.2.1), RColorBrewer (version 1.1-3), ggnewscale (version 0.4.10.9000), tidytree (version 0.4.2), phytools (version 1.5-1), ggtreeExtra (version 1.7.0), ggtree (version 3.5.0.901), R.utils (version 2.12.2), R.oo (version 1.25.0), R.methodsS3 (version 1.8.2), patchwork (version 1.2.0.9000), naniar (version 1.0.0), ape (version 5.7-1), maps (version 3.4.1), viridis (version 0.6.4), viridisLite (version 0.4.2), kableExtra (version 1.3.4), lubridate (version 1.9.2), forcats (version 1.0.0), stringr (version 1.5.0), dplyr (version 1.1.2), purrr (version 1.0.1), readr (version 2.1.4), tidyr (version 1.3.0), tibble (version 3.2.1), ggplot2 (version 3.5.1), and tidyverse (version 2.0.0). |
|---|---|

For manuscripts utilizing custom algorithms or software that are central to the research but not yet described in published literature, software must be made available to editors and reviewers. We strongly encourage code deposition in a community repository (e.g. GitHub). See the Nature Portfolio guidelines for submitting code & software for further information.

# Data

Policy information about availability of data

All manuscripts must include a data availability statement. This statement should provide the following information, where applicable:

- Accession codes, unique identifiers, or web links for publicly available datasets
- A description of any restrictions on data availability
- For clinical datasets or third party data, please ensure that the statement adheres to our policy

All heat tolerance data were compiled in a previous study (Pottier et al. 2022. Scientific Data), climatic data were taken from the National Center for Environmental Predictions (NCEP) and TerraClimate, species distribution ranges were taken from the International Union for the Conservation of Nature (IUCN) red list, ecotype and body mass data were taken from Wu et al. (2024, EcoEvoRxiv), Johnson et al. (2023, Global Ecology and Biogeography), and Santini et al. (2018, Integrative Zoology), and phylogenetic data were taken from Jetz & Pyron (2018, Nature Ecology & Evolution). All data sources are acknowledged and referenced in the manuscript.

Raw and processed data are available at https://github.com/p-pottier/Vulnerability_amphibians_global_warming, and are archived in Zenodo (https://doi.org/10.5281/zenodo.14498866). Note, however, that some intermediate data files were too large to be shared online. These files are available upon request. TerraClimate data is available from https://www.climatologylab.org/terraclimate.html and NCEP data is available from https://psl.noaa.gov/thredds/catalog/Datasets/ncep.reanalysis2/gaussian_grid/catalog.html.

# Research involving human participants, their data, or biological material

Policy information about studies with human participants or human data. See also policy information about sex, gender (identity/presentation), and sexual orientation and race, ethnicity and racism.

| Reporting on sex and gender | NA |
|---|---|
| Reporting on race, ethnicity, or other socially relevant groupings | NA |
| Population characteristics | NA |
| Recruitment | NA |
| Ethics oversight | NA |

Note that full information on the approval of the study protocol must also be provided in the manuscript.

# Field-specific reporting

Please select the one below that is the best fit for your research. If you are not sure, read the appropriate sections before making your selection.

☐ Life sciences ☐ Behavioural & social sciences ☒ Ecological, evolutionary & environmental sciences

For a reference copy of the document with all sections, see nature.com/documents/nr-reporting-summary-flat.pdf

# Ecological, evolutionary & environmental sciences study design

All studies must disclose on these points even when the disclosure is negative.

| | |
|---|---|
| Study description | In this study, we assessed the global vulnerability of amphibians to extreme heat events in different climatic scenarios and thermal refugia. We developed a new approach to solve taxonomical and geographical biases in thermal limits using Bayesian phylogenic data imputation. We then integrated predicted thermal limits with body temperatures estimated from biophysical models to quantify the proximity of heat tolerance limits to field body temperatures experienced in shaded microhabitats. |
| Research sample | All heat tolerance data were compiled in a previous study (Pottier et al. 2022. Scientific Data), climatic data were taken from the National Center for Environmental Predictions (NCEP) and TerraClimate, species distribution ranges were taken from the International Union for the Conservation of Nature (IUCN) red list, ecotype and body mass data were taken from Wu et al. (2024, in prep), Johnson et al. (2023, Global Ecology and Biogeography), and Santini et al. (2018, Integrative Zoology), and phylogenetic data were taken from Jetz & Pyron (2018, Nature Ecology & Evolution). All data sources are acknowledged and referenced in the manuscript. Heat tolerance data were filtered to 2,661 estimates from 524 species using predefined inclusion criteria, and our data imputation procedure has expanded this sample to data from 5203 species (spanning up to 204,808 populations for each microhabitat and climatic scenario). Detailed sample sizes are provided in the Results section. |
| Sampling strategy | Heat tolerance data were only included if they were measured using a dynamic methodology, if the acclimation temperature was recorded, if the species was listed in the phylogeny from Jetz & Pyron (2018, Nature Ecology & Evolution), and if their geographical range was reported in the IUCN red list. |
| Data collection | PPottier, NCW, PPollo and ANRV collected all data. Heat tolerance data were compiled by PPottier, PPollo and ANRV, climatic data, distribution ranges, and phylogenetic data were compiled by PPottier, and ecotype and body mass data were compiled by PPottier, NCW, PPollo, and ANRV. |
| Timing and spatial scale | Heat tolerance data was collected up to May 2021, and cover a large spatial scale. Geographic biases are discussed in the manuscript and resolved using data imputation. |
| Data exclusions | Data not matching our inclusion criteria (i.e., heat tolerance data were only included if they were measured using a dynamic methodology, if the acclimation temperature was recorded, if the species was listed in the phylogeny from Jetz & Pyron (2018, Nature Ecology & Evolution), and if their geographical range was reported in the IUCN red list) were excluded. |
| Reproducibility | All analyses are reproducible using the code provided. We also provided a rendered html file to walk the reader through the analyses and facilitate reproducibility. |
| Randomization | No randomization was involved in this study. |
| Blinding | No blinding was involved in this study. |

Did the study involve field work? ☐ Yes ☒ No

# Reporting for specific materials, systems and methods

We require information from authors about some types of materials, experimental systems and methods used in many studies. Here, indicate whether each material, system or method listed is relevant to your study. If you are not sure if a list item applies to your research, read the appropriate section before selecting a response.

## Materials & experimental systems

| n/a | Involved in the study |
|---|---|
| ☒ | ☐ Antibodies |
| ☒ | ☐ Eukaryotic cell lines |
| ☒ | ☐ Palaeontology and archaeology |
| ☒ | ☐ Animals and other organisms |
| ☒ | ☐ Clinical data |
| ☒ | ☐ Dual use research of concern |
| ☒ | ☐ Plants |

## Methods

| n/a | Involved in the study |
|---|---|
| ☒ | ☐ ChIP-seq |
| ☒ | ☐ Flow cytometry |
| ☒ | ☐ MRI-based neuroimaging |

# Plants

Seed stocks

*Report on the source of all seed stocks or other plant material used. If applicable, state the seed stock centre and catalogue number. If plant specimens were collected from the field, describe the collection location, date and sampling procedures.*

Novel plant genotypes

*Describe the methods by which all novel plant genotypes were produced. This includes those generated by transgenic approaches, gene editing, chemical/radiation-based mutagenesis and hybridization. For transgenic lines, describe the transformation method, the number of independent lines analyzed and the generation upon which experiments were performed. For gene-edited lines, describe the editor used, the endogenous sequence targeted for editing, the targeting guide RNA sequence (if applicable) and how the editor was applied.*

Authentication

*Describe any authentication procedures for each seed stock used or novel genotype generated. Describe any experiments used to assess the effect of a mutation and, where applicable, how potential secondary effects (e.g. second site T-DNA insertions, mosiacism, off-target gene editing) were examined.*

