## [Peer Review File · Nature]

Vulnerability of amphibians to global warming

Corresponding Author: Dr Patrice Pottier

Version 0:

Reviewer comments:

Referee #1

(Remarks to the Author)

The authors of this study combined upper thermal tolerance data for amphibians with data on geographical ranges and gridded microclimate data to predict vulnerability to warming across all amphibians with well-described ranges. They impute thermal tolerance data to leverage CTmax estimates for 524 species to produce imputed values for 5203 species. They model body temperatures in 3 microhabitats across warm seasons experienced from 2006-2015, allow for acclimation in CTmax, and assess the seasonal thermal safety margin, number of occurrences in which operative temperatures reached CTmax, and the number of days spent above CTmax. They find new and useful patterns that can guide conservation action, including what are likely improvements on estimates of climate vulnerability using CTmax, and show that +4°C will lead to much more extreme responses compared to +2°C for amphibian resilience.

I found this paper to be highly significant and well executed, bringing together what I recognize to be our most cutting-edge approaches (many developed by the authorship) to model body temperatures or organisms in nature as well as dynamic upper thermal tolerance limits, given the data available. It was also well written. Some larger and then smaller comments follow.

Given the broad readership of this journal and the possible uptake of results, I think it is important to validate some aspects of the model, and I did not see that coming through in the methods and results. The study relies heavily on over 5000 imputed estimates of CTmax, based only 524 experimental observations (and those experiments also varied in methods, which was simultaneously modeled to the authors' credit, but this also likely ate up degrees of freedom). Given the importance of this method, I was surprised not to see more validation of the imputed estimates. Line 364 says that there was "strong congruence" between known and imputed data, but the evidence appears to be overlapping density-distributions of values - I would expect to see one plotted against the other to at least assess the accuracy level of the imputation. Also, for non-experts on data imputation, some assurance from the literature that the imputation based on 10% observations does not overstretch the modelling approach would be useful.

Beyond the CTmax estimates, I think some other levels of validation would be useful. Is there any evidence that the species highlighted as having high predicted vulnerability (e.g. in Fig. 2) have responded to warming over the time-period of study? Are estimates of operative body temperatures anywhere close to field body temperatures on hot days?

My overall sense of the imputed CTmax values is that more should be done to communicate their potential for error. The authors state that standard error of imputed CTmax was calculated and propagated throughout the analysis, but this is not clearly evident in the figures. Wouldn't this mean that there should be a range of all three metrics (TSM, number of overheating events, and number of days overheating) for every species x cell in subsequent analyses and figures? In principle the error propagation is a reassuring aspect to the imputation approach, so illustrating the error would be particularly useful towards communicating that these are estimates with error. Better communicating that these are model estimates is probably needed throughout the MS for fair interpretation.

I appreciated all the efforts to apply an acclimation 'correction' to the CTmax estimates. However, it was a bit difficult to follow the methods around how acclimation response was predicted. Based on Fig. S2, there appears to be an acclimation response slope fit to CTmax change with acclimation temperature and duration. Was this estimated from the 524 experiments? Was there one global slope fit or was the acclimation response allowed to vary among species or latitude? Based on previous quantitative reviews of acclimation responses of CTmax in ectotherms, we expect the acclimation response to vary systematically with latitude as well as across species that use different habitats (lots of papers on this). If

the authors did not allow variation in the slope, how could a systematic dampening of the acclimation response slope at lower latitudes affect their results? Given the importance of the plasticity response I think this should be made clearer so the limitations could be discussed.

In relation to this, if I understand it correctly, Extended Data Fig. 3 d-f shows the acclimated response of CT_{max} under the different future scenarios, which appears very small (almost overlapping points). This small acclimation response is highlighted in the discussion but is not really brought out in the results, so I think this should be better referenced. As an additional point, I think the authors should also mention, at least in the discussion, the possibility of (and evidence for) local adaptation in CT_{max} across a species' range and how that might impact their results.

The terms "population" and "community" were used conventionally in this paper, and I do not think it is justified to redefine these biological terms for their study. The authors used "population" to refer to predicted organismal responses of a single species within a grid cell, which itself was defined by the climate data. There was no "population-level response" as might be applied to demographic responses that require a population, but an organismal response applied to a unit of space. Similarly, what authors call "communities" are at best assemblages of species predicted to exist in the same grid cell according to their IUCN polygons. No community processes are applied and there is no evidence these species interact or even actually exist in each grid cell. This is perhaps just a semantics issue, but these terms have meanings I do not think they are justified in this context.

I was somewhat distracted in the methods section by the unconventional author attribution style. Is this a new standard? I have never seen attribution to each author in the methods, and I found it difficult to get used to. It also takes up a lot of word space.

At some points in the paper I felt that the use of 4°C was an unrealistic value, but at other times I felt it was justified, and even wondered about the influence of heat waves. If the authors could consider the impact of heat waves or climate variability over and above the influence of climatology, I think it could help justify the importance of 4°C simulation.

These issues are all surmountable and I think this will be a significant contribution to the field.

Smaller points:

Lines 119-120: How many species were removed based on the need for IUCN ranges? (would be interesting to know how many were removed with each of these criteria)

Lines 165-169: After reading the paper, I deduced that imputation must lead to a new column of imputed data for all species, so what is shown hereafter are the imputed data and not the raw "known" data. If so, this could be explained very quickly here. Also how many species had more than one CT_{max} estimate in the model?

Lines 221-228: It might be better for the authors to rearrange the order here and say that the more obvious approach (estimating a body-mass specific Tb for each grid cell, species, and microhabitat) was computationally intensive to instead a median TB was used, etc. It would also be good to explore how much this assumption might have impacted the results.

Lines 266-268: The explanation of the methods here is not sufficiently clear. What was the model? How many estimates were there per species? What is meant by precision of imputed measurements – SD around a mean? Reference to "meta-analytic" does not get around the need to describe the model. It might be better to explain this model in plainer language.

Line 293: allowing "one" to identify? Same at 295

Line 333: By "fitted without an intercept" does this specify a certain contrast structure? I found myself confused by this.

Line 354: The description of "coverage" is confusing here. What is meant by "most of the distribution of amphibians"? 60% of all described amphibian species?

Line 355: "populations" these are not really populations

Line 358: these are not really communities (see notes above). Potential assemblages?

Line 361: not sure I agree to call this "knowledge" – can this be nuanced a bit more?

Line 364-365: This does not seem like enough evidence of congruence

Fig. 2: Colour choice of TSM and overheating events could be more similar (rather than opposite), so one might more easily look for connections between TSM and # overheating events.

Fig. 2: Could the authors not add information here about which species have data?

Line 398: "populations predicted to overheat had TSMs well above zero"- where is support for this statement? Is this the match up between some pink lines and yellow bars in Fig. 2? Is there a better way to show this (e.g. binomial plot with event as a function of TSM in the supplement?)

Line 404: "patterns" What pattern? Spatial pattern?

Lines 425-427: Is this because CTmax is lower in arboreal species?

Line 445: I would avoid "significant" here but I would describe this result just a little more, because it is surprising that so few species in some cases makes up such a high proportion. Looks like the proportion of diversity vulnerable is higher in the deserts because richness is lower.

Line 497: Are there actually 1500 points in Fig. 5? Otherwise I don't really see these numbers there.

Line 525: Agree on this point, but it would be nice to point out that some studies estimated TSM with shorter-time frame data (a TSM based on the hottest hour of the hottest day must be more similar to probability of overheating).

Line 542: "thermoregulation" I think this is less about thermoregulation and more about opportunities for cooling.

Line 566: Considering effects of non-lethal warming might not only lead to more extreme vulnerability, but could also sometimes lead to less extreme estimates, as some species might have greater fitness before overheating

Line 574: Were high-risk species more likely those that were imputed? If so, this might mean that some local adaptation could exist but could not be included

Line 586: I feel that this sentence should end with something like "due to overheating according to our model"

Line 591: The range-shift field tends to avoid the term "colonization" as it can be a trigger-word for indigenous people. Suggest change to "establishment"

Line 593: "likely to be rare" can you back this up? There are plenty of amphibians with range shifts in Lenoir et al. 2020

Line 595-596: This is true, but species with large ranges might have a quite different situation at their cold range edges, where warming might improve their foraging success.

Line 608: what ecosystems do amphibians support? Reference needed

Line 609: "planet undergoing metamorphosis" seems like an unnecessary biological metaphor – why metamorphosis and not change? If the authors feel strongly about the metaphor, I suggest they develop it or cite a reference.

Extended Data Fig. 1. Very nice didactic figure! This reader thanks you.

Extended Data Fig. 2. Not enough information given about density distribution in (a). How many data points are in this figure? From the reference to this figure (Line 365) it seems to be from the 375 "validated" species but where are the 77 cross-validated species from Line 345? Why not plot one value against the other so we can visualize the relationship we are looking for?

Line 1013: Open code and Markdown explainers - this is a very useful resource! Be good to make sure this is archived for publication.

Referee #2

(Remarks to the Author)

Increasing attention is being focused on understanding how climate change affects species through more mechanistic means than simply correlating their current distribution with climate and extrapolating it. The authors of this manuscript seek to add to this body of work by predicting physiological limits for a large number of amphibian species based on those with that information and then predicting responses to climate change.

The paper follows a handful of papers that seek to use physiology to predict climate change responses for a smaller number of species and modeling approaches that seek to expand trait-based approaches by filling in missing species trait data. I find that the combined approach can be a powerful tool to move beyond simple non-mechanistic models, while countering the problem of limited data. The directive to maintain microhabitat features is an important outcome.

In general, I find the topic interesting, and the manuscript is well-written. I am not a physiological ecologist, so I hope someone with expertise in this area is available to critique the methods with regard to this aspect. I have some major and minor concerns that, if addressed, should improve the manuscript.

Major

It's unclear what is the 'new approach' (L90) since phylogenetic imputation of traits has been around in various forms. I don't think it needs to be new, but if it is claimed as such, it should be made clear what is the advance.

The paper makes a broad claim about the conservatism of the analysis by assuming that microenvironments are available.

One of the important outcomes is to protect these microenvironments. Yet, a comparison of results with and without microenvironments is not done. I think it would strengthen these recommendations and quantify the potential difference that would result if this analysis was performed.

It's not clear how some of the ecotypes were treated in the analysis. For instance, most salamanders are fossorial or live under logs or substrate. Clearly, they will be experiencing very different temperatures, yet it's not clear this is accounted for and thus could be overestimating risks at least for this subset of species.

Several models were not run as expected because they ran too slowly or would not converge. Generally, a bit of hard coding and optimization in a language like STAN would solve these problems. The authors might have some colleagues with some experience to get some of these models running as they wanted. It's not absolutely critical but should be easy enough.

I think the sensitivity analyses should be renamed cross-validation or validation for short. My biggest concern here is that the species that are being imputed are not a random subset – the whole point is that the data is biased, and therefore the types of species being imputed also likely are different. The chance exists that the imputation method is making them like the ones with data (northern hemisphere). I would like to see a validation test that focuses on the types of species with the least information to see how well the model performs.

Along these lines, it is claimed that because the experimental and imputed means were similar that it 'confirms' the accuracy of the procedures. This is not a strong test of the procedure because the imputed species likely differ from the experimental species. Therefore, I would even predict that if the method was working that these means should differ. A better approach would be to evaluate the RSME of observed versus imputed for the validation test, focusing on the types of species that needed to be imputed.

All throughout, the text indicates the method's success and effectiveness. I like the approach and I agree that it could be useful, but we just don't know if it worked until we collect that data that does not exist and test the predictions. Until then, it's just untested predictions that need to be evaluated. Please tone down this language.

Line comments

L 47, I'm not sure if this assumption of access to microenvironments makes the predictions conservative – would not some form of shade exist wherever amphibians live?

L 65, just extreme heat, no need for the word, excursions

L 77, I wouldn't call 616 species as a 'few'

L 91, phylogenetic not phylogenic

L102, I prefer the standard author contribution statement at the end. I find this method of calling out individuals detracts from the paper and the science in favor of emphasizing assignment of individual credit.

L114, provide reasoning for why this methodology was the only one used.

L 154, quantify what reasonable error is in the main text

L 160, would you not weight them by their certainty or inverse sampling variance rather than variance?

L 191-5, Are there sensitivity analyses to understand how these assumptions might change results?

L 251, Is there literature that backs up this period of 7 days for acclimation for amphibians? Are there not limits to acclimation at higher temperatures?

L 293-4, I would use assemblage instead of community at this scale.

L 304, I'm not sure what this means that brms exceeded your computational capacities. It would be a better approach and a cluster can solve computation length.

L 311, should be inverse variance weighting

L 322, similarly, coding these models in STAN or another program would be easy enough and could be tuned to convergence.

L 349, this should be in the supplement if of value.

L 362, here and elsewhere, it is claimed that taxonomic and geographic biases have been "solved." The only way we would know that is if we knew the true values and we do not, so all this can do is hope and maybe encourage the collection of these data in the future to test if it is a solution.

L 387-97, More should be said about the result that CTmax basically did not vary with latitude (EXT fig 3 d, e, f). This is why TSM varies with latitude because CTmax is a constant. Has this been found before? Is there a chance this is an artifact?

L 478-9, isn't this just the result of a variance and the distance from a mean to a threshold? So, it could be predicted based on pure mathematics?

L 496-7, maybe here make the point that the physiological limits chosen result in death over some short timeline and not just a chronic limit.

L 508, 'some' populations – not all.

L 550 – warming impacts may exceed projections – I would look at it both ways in this section – how your estimates are possibly high because species go underground. What about local adaptations across ranges of populations? And the possibility for adaptive evolution?

L 558, amphibians 'in these degraded systems' – not all of them

L 570, how do you know it's effective without validation with true estimates?

L: 577, are local experts citizens? Regardless, do you think citizens can do CTmax experiments? I think so, but lots to think through including animal welfare, providing the equipment, etc.

L 592-3, range shifts will only be rare for some species. Some amphibians disperse well (large ranids, toads) and many do not require water bodies (many salamanders). It's these broad generalizations that reduce the credibility of these statements. The question should be turned around. Some will not expand their range quickly, therefore we need to know which ones, and how physiology will play a role.

L 605, it seems like somewhere, likely here, you will want to mention climate change refugia, defined as habitats with low variability and resistance to change, which would certainly be the types of habitats to be restored, created, and protected.

Version 1:

Reviewer comments:

Referee #1

(Remarks to the Author)

I find that the changes made by the authors to be very satisfactory enjoyed the discourse in the responses. I see only two remaining issues.

I am still not sure that uncertainty in estimates based the imputed CTmax is being communicated sufficiently. The estimation of uncertainty is mentioned as a positive feature of the imputation method, but no figure in the main text illustrates the uncertainty of imputed CTmax or resulting TSMs. In addition, the authors stated in the response to reviewers (REPLY 4) that they cannot propagate error into estimates of number of overheating events or probability of overheating. However, this seems easily surmountable by iterating these calculations across multiple draws of each value from a normal distribution with SD set to the SE of each CTmax estimate. From each draw, the metrics can be calculated, and a distribution (e.g. mean and SD) across iterations can be calculated or visualized. I do not wish to prescribe a method here, but just to say I do not see that this is insurmountable, and this seems important given the emphasis in Nature journals to display uncertainty.

Second, I appreciate the authors' perspective that validation of their predictions writ large would constitute a whole other study, and I am mostly satisfied that the authors have instead added more nuance to their imputed CTmax values and climate vulnerability estimates as hypothetical. However, the authors have added a figure that shows the range of hourly operative body temperature from their model for several frog species at two sites compared to field body temperatures measured (Fig. S12). Although the authors presented this figure as evidence for validation of their estimated hourly operative body temperatures, I did not find this figure to be supportive, for the reason that the range of modelled operative temperatures was so broad (grey box covering ~14-28°C) with no information about central tendency or the source of variation within that range (e.g. I presume there were diurnal patterns, plus possibly some seasonal variation between the months considered). This range of possible body temperatures is so broad it could arguably encompass the field body temperatures of any frog anywhere, and therefore I do not find this to be a useful validation. Was there any information about time of day of the field body temperature estimates? Is there any information about the spatial scale of the operative body temperature estimate vs. the field measures that can explain and possibly connect the high variation in both? I feel some refinement is needed to increase the precision of the prediction for this figure to support the claim.

As an aside, given the broad range of estimated operative temperatures shown, which I assume is due to some combination of seasonal and daily variation, this figure also made me question which operative body temperature estimate was used for the estimation of climate vulnerability. I found in the methods again that it was the maximum daily, which I believe is sensible, but if this figure is refined and used, I would recommend that some reference to the daily maxima is also made, to keep this clear with readers.

These issues are surmountable and I commend the authors on a very thorough study.

Referee #2

(Remarks to the Author)

when I reviewed this manuscript, I thought combining data imputation and mechanistic physiological models was a valuable way to make progress toward predicting a large number of species while applying biological principles. However, I thought some statements were oversold and had some questions about methods. The authors have largely addressed all of my concerns. A study of this size and complexity will have many gaps, and the authors cannot fill them all. Given the computational resources on hand and the many sensitivity analyses in the supplement, I think they have done enough for publication.

One small note - on L 640, the authors still note that they weight heat tolerance estimates by their variance. They suggested in their response that MCMCglmm accepts a weight of the variance. But my understanding is that the model then applies the inverse to it. Whatever the case, one would never weight an estimate by variance, essentially assigning more weight to the least certain estimates. The authors did make this mistake in another place and corrected it based on my comments, so they should also check here/correct the text.

The paper will provide an important means to extend mechanistic models to more species and generate more accurate predictions of which species and regions are most threatened by climate change.

Referee #3

(Remarks to the Author)

The manuscript by Pottier et al. 2024 is a novel contribution to the global change literature that examines the vulnerability of amphibians to climate warming. I was impressed by the sheer scope of the work presented in that it links a broad range of disciplines together – phylogenetics, biophysical ecology, and macroecology – to tackle a key challenge of understanding the vulnerability of amphibian taxa to climate warming.

I was asked to review specifically the use of climate data in this work. One of the core strengths of this work is the use of microclimatic data to model amphibian vulnerability to warming. The authors rely on the NicheMapR package to downscale gridded global climate data both in time and in spatial resolution while accounting for a range of important biophysical processes such as terrain effects and canopy shading. The NicheMapR package and its applications have been vetted extensively elsewhere along with the relevant biophysics underpinning the downscaling routines. The authors clearly state in the main text the assumptions that are inherent in this downscaling work related to canopy shading. They also note that their approach doesn't account for uncertainty in water availability and changing hydrological conditions. Given the scope and scale of analysis, I think the assumptions employed in the work are reasonable and clearly articulated.

I have one question/concern about the use of climate data as described in the methods. Based on what I can ascertain from the methods, the authors use NCEP reanalysis data for the historic reference period and downscale these data using NicheMapR. The authors then use TerraClimate +2 and +4C projections for estimating future conditions and similarly downscale these to hourly and finer spatial resolutions. It's not clear whether the authors bias correct between the NCEP reference dataset and the TerraClimate futures? The methods state: 'We ran all microclimatic estimations between 2005 and 2015 to match the range of pseudo-years available for TerraClimate future climate projections.' Is this a bias correction step? If yes, a simple addition to the paragraph stating explicitly that bias correction was done and how it was done, would improve the clarity of the methods. If bias correction between these datasets was not done then it would be important to describe why and what implications that may have for the results.

Beyond this, I found the manuscript to be well written, clear, impactful, and an important contribution to multiple fields. Thanks for the opportunity to review.

(Remarks on code availability)

I briefly examined the code provided by the authors but did not review it in detail or entirety given its length and sophistication. What I can say is that the authors appear to make a concerted effort to provide the community with a valuable set of resources. Thoroughly replicating the analysis will prove to be difficult given the size of the dataset some of which are not available directly online without contacting the authors. Additionally, the use of sophisticated climate downscaling routines will further make the effort challenging.

Version 2:

Reviewer comments:

Referee #2

(Remarks to the Author)

I previously reviewed this article and thought it had done a nice job of breaking the "n-biology barrier," meaning that it can make predictions for many species but also uses biologically meaningful information by applying imputation techniques. My remaining questions regarding the methods were answered and corrected in this version.

I also was asked to review responses from reviewer 1, who was unable to re-review the manuscript. These responses can be broken down into x points:

Point 1: Need to incorporate uncertainty from the imputation technique

In response, the authors now include this uncertainty as the standard error of the simulated CTmax distributions. They additionally demonstrate that alternative methods can inflate heat risks and thus are undesirable in supplemental analyses and figures. I think there are many ways to approach this, and this is still a developing field. I might have liked having seen a more robust sampling approach applied, but I understand the computational limitations imposed. The applied approach seems to be a good compromise between what is needed and what is practical.

Point 2: Validation was not conclusive

In response, the authors have refined their analyses somewhat after restricting temperatures to those used during sampling. Although this refinement does demonstrate a closer relationship between observations and predictions, the limitations of the data and few species evaluated do not make as strong of a case, at least in my mind, as is indicated in the main text: this validation "confirmed our imputation approach was accurate and unbiased by demonstrating a strong congruence between experimental and imputed data." I recommend toning down this statement quite a bit. I'm glad that some validation was attempted, but I think it's too limited for over-confidence.

Point 3: Emphasize that daily maximum hourly temperatures are used

This point has been fully addressed.

Overall, except for some toning down of the confidence of statements, I find that the authors made a substantial effort to address these legitimate comments.

(Remarks on code availability)

I don't have the time to review the code again, but did so earlier. Another reviewer also took a look this round.

Referee #3

(Remarks to the Author)

I previously reviewed this manuscript focusing on the use of climate data in the analysis. The authors have clearly and succinctly responded to my questions/comments about bias correction between global climate datasets and have addressed my concerns. I have no further comments to add.

Dear Editor(s),

We are thankful for the opportunity to submit a revised version of our manuscript, “*Vulnerability of amphibians to global warming*”, for consideration in *Nature*.

We have carefully revised our manuscript and addressed the excellent comments made by the referees. We appreciate the relevance of their suggestions and acknowledge how this revision has significantly improved the quality and clarity of our manuscript. We have now also reformatted our manuscript to match *Nature*'s formatting requirements, and we hope you find these changes helpful when reviewing our manuscript.

We provide point-by-point replies to all reviewer comments below.

Yours sincerely,

Patrice Pottier, on behalf of all authors (15th May 2024).

Referees' comments:

Referee #1 (Remarks to the Author):

The authors of this study combined upper thermal tolerance data for amphibians with data on geographical ranges and gridded microclimate data to predict vulnerability to warming across all amphibians with well-described ranges. They impute thermal tolerance data to leverage CTmax estimates for 524 species to produce imputed values for 5203 species. They model body temperatures in 3 microhabitats across warm seasons experienced from 2006-2015, allow for acclimation in CTmax, and assess the seasonal thermal safety margin, number of occurrences in which operative temperatures reached CTmax, and the number of days spent above CTmax. They find new and useful patterns that can guide conservation action, including what are likely improvements on estimates of climate vulnerability using CTmax, and show that +4°C will lead to much more extreme responses compared to +2°C for amphibian resilience.

I found this paper to be highly significant and well executed, bringing together what I recognize to be our most cutting-edge approaches (many developed by the authorship) to model body temperatures or organisms in nature as well as dynamic upper thermal tolerance limits, given the data available. It was also well written. Some larger and then smaller comments follow.

REPLY 1: We are deeply thankful for your positive comments on our manuscript. We are glad you recognise the intricacies and general implications of this work and thank you for your excellent comments and suggestions.

Given the broad readership of this journal and the possible uptake of results, I think it is important to validate some aspects of the model, and I did not see that coming through in the methods and results. The study relies heavily on over 5000 imputed estimates of CTmax, based only 524 experimental observations (and those experiments also varied in methods, which was simultaneously modeled to the authors' credit, but this also likely ate up degrees of freedom). Given the importance of this method, I was surprised not to see more validation

of the imputed estimates. Line 364 says that there was “strong congruence” between known and imputed data, but the evidence appears to be overlapping density-distributions of values - I would expect to see one plotted against the other to at least assess the accuracy level of the imputation. Also, for non-experts on data imputation, some assurance from the literature that the imputation based on 10% observations does not overstretch the modelling approach would be useful.

REPLY 2: Thank you for your suggestions. We agree we should have provided more validations of the imputation approach.

To provide more validation of the imputation model, we have now added an additional panel in Extended Data Figure 2 (Extended Data Figure 2b, see below) that shows how well the imputed-derived CTmax match the empirical-derived CTmax values and communicated more accurately their uncertainty by showing the standard error (SE) of the imputed values for all species (Extended Data Figure 2c, see below). We show that that experimental and imputed CTmax values were highly correlated ($r = 0.86$) and unbiased. We also provided references supporting that large proportions of missing observations are still likely to produce reliable imputation results. Through simulations, the work we cited suggest that the uncertainty in imputed estimates increases with the amount of missing data, though point estimates remain unbiased. As expected, we also see that the uncertainty in our imputed estimates is highest for clades for which we have the least information (i.e., the CTmax of species that are phylogenetically distant from species present in the experimental data tend to have a higher standard error; Extended Data Fig. 2c). The error associated in the imputation is also clustered phylogenetically, likely because some predictors (e.g., ecotype) are structured between clades. This confirms that our imputed estimates reflect both phylogenetic and biological variation in the data. We have now clarified this in the manuscript:

“We confirmed our imputation approach was accurate and unbiased by demonstrating a strong congruence between experimental and imputed data in cross-validations (experimental mean \pm standard deviation = 36.19 ± 2.67 ; imputed mean = 35.93 ± 2.54 ; $n = 375$; $r = 0.86$; Extended Data Fig. 2a,b), though, as expected, the uncertainty in imputed predictions was higher in understudied clades (Extended Data Fig. 2c). (lines 100-104)

“Although the proportion of missing data was large (89.9%), imputations based on large amounts of missing data are common^{13,87}, and although estimate uncertainty increases with the proportion of missing data, as expected, simulation studies have shown estimations remain unbiased^{88,89}” (lines 628-631)

Extended Data Fig. 2 | Accuracy of the data imputation procedure. a) Probability density distributions ($n = 375$ observations, 77 species) of experimental CT_{max} (blue) and CT_{max} cross-validated using our data imputation procedure (pink). b) Correlation between experimental and imputed CT_{max} values. c) Variation in the uncertainty (standard error, SE) of imputed CT_{max} predictions (outer heat map) across studied (blue) and imputed (grey) species.

Beyond the CTmax estimates, I think some other levels of validation would be useful. Is there any evidence that the species highlighted as having high predicted vulnerability (e.g. in Fig. 2) have responded to warming over the time-period of study? Are estimates of operative body temperatures anywhere close to field body temperatures on hot days?

REPLY 3: Thank you for your suggestions.

Unfortunately, it would be difficult to correlate predicted vulnerability based on thermal tolerance with other responses to warming, given the data we have available. For instance, range shifts are often documented as deviations from the centroid of the distribution (e.g., >50% of recorded range shifts in Lenoir et al., 2020. *Nature Ecology & Evolution*), but local extirpations may only occur in some parts of the distribution (most likely the leading edge), and only during occasional heat events. Therefore, high predicted vulnerability based on our models may not necessarily translate into overall range shifts, or translate into major population declines detected in other datasets. Doing such validation would likely be a project in itself, where different variables likely to impact population responses and predicted climate vulnerability are analysed in tandem (see, for example, Johnson et al., 2024. *Nature*, for a complex analysis of biodiversity trends). We think it is an incredibly interesting avenue for future research, but one that may be beyond the scope of the current study. However, if you have a clear idea of how to test such patterns using the data we have available, we would be very happy to consider an additional analysis.

Providing systematic validations of field body temperatures would also be impossible for a global study. From our experience, published estimates of amphibian field body temperatures often lack important information (i.e., exact date and location of sampling, microenvironmental features where the animals were captured) for direct validation our predicted body temperature estimates. However, as a case in point, we compared our predictions to body temperatures estimates from 11 species of frogs in Mexico sampled during our study timeframe (Lara-Resendiz & Luja, 2018. *Revista Mexicana de Biodiversidad*). This study was chosen because it provided the date and location of body temperature measurements and covered multiple species across different sites. We modelled hourly operative body temperatures in shaded terrestrial conditions at two sites between June and October of 2013 and 2015 (sampling dates in this study). We then compared our predictions with mean, minimum, and maximum field body temperatures recorded at these sites, for each species.

The predicted operative body temperatures are within the observed field body temperatures at both sites and for all species (Fig. S12, see below). Interestingly, we note that recorded field body temperatures are sometimes higher than our predictions. This can be explained by differences in microhabitat use, where frogs in more open habitats are likely to reach higher body temperatures than what was predicted in our study (shaded conditions). While this is not a comprehensive validation of our biophysical models, we believe it provides convincing evidence that our approach was accurate for all species at these sites. Moreover, other studies providing more extensive validations of biophysical model predictions also point to similar conclusions (e.g., Enriquez-Urzelai et al., 2019. *Global Change Biology*; Kearney et al., 2018. *Ecology Monographs*).

Fig. S12 | Validation of operative body temperature estimations. Terrestrial operative body temperatures estimated from biophysical models were compared to field body temperatures recorded around Tepic (21.48° N, -104.85° W; panel a) and El Cuarenteño (21.45° N, -105.03° W; panel b) between June and October of 2013/2015, for 11 species of frogs⁵. The range of hourly operative body temperatures predicted from our models are presented by the grey boxes and the mean by the black horizontal line. The mean (point) and range (bars) of field body temperatures recorded for each species are presented in colour.

We have now pointed to this supplementary figure in the manuscript, highlighted that operative body temperatures should reflect core body temperatures in most cases, and that biophysical model predictions have been validated in previous studies.

“We then used microclimate estimates to generate operative body temperatures using the *ectotherm* function in *NicheMapR*⁹⁴. This modelling system has been extensively validated with field observations^{104–106} (see also Fig. S12).” (lines 700-702)

“Given that most amphibians in our dataset are small (median = 1.4 g, mean = 27.5 g), body temperatures equilibrate quickly with the environment, and operative body temperatures are likely representative of core body temperatures.” (lines 710-712).

“We also confirmed that predicted operative body temperatures were comparable to field body temperatures measured in wild frogs (see *Supplementary methods*; Fig. S12).” (lines 859-861)

My overall sense of the imputed CTmax values is that more should be done to communicate their potential for error. The authors state that standard error of imputed CTmax was calculated and propagated throughout the analysis, but this is not clearly evident in the figures. Wouldn't this mean that there should be a range of all three metrics (TSM, number of overheating events, and number of days overheating) for every species x cell in subsequent analyses and figures? In principle the error propagation is a reassuring aspect to the imputation approach, so illustrating the error would be particularly useful towards communicating that these are estimates with error. Better communicating that these are model estimates is probably needed throughout the MS for fair interpretation.

REPLY 4: Thank you for this excellent comment.

We agree that could have better communicated the uncertainty in our imputed estimates. There is indeed a standard error estimate associated with each imputed CTmax metric, and hence the resulting TSM. However, the number of overheating events and the probability of overheating are not associated with a measure of uncertainty because of the binary nature of how they were calculated (the species is predicted to overheat or not based on the distance between CTmax and operative body temperatures). Instead, we provided sensitivity analyses for these metrics, where we considered overheating events to occur only when operative body temperatures exceeded 50 or 95% of the predicted distribution (confidence intervals) of the imputed CTmax. These are extremely conservative analyses because the uncertainty in imputed estimates is large, as expected from imputation procedures. Nevertheless, these analyses point to similar conclusions – the number of overheating events is higher in the tropics in the Southern Hemisphere, and in non-tropical regions in the Northern Hemisphere, with stark differences between warming scenarios (see Fig. S8).

While we tried to incorporate uncertainty in other figures by changing the point size according to the estimate precision, it was difficult to distinguish data points by size with such a large dataset, and it made the figure difficult to visualise. Therefore, we kept most figures unchanged, but we hope that Extended Data Fig. 2c (see REPLY 2) more clearly demonstrate the uncertainty in our predictions. We have now also clarified our approach with the error propagation in the methods. We have also now used the word “predictions” more clearly to emphasise that the data discussed is imputed.

“We confirmed our imputation approach was accurate and unbiased by demonstrating a strong congruence between experimental and imputed data in cross-validations (experimental mean \pm standard deviation = 36.19 ± 2.67 ; imputed mean = 35.93 ± 2.54 ; n =

375; $r = 0.86$; Extended Data Fig. 2a,b), though, as expected, the uncertainty in imputed predictions was higher in understudied clades (Extended Data Fig. 2c).” (lines 100-104)

“Because we used out-of-sample model predictions, we propagated errors from the imputation when estimating the predicted CT_{max} across geographical coordinates. Predicted CT_{max} values and their associated standard errors thus reflect variation in both the imputation procedure and the estimation of plastic responses.” (lines 764-767).

We have also pointed to our sensitivity analyses more clearly in the methods section:

“Thermal safety margin estimates were weighted by the inverse of their sampling variance to account for the uncertainty in the imputation and predictions across geographical coordinates. However, addressing prediction uncertainty for overheating risk and the number of overheating events was complex due to the dichotomous nature of these metrics (i.e., the species overheats or not). As a remedy, we provide conservative analyses where overheating events were counted only when operative body temperatures exceeded 50% or 95% of the predicted distribution of heat tolerance limits (see *Sensitivity analyses*; Fig. S8).” (lines 809-816)

I appreciated all the efforts to apply an acclimation ‘correction’ to the CT_{max} estimates. However, it was a bit difficult to follow the methods around how acclimation response was predicted. Based on Fig. S2, there appears to be an acclimation response slope fit to CT_{max} change with acclimation temperature and duration. Was this estimated from the 524 experiments? Was there one global slope fit or was the acclimation response allowed to vary among species or latitude? Based on previous quantitative reviews of acclimation responses of CT_{max} in ectotherms, we expect the acclimation response to vary systematically with latitude as well as across species that use different habitats (lots of papers on this). If the authors did not allow variation in the slope, how could a systematic dampening of the acclimation response slope at lower latitudes affect their results? Given the importance of the plasticity response I think this should be made clearer so the limitations could be discussed.

REPLY 5: Thank you for this important comment. We agree on the lack of clarity on how CT_{max} was adjusted in response to acclimation.

While Figure S2 showed one acclimation slope fitted across all data points, we allowed slopes to vary among species, and accounted for both phylogenetic and non-phylogenetic contribution to variance estimations. To some extent, such adjustments incorporate variation that is recorded between latitudes, as some species and clades would have varying levels of plasticity because of local adaptation signals captured in our original data. The imputed estimates thus reflect variation in plasticity existing in the training data. However, our inferences across geographic ranges (based on plastic responses inferred from the three imputed estimates for each species) assume that these plastic responses are homogeneous for each species. This is certainly a limitation of our study, which we have now more clearly emphasised. Importantly, this is a common assumption in other macrophysiological studies (e.g., Pinsky et al., 2019. *Nature*). Evidence for within-species variation in plasticity across latitudes is also weak. A large body of research have not found strong evidence for latitudinal variation in plasticity in amphibians (see e.g., Morley et al., 2019; Gunderson & Stillman, 2015; Ruthsatz et al., 2024). We also found that plasticity is generally weak and consistent across species (Fig. S2-3). Therefore, within-species variation in plasticity is likely to have negligible influence on the estimated vulnerability of amphibians.

We have now updated Fig. S2 to incorporate species-specific slopes when available, clarified our methods, and discussed how variation in acclimation slopes are unlikely to have a major impact on our conclusions.

“We also weighted heat tolerance estimates based on their sampling variance, accounted for phylogenetic non-independence using a correlation matrix of phylogenetic relatedness, and fitted random intercepts for species-specific effects and phylogenetic effects, as well as their correlation with acclimation temperatures (i.e., random slopes). In other words, we modelled species-specific slopes (plasticity; see Fig. S2) and partitioned the variance among phylogenetic and non-phylogenetic effects.” (lines 640-645)

“Our approach to accounting for plasticity assumes that plasticity is homogeneous within species and ignores the possible influence of local adaptation. However, given the low variability in plasticity among species (Fig. S2-3), lack of evidence for latitudinal variation in plasticity (^{33,36,114}), high phylogenetic signal in thermal tolerance (Pagel’s $\lambda^{28} = 0.95$ [0.91 – 0.98]; see *Sensitivity Analyses*), and evidence for slow rates of evolution and physiological constraints on $CT_{max}^{62,64}$, geographic variation in thermal tolerance and plasticity is unlikely to have a major influence on our results.” (lines 768-774).

Fig. S2 | Correlations between critical thermal maximum (CT_{max}) and predictors used for the imputation. CT_{max} from the experimental dataset was plotted against acclimation temperature (a), acclimation time (b, log scale), ramping rate (c). Colours are proportional to the values of the continuous predictors and the line refers to predictions from a simple linear regression between CT_{max} and the predictors. Individual slopes for each species are depicted for species when CT_{max} was estimated at different acclimation temperatures (a). Depicted is also the variation in CT_{max} with different endpoints (d), media used to infer body temperature (e), life stages (f), and ecotypes (g). Boxplots depict median (horizontal line), interquartile ranges (boxes), and whiskers extend to 1.5 times the interquartile range. LRR: loss of righting response. OS: onset of spasms.

In relation to this, if I understand it correctly, Extended Data Fig. 3 d-f shows the acclimated response of CT_{max} under the different future scenarios, which appears very small (almost overlapping points). This small acclimation response is highlighted in the discussion but is not really brought out in the results, so I think this should be better referenced. As an additional point, I think the authors should also mention, at least in the discussion, the possibility of (and evidence for) local adaptation in CT_{max} across a species' range and how that might impact their results.

REPLY 6: Thank you for your suggestion.

Indeed, acclimated responses are small (mean = 0.13; range = 0.05 – 0.22; n = 5203; see the new Figure S3 below), which is concordant with previous evidence in amphibians and other terrestrial ectotherms (e.g., Gunderson & Stillman, 2015; Morley et al., 2019).

Fig. S3 | Variation in plastic responses across species. The acclimation response ratio (ARR) represents the magnitude change in heat tolerance limits for each degree change in environmental temperature. We found limited variation in ARR (mean \pm standard deviation = 0.134 ± 0.008 ; range = $0.049 - 0.216$; $n = 5203$).

We also agree that we did not sufficiently discuss the acclimation response, and we have now added a sentence in the results section to highlight this.

“However, warming substantially reduce TSM at all latitudes (Fig. 3), likely reflecting the contrast between weak plastic responses in CT_{max} across latitudes^{11,15} (Extended Data Fig. 3; Fig. S3) and large variation in environmental temperatures (Extended Data Fig. 3).” (lines 161-164)

We have also added some discussion points on local adaptation (see also REPLY 5).

“Our approach to accounting for plasticity assumes that plasticity is homogeneous within species and ignores the possible influence of local adaptation. However, given the low variability in plasticity among species (Fig. S2-3), lack of evidence for latitudinal variation in plasticity (^{33,36,114}), high phylogenetic signal in thermal tolerance (Pagel’s $\lambda^{28} = 0.95$ [0.91 – 0.98]; see *Sensitivity Analyses*), and evidence for slow rates of evolution and physiological constraints on CT_{max} ^{62,64}, geographic variation in thermal tolerance and plasticity is unlikely to have a major influence on our results.” (lines 768-774).

The terms “population” and “community” were used conventionally in this paper, and I do not think it is justified to redefine these biological terms for their study. The authors used “population” to refer to predicted organismal responses of a single species within a grid cell, which itself was defined by the climate data. There was no “population-level response” as might be applied to demographic responses that require a population, but an organismal response applied to a unit of space. Similarly, what authors call “communities” are at best assemblages of species predicted to exist in the same grid cell according to their IUCN polygons. No community processes are applied and there is no evidence these species interact or even actually exist in each grid cell. This is perhaps just a semantics issue, but these terms have meanings I do not think they are justified in this context.

REPLY 7: Thank you for your suggestion. We agree that the terminology we used can be confusing to some readers. We have now rephrased all mentions of “population” to “local species occurrence”, and community to “assemblage” (as also recommended by Reviewer 2, see REPLY 59).

The paragraphs defining these terms now read as:

“In total, we predicted vulnerability metrics for 203,853 local species occurrences (individual species in $1^\circ \times 1^\circ$ grid cells) in terrestrial conditions (5,177 species), 204,808 local species occurrences in water bodies (5,203 species); and 56,210 local species occurrences (1,771 species) in above-ground vegetation, for each warming scenario. The number of species examined in arboreal conditions was lower to reflect morphological adaptations required for climbing in above-ground vegetation. These estimates were then grouped into assemblages (all species occurring in $1^\circ \times 1^\circ$ grid cells), tallying 14,090 and 14,091 assemblages for terrestrial and aquatic species and 6,614 assemblages for arboreal species, respectively.” (lines 124-132)

“The objective of this study was to characterise the vulnerability of amphibians to global warming. We investigated patterns at the level of local species occurrences (presence of a given species in a $1^\circ \times 1^\circ$ grid cell based on IUCN data), allowing one to identify specific

populations and species that may be more susceptible to heat stress and direct targeted research efforts. We also analysed data at the assemblage level, the species composition within a grid cell. In such case, we calculated the weighted mean and standard error of TSM (*sensu*¹¹⁵) across species in each grid cell. Assemblage-level analyses allow one to identify areas containing a higher number of vulnerable species, offering actionable insights for broader-scale conservation initiatives” (lines 788-796).

I was somewhat distracted in the methods section by the unconventional author attribution style. Is this a new standard? I have never seen attribution to each author in the methods, and I found it difficult to get used to. It also takes up a lot of word space.

REPLY 8: We apologise for the confusion with the author attribution. This is a new approach proposed by Nakagawa et al. 2023 (Nature Communications). This provides more transparency in author contributions, but we have now removed mentions of author contributions in the methods. We now listed author contributions in the author contribution section for clarity.

“This study was conceptualized by PPottier, MRK, SB, SMD, and SN. All data manipulation and analyses were performed by PPottier (with conceptual and technical input from SMD and SN for the imputation methods and statistical analyses, MRK, ARG, JER, and NCW for the biophysical modelling and climate vulnerability analyses). All code was reviewed by NCW, ARG, and JER following the recommendations of¹²³. Ecotype information was collected by NCW, PPollo, and ANRV. PPottier, NCW, and SMD contributed to data visualization. PPottier wrote the initial draft, and all authors were involved in the review and editing. PPottier oversaw the project administration, while SMD and SN were in charge of the supervision.” (lines 1003-1010)

At some points in the paper I felt that the use of 4°C was an unrealistic value, but at other times I felt it was justified, and even wondered about the influence of heat waves. If the authors could consider the impact of heat waves or climate variability over and above the influence of climatology, I think it could help justify the importance of 4°C simulation.

REPLY 9: Thank you for your suggestion.

We have used 4°C of global warming because this was the data we had available in the TerraClimate database. The package we used (*NicheMapR*) has dedicated functions to integrate TerraClimate projections and simulate future climatic conditions using 2°C or 4°C of global warming. Please note that these represent scenarios of global warming, meaning that it is not simply adding 4°C over historical climate values - it incorporates projections from different environmental variables (e.g., precipitation, vapour pressure deficit). This means that the level of warming predicted is heterogeneous spatially, as expected from future climate projections. Our projections are also heterogeneous in time, and effectively capture the impact of heat waves by imposing levels of projected warming over historical records that incorporate extreme heat events. However, given the increased likelihood of extreme heat events in warming climates (e.g., Meehl & Tebaldi, 2004. Science), we may have underestimated the frequency and intensity of such events in both warming scenarios.

We would also like to point out that we could not use more recent and comprehensive sources of climate data for our study. We used data from the National Centre for Environmental Predictions (NCEP) for historical data and TerraClimate for future projections. Climatic data at finer resolutions are available from sources such as ERA5, while TerraClimate projections are based on CMIP5 models, as opposed to the most recent CMIP6 projections. While we are aware of these limitations, we were limited by our computational resources and could only use climatic data already embedded and optimized

in *NicheMapR* functions. In fact, estimating body temperatures in a single location already takes between ~5 (current terrestrial conditions), and ~40 minutes (future aquatic conditions), at best. Considering that we have run 104,385 biophysical models (6614-14,091 locations for 3 habitats and 3 warming scenarios), it would have been impossible for us to leverage climatic data at higher resolutions without drastically increasing computational time. We also expect that results should remain relatively robust to the choice of climatic data sources.

4°C of global warming is indeed a high warming scenario (akin to the Shared Socioeconomic Pathway SSP 3-7.0 of the latest IPCC report, which projects warming between 2.5 and 4.7°C by 2100) or the SSP 5-8.5 scenario (projecting between 3.3 and 5.7°C of warming by the end of the century). However, projected levels of warming have increased in most recent IPCC reports, highlighting that the intensity of global warming may be higher than we currently anticipate. Recent historical CO₂ emissions also most closely align with high warming scenarios (RCP 8.5, which projects 4.3°C of predicted warming by 2100; see Schwalm et al., 2020. PNAS). Previous macroecological studies on thermal tolerance and climate vulnerability also routinely use the SSP 5-8.5 (or RCP 8.5) scenario (e.g., Pinsky et al., 2019. Nature; Jorgensen et al., 2023. Nature; Murali et al., 2023. Nature). Therefore, using a high warming scenarios make our study more comparable to previous research. Based on this evidence, we believe that using 2°C or 4°C of global warming are both relevant and interesting scenarios to compare likely projected levels of warming to more extreme projections.

We have clarified our methods to justify the use of these climate projections:

“We also used projected future climate data from TerraClimate²⁹ to generate projections assuming 2°C or 4°C of global warming above pre-industrial levels. These temperatures are within the range projected by the end of the century under low and intermediate/high greenhouse gas emission scenarios, respectively³⁰. Notably, recent historical CO₂ emissions most closely align with high warming scenarios³¹ (i.e., 4.3°C of predicted warming by 2100).” (lines 112-117)

“We used projected future monthly climate data from TerraClimate²⁹ to generate hourly projections assuming 2°C or 4°C of global warming above pre-industrial levels. These temperatures are within the range projected by the end of the century under low (Shared Socioeconomic Pathway SSP 1-2.6 to SSP 2-4.5) and high (SSP 3-7.0 to SSP 5-8.5) greenhouse gas emission scenarios, respectively³⁰” (lines 687-691)

“We did not use a larger range of historical records and only used climate projections available in TerraClimate (i.e., 2°C and 4°C) to reduce computational demands.” (lines 698-699)

These issues are all surmountable and I think this will be a significant contribution to the field.

REPLY 10: Thank you. We hope our response to your general comments and to the following points addressed your suggestions.

Smaller points:

Lines 119-120: How many species were removed based on the need for IUCN ranges? (would be interesting to know how many were removed with each of these criteria)

REPLY 11: Thank you for your suggestion. We added this information in the methods.

“We selected data-deficient species from a species list that matched the phylogeny from ³² (7,238 species), was listed in the IUCN red list⁸¹ along with geographic distribution data (5,792 species), and for which ecotypes were known (6,245 species). We did not consider Caecilians (order Gymnophiona) because, to our knowledge, heat tolerance limits are unknown for all Caecilian species¹². Of the 5,792 species for which we had distribution and phylogenetic data, 5,268 were data-deficient for CT_{max}, of which 4,822 had a known ecotype. After removing Caecilians, we were left with 4,679 species to impute.” (lines 605-612)

Lines 165-169: After reading the paper, I deduced that imputation must lead to a new column of imputed data for all species, so what is shown hereafter are the imputed data and not the raw “known” data. If so, this could be explained very quickly here. Also how many species had more than one CTmax estimate in the model?

REPLY 12: Indeed, we imputed three new CTmax values for each species. We then used these for the analyses because they are more standardised than the experimental data and could be directly compared.

287 species had more than one CTmax estimate in the original data, and we generated three estimates per species, which were used to estimate plasticity and infer CTmax across geographical coordinates.

We added more details to the methods to address your comments:

“In total, we selected 2,661 heat tolerance limits estimates with metadata for 524 amphibian species (mean = 5.08; range = 1 - 146 estimates per species; 287 species with more than one estimate).” (lines 594-596)

“For both known species and data-deficient species, we generated three ecologically relevant and standardised heat tolerance estimates, and all analyses were built upon these standardised imputed estimates.” (lines 656-658)

Lines 221-228: It might be better for the authors to rearrange the order here and say that the more obvious approach (estimating a body-mass specific Tb for each grid cell, species, and microhabitat) was computationally intensive to instead a median TB was used, etc. It would also be good to explore how much this assumption might have impacted the results.

REPLY 13: This is a great suggestion, thank you. We have rephrased this sentence and incorporated this suggestion:

“Estimating body mass-specific operative body temperatures for each grid cell, species, and microhabitat was too computationally extensive, given the geographic and taxonomic scale of our study (464,871 local species occurrences). Therefore, we ran the ectotherm models using the median body mass of the species assemblage in each geographical coordinate.” (lines 704-708)

We have also shown in a new sensitivity analysis that estimates of operative body temperatures are generally robust to changes in body mass parameters (see Fig. S9; see also REPLY 58 for analyses with other parameters), although vulnerability may be slightly higher for heavier animals, and lower for lighter individuals (Fig. S9). We added a few sentences in the manuscript to discuss how modelling different biological and environmental characteristics in biophysical models may impact our conclusions (see also REPLY 58).

“We investigated alternative ways to i) calculate thermal safety margins, ii) account for acclimation responses, and iii) control for prediction uncertainty (see *Supplementary methods*; Fig. S6-8) and investigated the influence of different parameters of our biophysical

models (i.e., shade and burrow availability, plant height, solar radiation, wind speed, pond depth) on predicted vulnerability risks (see *Supplementary methods*; Fig. S9-11). Our results were generally robust to changes in model parameters, although amphibians are likely to experience more overheating events in open habitats^{11,52} (Fig. S9) and shallow ponds (Fig. S10), and lower risks in underground conditions¹²⁰ (Fig. S9).” (lines 852-859)

Lines 266-268: The explanation of the methods here is not sufficiently clear. What was the model? How many estimates were there per species? What is meant by precision of imputed measurements – SD around a mean? Reference to “meta-analytic” does not get around the need to describe the model. It might be better to explain this model in plainer language.

REPLY 14: Thank you for your recommendations. We agree this section of the methods lacked clarity. We have added a more thorough description of the model.

“Using the imputed data, we fitted an individual meta-analytic model for each species to estimate the plasticity of imputed heat tolerance limits (CT_{max}) to changes in operative body temperatures using the *metafor* package¹⁰⁹ (v. 4.2-0). CT_{max} was used as the response variable, acclimation temperature (i.e., median, 5th percentile, or 95th percentile daily maximum body temperature experienced by a species across its distribution range) was used as the predictor variable, and imputed estimates were weighted based on their standard error. From these models, we used out-of-sample model predictions (using the *predict* function) to estimate the CT_{max} of each species in each 1° x 1° grid cell across their distribution range in different warming scenarios, based on predicted mean weekly body temperatures. Specifically, we assumed that species were, on any given day, acclimated to the mean daily body temperature experienced in the 7 days prior¹⁹. Therefore, CT_{max} was simulated as a plastic trait, which varied daily, as animals acclimate to new environmental conditions (Extended Data Fig. 1).” (lines 750-761)

Line 293: allowing “one” to identify? Same at 295

REPLY 15: Thank you, we incorporated your suggestions.

Line 333: By “fitted without an intercept” does this specify a certain contrast structure? I found myself confused by this.

REPLY 16: We apologise for the confusion. Models without an intercept do not have a contrast structure and were used to estimate mean estimates in each category of the predictor fitted (in our case, the mean estimate for each climate scenario / microhabitat). Note that we also fitted models with a contrast structure (assessing statistically significant differences between microhabitat and climate scenarios), but these models almost always indicate statistically significant differences between conditions because of the large amount of data analysed.

We have now rephrased this sentence for clarity: “All models were fitted without a contrast structure to estimate mean effects in each microhabitat and climatic scenario, and with two-sided contrasts to draw comparisons with current terrestrial conditions.” (lines 838-840)

Line 354: The description of “coverage” is confusing here. What is meant by “most of the distribution of amphibians”? 60% of all described amphibian species?

REPLY 17: We agree this sentence was confusing. This sentence has been removed from the revised version and we only refer to the number of species.

“Our phylogenetic model-based imputation approach has expanded our understanding of amphibian thermal tolerance by generating testable predictions for 4,679 unstudied species, particularly in biodiversity hotspots (Fig. 1-2).” (lines 97-100)

Line 355: “populations” these are not really populations

REPLY 18: We have replaced mentions of the word “population” to “local species occurrences” throughout the manuscript (see REPLY 7).

Line 358: these are not really communities (see notes above). Potential assemblages?

REPLY 19: We rephrased all instances of the word “communities” and instead used “assemblages” throughout the manuscript (see REPLY 7).

Line 361: not sure I agree to call this “knowledge” – can this be nuanced a bit more?

REPLY 20: Thank you for the suggestion. We have rephrased this sentence to incorporate more nuance.

“Our phylogenetic model-based imputation approach has expanded our understanding of amphibian thermal tolerance by generating testable predictions for 4,679 unstudied species, particularly in biodiversity hotspots (Fig. 1-2).” (lines 97-100)

Line 364-365: This does not seem like enough evidence of congruence

REPLY 21: We have now incorporated a new panel in Extended Data Figure 2 to visualise the correlation between experimental and imputed values (see REPLY 2).

Fig. 2: Colour choice of TSM and overheating events could be more similar (rather than opposite), so one might more easily look for connections between TSM and # overheating events.

REPLY 22: We have changed the colour of TSM and the number of overheating events to incorporate your suggestion.

The updated figure is provided below.

Fig. 2 | Phylogenetic coverage and taxonomic variation in climate vulnerability. Heat maps show heat tolerance limits (CT_{max}) and thermal safety margins (TSM), while histograms show the number of overheating events (days) averaged across each species' distribution range. Pink bars refer to species with prior knowledge, while grey bars refer to entirely imputed species. This figure was constructed assuming ground-level microclimates occurring under 4°C of global warming above pre-industrial levels. Phylogeny is based on the consensus of 10,000 trees sampled from a posterior distribution (see ³² for details). Highlighted species starting from the right side, anti-clockwise: *Neurergus kaiseri*, *Plethodon kiamichi*, *Bolitoglossa altamazonica*, *Cophixalus aenigma*, *Tomoptera cryptotis*, *Lithobates palustris*, *Allobates subfolionidificans*, *Phyzelaphryne miriamae*, *Barycholos ternetzi*, *Pristimantis carvalhoi*, *Pristimantis ockendeni*, *Boana curupi*, *Teratohyla adenocheira*, *Atelopus spumarius*.

Fig. 2: Could the authors not add information here about which species have data?

REPLY 23: We added a heat map to differentiate the species that were experimentally assessed (pink) from the ones that were imputed (grey; see Fig. 2 or REPLY 22).

Line 398: "populations predicted to overheat had TSMs well above zero"- where is support for this statement? Is this the match up between some pink lines and yellow bars in Fig. 2? Is there a better way to show this (e.g. binomial plot with event as a function of TSM in the supplement?)

REPLY 24: This statement is supported by the statistical results and the data presented in Figure 5c,d. We have now referred to this figure for clarity.

“Interestingly, we found that species predicted to overheat locally have TSMs well above zero, although some are living particularly close to their heat tolerance limits during the warmest months in both terrestrial (mean [95% confidence intervals]; current = 9.13 [7.35 – 11.04], range: 3.02 – 15.36; +4°C = 6.93 [5.17 – 8.84], range: 0.97 – 14.91) and above-ground conditions (current = 9.57 [7.69 – 11.43], range: 3.70 – 11.40; +4°C = 7.36 [5.58 – 9.27], range: 1.75 – 10.00; Fig. 5c,d).” (lines 246-251)

Line 404: “patterns” What pattern? Spatial pattern?

REPLY 25: Thank you for catching this. We removed this sentence to avoid confusion.

Lines 425-427: Is this because CTmax is lower in arboreal species?

REPLY 26: In this sentence, we specifically listed the number of species with an arboreal ecotype that are predicted to overheat in terrestrial conditions. These results are used to show differences between arboreal and terrestrial microenvironments for arboreal species separately, as the number species analysed in the terrestrial analyses is much higher than in arboreal analyses. Comparing the results without sub-setting ecotypes could mislead the reader to think that amphibians face little risk of overheating in arboreal environments relative to terrestrial environments, although we show that these risks are still substantial (see Extended Data Fig. 4). We revised this paragraph for clarity.

“While the overheating risk is lower in arboreal conditions, considerably fewer species were examined than in terrestrial conditions (1,771 vs. 5,177 species). In fact, comparing the responses of arboreal species in different microhabitats revealed that occupying above-ground vegetation is only partially beneficial (Extended Data Fig. 4). In current climates, up to 40 arboreal species (469 local species occurrences) are predicted to experience overheating events in terrestrial conditions, whereas 27 arboreal species (286 local species occurrences) are predicted to overheat in above-ground vegetation (Extended Data Fig. 4). Furthermore, under 4°C of warming, 121 arboreal species (1,424 local species occurrences) are predicted to overheat in terrestrial conditions, while retreating to above-ground vegetation only reduces the number of species exposed to overheating events by 21.5% (95 species, 965 local species occurrences) (Extended Data Fig. 4).” (lines 201-212)

Line 445: I would avoid “significant” here but I would describe this result just a little more, because it is surprising that so few species in some cases makes up such a high proportion. Looks like the proportion of diversity vulnerable is higher in the deserts because richness is lower.

REPLY 27: Thank you for the suggestion. Indeed, some areas where the proportion of species at risk is high are also areas where the species richness is lower. We have added a sentence to highlight some areas with high species richness that have a high predicted proportion of species at risk.

“In addition, the proportion of species predicted to experience overheating events in each assemblage varies geographically and between warming scenarios (Extended Data Fig. 5; Extended Data Table 4). The proportion of species at risk is high in some areas with high species richness (e.g., Northern Australia, Southeastern United States) and not linearly predicted by latitude (Extended Data Fig. 5).” (lines 192-197)

Line 497: Are there actually 1500 points in Fig. 5? Otherwise I don't really see these numbers there.

REPLY 28: Yes, 1497 points are displayed in Fig. 5a (current conditions). Most of these points are overlapping, which makes it hard to visualise all individual points. We reduced the size of the points and increased the random dispersion (jitter) to visualise these points more easily. Note that these changes were also applied to Extended Data Fig. 4, Fig. S7 and Fig. S8, for consistency. The updated Fig. 5 is attached below.

Fig. 5 | Latitudinal variation in the number of overheating events in terrestrial (a,c) and arboreal (b,d) microhabitats as a function of latitude (a,b) and thermal safety margin (c,d). The number of overheating events (days) were calculated as the sum of overheating events (when daily maximum temperatures exceed CT_{max}) during the warmest quarters of 2006-2015 for each species in each grid cell. Blue points depict the number of overheating events in current microclimates, while orange and pink points depict the number of overheating events assuming 2°C and 4°C of global warming above pre-industrial levels, respectively. For clarity, only the species predicted to experience overheating events across latitudes are depicted (a,b).

Line 525: Agree on this point, but it would be nice to point out that some studies estimated TSM with shorter-time frame data (a TSM based on the hottest hour of the hottest day must be more similar to probability of overheating).

REPLY 29: Thank you for your suggestion. We have added more nuance to this statement and cited analyses using TSM based on the hottest hour of the hottest day, which is indeed similar to the probability of overheating.

“While the reliability of TSM-based assessments has been questioned in previous studies⁹, our work further emphasises the need to consider natural climatic variability and extreme hourly temperatures^{15,19–21} when evaluating the vulnerability of ectotherms to global warming.” (lines 302-304)

Line 542: “thermoregulation” I think this is less about thermoregulation and more about opportunities for cooling.

REPLY 30: Thank you for your suggestion. We have changed “thermoregulation” to “finding access to cooler microhabitats”.

“Our findings add to the growing evidence that finding access to cooler microhabitats is the main strategy amphibians and other ectotherms can use to maintain sub-lethal body temperatures^{11,26,39,40}.” (lines 314-316)

Line 566: Considering effects of non-lethal warming might not only lead to more extreme vulnerability, but could also sometimes lead to less extreme estimates, as some species might have greater fitness before overheating

REPLY 31: Thank you for this insight. We have added a paragraph to discuss the possibility of opposite responses.

“Alternatively, species that can retreat underground during heat events are likely to experience fewer overheating events than our models predict (see Fig. S9), and prolonged exposure to high temperatures in the permissive range (*sensu*⁵⁹) can enhance performance and fitness, thereby reducing the impacts of extreme heat on natural populations. In addition, some species may adapt to changing temperatures. However, evidence for slow rates of evolution and physiological constraints on thermal tolerance^{62–64} challenges the likelihood of local adaptation to occur in rapidly warming climates.” (lines 343-349)

Line 574: Were high-risk species more likely those that were imputed? If so, this might mean that some local adaptation could exist but could not be included

REPLY 32: The majority (69%) of high-risk species with positive overheating risks in future climates were indeed imputed. This is expected mainly because most species studied were imputed, and the likelihood of detecting high-risk species in this sample is therefore higher. Local adaptation could exist, indeed, but strong evidence for local adaptation is not supported by our data (see REPLY 5). In fact, we detected a large phylogenetic conservatism in amphibian CT_{max} (Pagel’s $\lambda = 0.95$, after accounting for methodological variation and acclimation temperatures), which is consistent with a large body of evidence in ectotherms (e.g., Comte & Olden, 2017, *Global Change Biology*; Hoffmann et al., 2013, *Functional Ecology*). The presence of a large phylogenetic signal suggests that thermal limits are more constrained by evolutionary history rather than local adaptation, or that closely-related species are inhabiting similar environments and hence similar selection pressures on thermal tolerance. Our model takes into account variation predicted by phylogeny, non-phylogenetic species differences, and variation due to environmental temperatures. Therefore, we hypothesise that undetected local adaptation is unlikely to have major influence on our predictions.

We have added some sentences to discuss the potential for local adaptation to influence our conclusions, and highlighted the proportion of vulnerable species that were imputed.

“We found that these understudied regions frequently harbor species exhibiting the highest susceptibility to extreme heat events (Fig. 1,4-5), with 69% (338 out of 488) of vulnerable species remaining unstudied.” (lines 354-356)

“Our approach to accounting for plasticity assumes that plasticity is homogeneous within species and ignores the possible influence of local adaptation. However, given the low variability in plasticity among species (Fig. S2-3), lack of evidence for latitudinal variation in plasticity (^{33,36,114}), high phylogenetic signal in thermal tolerance (Pagel’s $\lambda^{28} = 0.95$ [0.91 –

0.98]; see *Sensitivity Analyses*), and evidence for slow rates of evolution and physiological constraints on $CT_{max}^{62,64}$, geographic variation in thermal tolerance and plasticity is unlikely to have a major influence on our results.” (lines 768-77).

We also added a paragraph to explain how phylogenetic signal was calculated:

“Finally, we confirmed the presence of a phylogenetic signal in the experimental dataset by fitting a Bayesian linear mixed model using all complete (no missing data) predictors (i.e., acclimation temperature, endpoint, acclimation status, life stage, and ecotype) in *MCMCglmm*. We accounted for phylogenetic non-independence using a correlation matrix of phylogenetic relatedness and fitted random intercepts for non-phylogenetic species effects. The phylogenetic signal (Pagel’s λ^{28} , which is equivalent to phylogenetic heritability^{121,122}) was calculated as the proportion of variance explained by phylogenetic effects relative to the total non-residual variance.” (lines 862-869)

Line 586: I feel that this sentence should end with something like “due to overheating according to our model”

REPLY 33: Thank you for this great suggestion.

This sentence now reads as “Hence, most species are likely to only experience local extirpation due to overheating, according to our models” (lines 369-370).

Line 591: The range-shift field tends to avoid the term “colonization” as it can be a trigger-word for indigenous people. Suggest change to “establishment”

REPLY 34: Thank you for this insightful thought. We have changed the word “colonization” to “establishment”.

“However, this is only possible if suitable habitats are available for establishment.” (lines 374-375)

Line 593: “likely to be rare” can you back this up? There are plenty of amphibians with range shifts in Lenoir et al. 2020

REPLY 35: Thank you for your suggestion. We provided more nuance to this statement.

“Given the low dispersal rates of some amphibians and their common reliance on water bodies for reproduction and thermoregulation, opportunities for range shifts are likely to be rare for many species. Identifying which species at high risk of overheating are simultaneously predicted to have limited ability to extend their range is an interesting avenue for research.” (lines 375-379)

Line 595-596: This is true, but species with large ranges might have a quite different situation at their cold range edges, where warming might improve their foraging success.

REPLY 36: Thank you for this insight. We added “at the warm edge of their distribution” to clarify this statement.

“In addition, we stress that amphibians living close to their physiological limits for extended times at the warm edge of their distribution are likely to experience heat stress that could hamper activity, foraging opportunities, and reproductive success, adding layers of complexity to their survival challenges and potentially leading to population declines^{59,60,71,72}.” (lines 379-383)

Line 608: what ecosystems do amphibians support? Reference needed

REPLY 37: We agree we should have referenced this statement. We have added a reference (Hocking & Babbitt, 2014. *Herpetological Conservation and Biology*) to point to ecosystem services amphibians provide.

“These actions are critical for the amphibians at risk and the ecosystems they support⁷⁵ in a planet undergoing perilous climatic changes.” (lines 395-396)

Line 609: “planet undergoing metamorphosis” seems like an unnecessary biological metaphor – why metamorphosis and not change? If the authors feel strongly about the metaphor, I suggest they develop it or cite a reference.

REPLY 38: We have changed the word “metamorphosis” to “changes” (see REPLY 37). Thank you for the suggestion.

Extended Data Fig. 1. Very nice didactic figure! This reader thanks you.

REPLY 39: Thank you!

Extended Data Fig. 2. Not enough information given about density distribution in (a). How many data points are in this figure? From the reference to this figure (Line 365) it seems to be from the 375 “validated” species but where are the 77 cross-validated species from Line 345? Why not plot one value against the other so we can visualize the relationship we are looking for?

REPLY 40: We have added more details on sample sizes in the figure legend and added a panel to visualise the relationship between imputed and experimental values (see REPLY 2).

Line 1013: Open code and Markdown explainers - this is a very useful resource! Be good to make sure this is archived for publication.

REPLY 41: Thank you. We are glad to hear this can be a very useful resource. We will archive all data, code, and materials in a permanent repository upon acceptance.

Referee #2 (Remarks to the Author):

Increasing attention is being focused on understanding how climate change affects species through more mechanistic means than simply correlating their current distribution with climate and extrapolating it. The authors of this manuscript seek to add to this body of work by predicting physiological limits for a large number of amphibian species based on those with that information and then predicting responses to climate change.

The paper follows a handful of papers that seek to use physiology to predict climate change responses for a smaller number of species and modeling approaches that seek to expand trait-based approaches by filling in missing species trait data. I find that the combined approach can be a powerful tool to move beyond simple non-mechanistic models, while countering the problem of limited data. The directive to maintain microhabitat features is an important outcome.

In general, I find the topic interesting, and the manuscript is well-written. I am not a physiological ecologist, so I hope someone with expertise in this area is available to critique the methods with regard to this aspect. I have some major and minor concerns that, if addressed, should improve the manuscript.

REPLY 42: Thank you for your positive feedback on our manuscript. We are thankful for your excellent suggestions and insights.

Major

It's unclear what is the 'new approach' (L90) since phylogenetic imputation of traits has been around in various forms. I don't think it needs to be new, but if it is claimed as such, it should be made clear what is the advance.

REPLY 43: Thank you for your comment. The approach we developed is new because it combines existing imputation methods into a new framework that we define as Bayesian Augmentation with Chained Equations (BACE). However, you are right that we are not developing a new imputation technique altogether. We have removed the word "new" from this sentence to avoid overstatements.

"We first developed an approach to predict standardised thermal limits for 5,203 amphibian species using data imputation based on phylogenetic niche clustering (Pagel's $\lambda^{28} = 0.95$ [0.91 – 0.98]) and known correlations between critical thermal limits (CT_{max}) and other variables ($n = 2,661$ estimates measured in 524 species; Fig. S2; Methods)." (lines 94-97)

"We developed a phylogenetic imputation procedure, here named Bayesian Augmentation with Chained Equations (BACE). The BACE procedure combines the powers of Bayesian data augmentation and multiple imputation with chain equations (MICE⁸⁴)." (lines 618-620)

The paper makes a broad claim about the conservatism of the analysis by assuming that microenvironments are available. One of the important outcomes is to protect these microenvironments. Yet, a comparison of results with and without microenvironments is not done. I think it would strengthen these recommendations and quantify the potential difference that would result if this analysis was performed.

REPLY 44: Our analyses indeed assume that microenvironments (terrestrial, aquatic, or arboreal conditions in full shade) are always available to amphibians throughout their range of distribution. While amphibians often have access to some level of shade, we believe this is not always true across their range of distribution, particularly in areas with high levels of human disturbance or in open habitats. We agree, however, that an analysis comparing different levels of access to shade is interesting. Therefore, we have provided an analysis (see also REPLY 58) where current terrestrial conditions were assumed to be covered by 50% of shade, which would represent a situation where animals only find partially shaded microhabitats, such as sparse vegetation (e.g., grass), reflecting a more open habitat. We believe running these simulations using 0% of shade (full sun) is not ecologically relevant because amphibians would try to retreat to more suitable habitats during the warmest days, as previous research shows that most ectotherms cannot survive these conditions (e.g. Sunday et al., 2014, PNAS).

Our additional analysis using 50% of shade shows the expected results, where overheating risk is much higher for animals in habitats more exposed to solar radiation (see below).

Fig. S9 | Influence of biophysical model parameters on the estimation of terrestrial thermal safety margins. Depicted is the variation in daily thermal safety margins (TSM) as density distributions according to body mass (a), shade availability and soil depth (b). All simulations were performed assuming 4°C of global warming above pre-industrial levels in a specific grid cell (latitude, longitude = -9.5, -69.5; where the highest number of overheating events was predicted), for the most vulnerable species (*Nobilella myrmecoides*). Negative daily TSMs were recorded as overheating events, and conditions depicted in dark grey reflect the results presented in the manuscript. The number of predicted overheating events is indicated in brackets for each condition.

It's not clear how some of the ecotypes were treated in the analysis. For instance, most salamanders are fossorial or live under logs or substrate. Clearly, they will be experiencing very different temperatures, yet it's not clear this is accounted for and thus could be overestimating risks at least for this subset of species.

REPLY 45: Thank you for your comment. We ran terrestrial and aquatic analyses for all species regardless of their ecotype as most species may use substrate or water sources, at least temporarily. The only exceptions were paedomorphic salamanders which were only assessed in aquatic microenvironments. Arboreal analyses were only performed for arboreal and semi-arboreal species because access to high vegetation requires a morphology adapted to climbing. All simulations assumed highly shaded and humid microenvironments. Therefore, terrestrial conditions (1 cm above ground on a vegetated cover, in our simulations) are likely to reflect microenvironments experienced by salamanders covered by vegetation or substrate.

In our study, we decided not to include analyses stratified by depth (e.g., conditions in a burrow or crevice) because we did not have information about which amphibian species actively use burrows, and to what depth. Moreover, while absolute overheating risks are certainly lower in burrows, relative risks between species should remain similar, as burrow temperatures are directly influenced by above-ground temperatures (Carlo et al., 2017. Ecology Letters). Therefore, species predicted to have a high overheating risk above ground will also likely face the highest risks in burrows.

We have now provided a sensitivity analysis (see Fig. S9; REPLY 44) on the most vulnerable species in our data (*Noblella myrmecoides*), assuming that it can retreat in burrows with depths between 2.5 cm and 20 cm. Interestingly, we found that this species is still predicted to be at risk in burrows up to 10 cm deep, but it is shielded from overheating events at higher depths. Notably, even a shallow burrow of 5 cm reduces the number of predicted overheating events by ~65% (45 vs. 127 days), but overheating risks in shallow burrows (2.5 cm) are comparable to what we predicted (129 vs. 127 days). As expected, burrows are important microhabitat features for amphibian thermoregulation, and our predictions of overheating risk are likely overestimated for burrowing species that have access to suitable conditions (i.e., shaded and humid soil). However, many amphibians do not possess morphological features adapted to burrowing, drought conditions can make soils hard to penetrate, and existing burrows may already be used by potential predators, for instance. Retreating to underground conditions can also carry costs, such as limiting activity, foraging, and reproductive opportunities. Nevertheless, it is an important microhabitat feature that we should not have overlooked, and we thank you for this suggestion.

We have incorporated more nuance to our statements and highlighted that overheating risk may be lower than we expect because of underground thermoregulation.

“Our results were generally robust to changes in model parameters, although amphibians are likely to experience more overheating events in open habitats^{11,52} (Fig. S9) and shallow ponds (Fig. S10), and lower risks in underground conditions¹²⁰ (Fig. S9)” (lines 856-859)

“Moreover, although burrows offer cooler microclimates (see Fig. S9), the ability to use underground spaces is not universal among amphibians and can greatly restrict activity, reproduction, and foraging opportunities.” (lines 322-324)

“Alternatively, species that can retreat underground during heat events are likely to experience fewer overheating events than our models predict (see Fig. S9)” (lines 343-344)

Several models were not run as expected because they ran too slowly or would not converge. Generally, a bit of hard coding and optimization in a language like STAN would solve these problems. The authors might have some colleagues with some experience to get some of these models running as they wanted. It's not absolutely critical but should be easy enough.

REPLY 46: Thank you for your suggestion. We have tried a variety of models, including some in *brms* (which uses STAN), and could not find a way to optimise these models to run. We believe the issue here is deeper than a question of model specification. As discussed in the manuscript, most amphibian species experience no or few overheating events. Therefore, the overheating risk and the number of overheating events is largely zero inflated, with non-zero values concentrated primarily to a limited number of species and observations. Therefore, complex models (e.g., adding various random effects, or a phylogenetic variance-covariance matrix) are unlikely to be able to estimate variance components. While we tried increasing the number of iterations for Bayesian models, running the models in clusters, and many other steps, the models did not converge properly, or were taking longer than what our local supercomputer allow (5 days). While some models did work with these optimisations, because most of them did not, we preferred to keep the model structures consistent throughout to increase the comparability of our estimates. Nevertheless, we believe results should remain qualitatively similar across modelling approaches, although errors may be underestimated in simpler models without complex random effect structures.

We added a note to discuss that models without phylogeny may underestimate the variance in predicted estimates but remain qualitatively similar.

“While the mean estimates from these simpler models should be unbiased, estimate uncertainty is likely underestimated¹¹⁹.” (lines 829-830)

I think the sensitivity analyses should be renamed cross-validation or validation for short. My biggest concern here is that the species that are being imputed are not a random subset – the whole point is that the data is biased, and therefore the types of species being imputed also likely are different. The chance exists that the imputation method is making them like the ones with data (northern hemisphere). I would like to see a validation test that focuses on the types of species with the least information to see how well the model performs.

REPLY 47: Thank you for your thoughtful comment.

We have renamed this section “*Cross-validation and sensitivity analyses*”.

You are right to point out that the species that are being imputed are not a random subset, and therefore the species imputed are likely to be different. However, we also have evidence that CTmax is largely phylogenetically clustered (Pagel's $\lambda = 0.95$), and evidence for little variation in plasticity across species (see also REPLY 5). Therefore, it is likely that knowledge on studied species can be extrapolated to data-deficient species. Unfortunately, it is impossible to validate observations in imputed species without acquiring experimental data. Nevertheless, our model deals with the uncertainty in predicted CTmax values by assigning a large standard error to imputed observations. The more distant species are to the training dataset (e.g., phylogenetically distant, experiencing more extreme environmental conditions), the larger the error for these observations are (which can now be visualised in Extended Data Fig. 2c; see REPLY 2). Our statistical models are then weighed relative to the uncertainty in those predicted values, and very uncertain values contribute little to the predicted patterns. We also provided sensitivity analyses where we considered overheating

events to occur only when operative body temperatures exceeded 50 or 95% of the predicted distribution (confidence intervals) of the imputed CT_{max} (see REPLY 4), hence excluding observations with large uncertainty.

We added a sentence to discuss the potential for local adaptation and geographic variation in plasticity:

“Our approach to accounting for plasticity assumes that plasticity is homogeneous within species and ignores the possible influence of local adaptation. However, given the low variability in plasticity among species (Fig. S2-3), lack of evidence for latitudinal variation in plasticity (^{33,36,114}), high phylogenetic signal in thermal tolerance (Pagel’s $\lambda^{28} = 0.95$ [0.91 – 0.98]; see *Sensitivity Analyses*), and evidence for slow rates of evolution and physiological constraints on CT_{max}^{62,64}, geographic variation in thermal tolerance and plasticity is unlikely to have a major influence on our results.” (lines 768-774).

We have also pointed to our sensitivity analyses more clearly in the methods section:

“Thermal safety margin estimates were weighted by the inverse of their sampling variance to account for the uncertainty in the imputation and predictions across geographical coordinates. However, addressing prediction uncertainty for overheating risk and the number of overheating events was complex due to the dichotomous nature of these metrics (i.e., the species overheats or not). As a remedy, we provide conservative analyses where overheating events were counted only when operative body temperatures exceeded 50% or 95% of the predicted distribution of heat tolerance limits (see *Sensitivity analyses*; Fig. S8).” (lines 808-815)

Along these lines, it is claimed that because the experimental and imputed means were similar that it ‘confirms’ the accuracy of the procedures. This is not a strong test of the procedure because the imputed species likely differ from the experimental species. Therefore, I would even predict that if the method was working that these means should differ. A better approach would be to evaluate the RSME of observed versus imputed for the validation test, focusing on the types of species that needed to be imputed.

REPLY 48: Thank you for your insights. Here, we did not compare the overall mean of all imputed estimates against the mean of all experimental values. These are certainly expected to differ. Instead, the imputed values in this cross-validation approach are predictions of known experimental values from the training dataset. In fact, we trimmed the training dataset, and considered known experimental values as missing values before running separate imputation models. The comparison between these experimental values and the imputed values thus informs us on how well the model performs in predicting missing data. This however assumes that known relationships in this dataset can be generalisable to data-deficient species, which is likely considering the high phylogenetic signal and little evidence for local adaptation, but nevertheless discussed in this revised manuscript (see REPLY 5, 47).

In our case, evaluating the root-mean-square error (RMSE) of imputed vs. predicted would not be highly informative because we are not comparing different approaches to determine which one is most appropriate (e.g., assessing the impact of different parameters on predicted values in simulations by comparing the resulting RMSE values). In our cross-validation, we are directly comparing known experimental values with imputed values. Therefore, we believe it is more informative to visualise the distribution of imputed vs. experimental values, or their correlation (i.e., the RMSE in this cross validation is 1.39, and

the absolute mean error is -0.25; but deviations from equality are more thoroughly described and interpretable in Extended Data Fig. 2b).

We have now added a panel in Extended Data Fig. 2 that shows the correlation between imputed and experimental values in this cross-validation (see REPLY 2).

All throughout, the text indicates the method's success and effectiveness. I like the approach and I agree that it could be useful, but we just don't know if it worked until we collect that data that does not exist and test the predictions. Until then, it's just untested predictions that need to be evaluated. Please tone down this language.

REPLY 49: Thank for your comment. Indeed, the only way to know that this approach truly works for all species is to collect new data and compare them against our predictions. Nevertheless, we think this is an important implication of our study, where we actively encourage to test our predictions and produce evidence in areas that did not receive enough research attention.

We have now toned down our language and invited further research to validate our measurements more clearly.

“We first developed an approach to predict standardised thermal limits for 5,203 amphibian species using data imputation based on phylogenetic niche clustering (Pagel's $\lambda^{28} = 0.95$ [0.91 – 0.98]) and known correlations between critical thermal limits (CT_{max}) and other variables ($n = 2,661$ estimates measured in 524 species; Fig. S2; Methods). Our phylogenetic model-based imputation approach has expanded our understanding of amphibian thermal tolerance by generating testable predictions for 4,679 unstudied species, particularly in biodiversity hotspots (Fig. 1-2). (lines 94-100)

“Our imputation approach has generated testable predictions of the thermal limits of 5,203 species, expanding the scope of previous research¹² (Fig. 2). We also addressed geographical biases by generating predictions in under-sampled but ecologically critical regions of Africa, Asia, and South America (Fig. 2). We found that these understudied regions frequently harbor species exhibiting the highest susceptibility to extreme heat events (Fig. 1,4-5), with 69% (338 out of 488) of vulnerable species remaining unstudied. Targeted research efforts in these vulnerability hotspots are instrumental in validating our model predictions and advancing our understanding of amphibian thermal physiology to inform their conservation. Though undeniable logistical and financial challenges exist in accessing some of these remote locations, collaboration with local scientists could expedite data collection and result in timely conservation measures. Exemplary initiatives to sample numerous species in South America (e.g., ^{25,65,66}) are promising steps in this direction, and we hope our findings will catalyse research activity in these regions.” (lines 351-363)

Line comments

L 47, I'm not sure if this assumption of access to microenvironments makes the predictions conservative – would not some form of shade exist wherever amphibians live?

REPLY 50: Thank for your comment. Indeed, some form of shade should exist wherever amphibians live, however, we doubt largely shaded and moist conditions would necessarily be available at all times across their entire range of distribution. For instance, urban and deforested areas, or drought conditions may not conform with these assumptions.

We have rephrased this sentence as “Our conservative estimates assume access to cool shaded microenvironments, thus the impacts of global warming on amphibians may exceed our projections.” (lines 48-50)

This is also discussed later in the manuscript.

“Our predictions are largely conservative, and likely overestimate the resilience of amphibians to global warming in two main ways. First, we assume that microhabitats such as shaded ground-level substrates, above-ground vegetation, and water bodies are available throughout a species’ range, and that amphibians can maintain wet skin. These assumptions will often be violated as habitats are degraded. Deforestation and urbanization are diminishing vital shaded areas^{41–44}, while increased frequencies of droughts will cause water bodies to evaporate^{45,46}. These changes compromise not only habitat integrity but also local humidity levels – key for effective thermoregulation^{47–49}. Consequently, amphibians will likely experience higher body temperatures and desiccation stress events than our models predict due to inconsistent access to cooler microhabitats⁵⁰, particularly in degraded systems.” (lines 326-335)

L 65, just extreme heat, no need for the word, excursions

REPLY 51: Thank for your suggestion. We removed the word “excursions”. (line 78)

L 77, I wouldn’t call 616 species as a ‘few’

REPLY 52: Thank you for your insight. This sentence has been removed from the revised version, and has been replaced by:

“However, the most exhaustive dataset on amphibian heat tolerance limits only covers 7.5% of known species and is geographically biased towards temperate regions¹² (Fig. 1).” (lines 67-71)

L 91, phylogenetic not phylogenic

REPLY 53: Thank you, we corrected this typo.

L102, I prefer the standard author contribution statement at the end. I find this method of calling out individuals detracts from the paper and the science in favor of emphasizing assignment of individual credit.

REPLY 54: We agree with your suggestion. Referring to individual contributions is a new approach proposed by Nakagawa et al. (2023). Nature Communications. This provides more transparency in author contributions, but we have now mostly removed mentions of author contributions in the methods. Instead, we listed author contributions in the author contribution section.

“This study was conceptualized by PPottier, MRK, SB, SMD, and SN. All data manipulation and analyses were performed by PPottier (with conceptual and technical input from SMD and SN for the imputation methods and statistical analyses, MRK, ARG, JER, and NCW for the biophysical modelling and climate vulnerability analyses). All code was reviewed by NCW, ARG, and JER following the recommendations of ¹²³. Ecotype information was collected by NCW, PPollo, and ANRV. PPottier, NCW, and SMD contributed to data visualization. PPottier wrote the initial draft, and all authors were involved in the review and editing. PPottier oversaw the project administration, while SMD and SN were in charge of the supervision. (lines 1003-1010)

L114, provide reasoning for why this methodology was the only one used.

REPLY 55: Thank you for your suggestion. We only used this methodology because it was by far the most common used and comparable methodology across species.

“First, we only included heat tolerance limits measured using a dynamic methodology (i.e., temperature at which animals lose their motor coordination when exposed to ramping temperatures, critical thermal maximum CT_{max}^{80}) because it was the most used and comparable metric” (lines 585-588)

L 154, quantify what reasonable error is in the main text

REPLY 56: Thank you for your suggestion. We added the estimates in brackets.

“Our cross-validation approach also demonstrated the ability of our models to predict back known experimental estimates with reasonable error (experimental mean \pm standard deviation = 36.19 ± 2.67 ; imputed mean = 35.93 ± 2.54 ; $r = 0.86$; Extended Data Fig. 2).” (lines 631-634)

L 160, would you not weight them by their certainty or inverse sampling variance rather than variance?

REPLY 57: Thank you for your attention to details. MCMCglmm takes the argument “*mev*”, which stands for measurement error variance. Therefore, this statement is correct.

L 191-5, Are there sensitivity analyses to understand how these assumptions might change results?

REPLY 58: Thank you for your suggestion.

We have not done sensitivity analyses to see how changing parameters change the results of all biophysical model outputs. Changing any of these parameters would require a very large number of computations and weeks of run time (we ran 104,385 biophysical models, each taking usually between ~5 and ~40 minutes to run), and it will be difficult to perform a thorough assessment of the consequences of changing each parameter in a global study. Ultimately, these represent simulations of representative microenvironments that amphibians are likely to experience, and it is impossible to model the diversity of microenvironmental features amphibians would experience in a study of this scale.

Nevertheless, we provide a variety of new sensitivity analyses on an example species, where we changed parameters such as burrow depth, shade availability, pond depth, height in above-ground vegetation, the proportion of solar radiation diffused by vegetation, and the proportion of wind reduced by vegetation. We show how these parameters change the estimation of daily thermal safety margins (TSM) and number of overheating events. We refer the reader to these analyses in the manuscript, with additional details in *Supplementary methods*. We found that, while the amount of wind, solar radiation, and the height in above-ground vegetation have negligible influence on TSM or overheating probabilities (Fig. S11), climate vulnerability is higher than predicted in shallower ponds (50 cm depth, relative to 1.5m depth in our standard models) or open habitats (50% shade, relative to 85% shade in our standard models) (Fig. S9-10). Heavier amphibians are also predicted to have a slightly higher risk of overheating than lighter amphibians (Fig. S9), although thermal inertia likely delays body temperature rises in heavier amphibians. On the other hand, underground conditions provide opportunities for thermoregulation, with burrows over 5 cm depth partially or completely protecting this species from extreme heat (Fig. S9), while our results are comparable to those predicted in burrows of 2.5 cm depth (Fig. S9).

Changes made to the manuscript are listed below:

“We investigated alternative ways to i) calculate thermal safety margins, ii) account for acclimation responses, and iii) control for prediction uncertainty (see *Supplementary methods*; Fig. S6-8) and investigated the influence of different parameters of our biophysical models (i.e., shade and burrow availability, plant height, solar radiation, wind speed, pond depth) on predicted vulnerability risks (see *Supplementary methods*; Fig. S9-11). Our results were generally robust to changes in model parameters, although amphibians are likely to experience more overheating events in open habitats^{11,52} (Fig. S9) and shallow ponds (Fig. S10), and lower risks in underground conditions¹²⁰ (Fig. S9).” (lines 852-859)

“We also investigated the influence of different parameters of our biophysical models (i.e., shade and burrow availability, height in above-ground vegetation, solar radiation, wind speed, pond depth) on predicted vulnerability risks (Fig. S9-11). Specifically, we modelled the responses of the species at highest risk in terrestrial and aquatic conditions, *Noblella myrmecoides*, in its most vulnerable location (latitude, longitude = -9.5, -69.5). For terrestrial conditions, we modelled the response of amphibians with different body sizes (0.5, 4.28, or 50 grams), and with different levels of exposure to open habitat conditions. Specifically, we modelled an amphibian exposed to 50% of shade to simulate an open habitat lightly covered by vegetation, and inferred temperatures at different soil depths (2.5, 5, 10, 15, or 20 cm underground). For aquatic conditions, we adjusted pond depths to simulate a very shallow pond (50 cm) and compared it to deeper ponds (1.5- or 3-meters depth). For arboreal conditions, we modelled the responses of *Pristimantis ockendeni*, in its most vulnerable location (-4.5, -71.5), and adjusted the height in above-ground vegetation (0.5, 2, or 5 meters), the percentage of radiation diffused by vegetation (50%, 75%, or 90% of radiation diffused), and the percentage of wind speed reduced by vegetation (0%, 50%, or 80% of wind speed reduced by vegetation). We did not estimate the influence of these parameters on all species and at all locations because of the scale of our study, but these results should provide insight into how varying microenvironmental features and biological characteristics may impact our general conclusions.” (Supplementary materials, lines 27-45)

Fig. S9 | Influence of biophysical model parameters on the estimation of terrestrial thermal safety margins. Depicted is the variation in daily thermal safety margins (TSM) as density distributions according to body mass (a), shade availability and soil depth (b). All simulations were performed assuming 4°C of global warming above pre-industrial levels in a specific grid cell (latitude, longitude = -9.5, -69.5; where the highest number of overheating events was predicted), for the most vulnerable species (*Noblella myrmecoides*). Negative daily TSMs were recorded as overheating events, and conditions depicted in dark grey reflect the results presented in the manuscript. The number of predicted overheating events is indicated in brackets for each condition.

Fig. S10 | Influence of pond depth on the estimation of aquatic thermal safety margins. All simulations were performed assuming 4°C of global warming above pre-industrial levels in a specific grid cell (latitude, longitude = -9.5, -69.5; where the highest number of overheating events was predicted), for the most vulnerable species (*Noblella myrmecoides*). Depicted is the variation in daily thermal safety margins (TSM) as density distributions. Negative daily TSMs were recorded as overheating events, and conditions depicted in dark grey reflect the results presented in the manuscript. The number of predicted overheating events is indicated in brackets for each condition.

Fig. S11 | Influence of biophysical parameters on the estimation of aquatic arboreal safety margins. All simulations were performed assuming 4°C of global warming above pre-industrial levels in a specific grid cell (latitude, longitude = -9.5, -69.5; where the highest number of overheating events was predicted), for the most vulnerable arboreal species (*Pristimantis ockendeni*). Depicted is the variation in daily thermal safety margins (TSM) as density distributions according to height of the animal in above-ground vegetation (a), the percentage of solar radiation diffused by vegetation (b) and the percentage of wind reduced by vegetation (c). Negative daily TSMs were recorded as overheating events, and conditions depicted in dark grey reflect the results presented in the manuscript. The number of predicted overheating events is indicated in brackets for each condition.

L 251, Is there literature that backs up this period of 7 days for acclimation for amphibians? Are there not limits to acclimation at higher temperatures?

REPLY 58: We have chosen 7 days of acclimation because it should allow enough time for most amphibians to fully acclimate, and it was the median acclimation time in the experimental dataset. There is surprisingly limited evidence on the time course of acclimation, but studies on small-bodied amphibians indicate that the full acclimation potential may be reached after 3-4 days (Brattstrom & Lawrence 1982; Layne & Claussen, 1982; Turriago et al., 2023). However, larger amphibians may not acclimate this quickly, and other evidence points to longer acclimation times (i.e., ~7 days, Dallas & Warne, 2023). Therefore, we chose 7 days to reflect on the potential variability in acclimation rates.

We added some justification to the methods:

“While evidence in small amphibians suggests the full acclimation potential is reached within 3-4 days^{110–112}, other evidence points to some variation after longer periods¹¹³. Therefore, we chose 7 days to reflect that some amphibians may require longer to acclimate.” (lines 761-763).

We also did not find evidence for limits to acclimation at higher temperatures in the experimental data. In fact, there is very little variation in acclimation responses in amphibians (Fig. S2; see REPLY 5), and no evidence for a reduction in slopes at higher temperatures. The acclimation temperatures used for our imputation (mean = 25.07°C, range = 4.88 - 31.73°C) are also within the range of temperatures in the experimental data, and therefore, should not be subject to such limits on acclimation.

L 293-4, I would use assemblage instead of community at this scale.

REPLY 59: We now used the word “assemblage” throughout the manuscript, thank you.

L 304, I'm not sure what this means that brms exceeded your computational capacities. It would be a better approach and a cluster can solve computation length.

REPLY 60: These models failed to run in the timeframe allowed by our local supercomputer (5 days, using clusters) or crashed our personal computers. As discussed above (REPLY 46), our data is largely zero-inflated, and not very heterogeneous across levels of our random effects. Using very complex models is therefore challenging given the nature of our data.

We changed “computational capacities” to “computational time and memory limits” for clarity. (line 803).

L 311, should be inverse variance weighting

REPLY 61: Thank you so much for flagging this mistake.

We mistakenly used the wrong weighting approach for generalized additive mixed models. Our results are now updated accordingly. Other weighted models were correctly specified and are not affected. We however noticed a minor mistake in the calculation of weighted standard errors for thermal safety margins, CTmax, and maximum temperatures calculated at the species occurrence and assemblage level. We updated our calculations and results accordingly, and these changes do not impact any of our conclusions.

L 322, similarly, coding these models in STAN or another program would be easy enough and could be tuned to convergence.

REPLY 62: Please see REPLY 46 and 60. Unfortunately we could not tune these models to convergence.

L 349, this should be in the supplement if of value.

REPLY 63: Thank you. Reviewer 1 found this to be a very useful resource (see REPLY 41). These files are too large to be shared in supplement so all data, code, and materials will be archived to a permanent online repository upon acceptance.

L 362, here and elsewhere, it is claimed that taxonomic and geographic biases have been “solved.” The only way we would know that is if we knew the true values and we do not, so all this can do is hope and maybe encourage the collection of these data in the future to test if it is a solution.

REPLY 64: Thank you for your recommendation. We have now toned down the language by emphasising that we only generated predictions and encouraged the collection of new data (see also REPLY 20, 49).

“Our phylogenetic model-based imputation approach has expanded our understanding of amphibian thermal tolerance by generating testable predictions for 4,679 unstudied species, particularly in biodiversity hotspots (Fig. 1-2). (lines 97-100)

“Our imputation approach has generated testable predictions of the thermal limits of 5,203 species, expanding the scope of previous research¹² (Fig. 2). We also addressed geographical biases by generating predictions in under-sampled but ecologically critical regions of Africa, Asia, and South America (Fig. 2). We found that these understudied regions frequently harbor species exhibiting the highest susceptibility to extreme heat events (Fig. 1,4-5), with 69% (338 out of 488) of vulnerable species remaining unstudied. Targeted research efforts in these vulnerability hotspots are instrumental in validating our model predictions and advancing our understanding of amphibian thermal physiology to inform their conservation. Though undeniable logistical and financial challenges exist in accessing some of these remote locations, collaboration with local scientists could expedite data collection and result in timely conservation measures. Exemplary initiatives to sample numerous species in South America (e.g., ^{25,65,66}) are promising steps in this direction, and we hope our findings will catalyse research activity in these regions.” (lines 351-363)

L 387-97, More should be said about the result that CT_{max} basically did not vary with latitude (EXT fig 3 d, e, f). This is why TSM varies with latitude because CT_{max} is a constant. Has this been found before? Is there a chance this is an artifact?

REPLY 65: Thank you for your attention to details. CT_{max} does vary with latitude, but very little in comparison to environmental temperatures. This is indeed congruent with other evidence showing little variation in CT_{max} with latitude (e.g., Sunday et al., 2014; Pinsky et al. 2019).

We added some notes about this in the manuscript:

“However, warming substantially reduce TSM at all latitudes (Fig. 3), likely reflecting the contrast between weak plastic responses in CT_{max} across latitudes^{11,15} (Extended Data Fig. 3; Fig. S3) and large variation in environmental temperatures (Extended Data Fig. 3).” (lines 161-164).

“Such an increase is attributable to the contrast between the rapid pace at which temperatures are increasing and the low ability of amphibians to acclimate to new thermal environments via plasticity (Extended Data Fig. 3; Fig. S3).” (lines 279-281)

L 478-9, isn't this just the result of a variance and the distance from a mean to a threshold? So, it could be predicted based on pure mathematics?

REPLY 66: Thank you for your comment.

Indeed, this could technically be predicted by mathematics, as species living closer to their thermal limits on average are more likely to experience overheating events. However, this plot shows that solely analysing TSM does not provide information about the number of overheating events animals are predicted to experience. In the literature, it is often assumed that climate vulnerability risk is inversely proportional to TSM, but we show that this relationship is not linear, and there exists some variation in responses (i.e., species with the same TSM may experience different number of overheating events). This is most likely due to differences in exposure to daily temperature extremes.

The results are now integrated into the discussion, which should clarify these points:

“Therefore, our findings are inconsistent with the expectation of a general latitudinal gradient in overheating risk based on thermal safety margins^{10,11,13,15}. In fact, the overheating risk does not increase linearly with TSM (Fig. 5c,d), and species with seemingly comparable TSMs can have markedly different probabilities of overheating due to varying exposure to daily temperature fluctuations (Fig. 5c,d). Therefore, TSMs alone hide critical tipping points for thermal stress (Fig. 5c,d).” (lines 291-296)

L 496-7, maybe here make the point that the physiological limits chosen result in death over some short timeline and not just a chronic limit.

REPLY 67: Thank you, we took your suggestion on board.

“Here, we show that nearly 200 species may already experience hourly temperatures that would likely result in death over minutes or hours of exposure in thermal refugia.” (lines 268-270).

L 508, 'some' populations – not all.

REPLY 68: Thank you, we added some nuance to this statement, as suggested.

“Our study clearly demonstrates, as others have suggested^{19,33,36,37}, that physiological plasticity is not a sufficient mechanism to buffer many populations from the impacts of rapidly rising temperatures.” (lines 281-283).

L 550 – warming impacts may exceed projections – I would look at it both ways in this section – how your estimates are possibly high because species go underground. What about local adaptations across ranges of populations? And the possibility for adaptive evolution?

REPLY 69: Thank you for your insights. These are great discussion points. We added more nuance to our statements, and discussed the potential for burrowing, increased performance at warmer temperatures, and adaptive evolution in the discussion.

“Alternatively, species that can retreat underground during heat events are likely to experience fewer overheating events than our models predict (see Fig. S9), and prolonged exposure to high temperatures in the permissive range (*sensu*⁵⁹) can enhance performance and fitness, thereby reducing the impacts of extreme heat on natural populations. In addition, some species may adapt to changing temperatures. However, evidence for slow rates of evolution and physiological constraints on thermal tolerance^{62–64} challenges the likelihood of local adaptation to occur in rapidly warming climates.” (lines 343-349)

L 558, amphibians ‘in these degraded systems’ – not all of them

REPLY 69: Thank you, we incorporated your suggestion.

“Consequently, amphibians will likely experience higher body temperatures and desiccation stress events than our models predict due to inconsistent access to cooler microhabitats⁵⁰, particularly in degraded systems.” (lines 333-335)

L 570, how do you know it’s effective without validation with true estimates?

REPLY 70: We have now removed the word “effectively” and generally toned down the language on how we solved geographical biases.

“Our imputation approach has generated testable predictions of the thermal limits of 5,203 species, expanding the scope of previous research¹² (Fig. 2).” (lines 351-352).

L: 577, are local experts citizens? Regardless, do you think citizens can do CTmax experiments? I think so, but lots to think through including animal welfare, providing the equipment, etc.

REPLY 71: We did not intend to refer to citizens here and we have now changed “experts” to “scientists”. Citizens are more likely to be involved in recording occurrences, or participating in ongoing research projects (e.g., help collecting wild individuals).

“Though undeniable logistical and financial challenges exist in accessing some of these remote locations, collaboration with local scientists could expedite data collection and result in timely conservation measures.” (lines 358-361)

L 592-3, range shifts will only be rare for some species. Some amphibians disperse well (large ranids, toads) and many do not require water bodies (many salamanders). It’s these broad generalizations that reduce the credibility of these statements. The question should be turned around. Some will not expand their range quickly, therefore we need to know which ones, and how physiology will play a role.

REPLY 72: Thank you for your insights. We now gave more nuance to these statements and incorporated your suggestions.

“Given the low dispersal rates of some amphibians and their common reliance on water bodies for reproduction and thermoregulation, opportunities for range shifts are likely to be rare for many species. Identifying which species at high risk of overheating are simultaneously predicted to have limited ability to extend their range is an interesting avenue for research.” (lines 375-377)

L 605, it seems like somewhere, likely here, you will want to mention climate change refugia, defined as habitats with low variability and resistance to change, which would certainly be the types of habitats to be restored, created, and protected.

REPLY 73: Thank you for your suggestion. We now defined refugia as “habitats with cooler and more stable temperatures”.

“These microhabitats provide conditions with cooler and more stable temperatures and increase the potential for amphibians and other ectothermic species to disperse to more suitable microhabitats.” (lines 390-392)

Dear Editor(s),

We are thankful for the opportunity to submit a revised version of our manuscript, “*Vulnerability of amphibians to global warming*”, for consideration in *Nature*.

We have thoroughly revised our manuscript and incorporated the excellent comments we received from the referees. We appreciate their positive feedback and recognise how this second revision has enhanced the quality of our manuscript. Particularly, we believe that propagating the uncertainty in thermal tolerance estimates has greatly improved the robustness and reliability of our predictions and strengthened our original conclusions. We provide point-by-point replies to all referee comments below.

Note that we have updated most figures and extended data tables to reflect changes in the calculation of climate vulnerability metrics. We have also made some changes to the manuscript to comply with *Nature*'s formatting requirements (i.e., added details to figure legends, referenced the summary paragraph). While we have significantly trimmed references from the previous version of the manuscript (from 76 to 60), please note that we have slightly exceeded the number of references recommended for the main text. Unfortunately, we believe we cannot remove further references without compromising the validity of our statements, and we hope it is possible to accommodate this larger reference list.

Yours sincerely,

Patrice Pottier, on behalf of all authors (16 August 2024).

Referees' comments:

Referee #1 (Remarks to the Author):

I find that the changes made by the authors to be very satisfactory enjoyed the discourse in the responses. I see only two remaining issues.

REPLY 1: Thank you for your positive feedback on the first revision. We are delighted to hear most of our changes were satisfactory.

I am still not sure that uncertainty in estimates based the imputed CTmax is being communicated sufficiently. The estimation of uncertainty is mentioned as a positive feature of the imputation method, but no figure in the main text illustrates the uncertainty of imputed CTmax or resulting TSMs. In addition, the authors stated in the response to reviewers (REPLY 4) that they cannot propagate error into estimates of number of overheating events or probability of overheating. However, this seems easily surmountable by iterating these calculations across multiple draws of each value from a normal distribution with SD set to the SE of each CTmax estimate. From each draw, the metrics can be calculated, and a distribution (e.g. mean and SD) across iterations can be calculated or visualized. I do not wish to prescribe a method here, but just to say I do not see that this is insurmountable, and this seems important given the emphasis in *Nature* journals to display uncertainty.

REPLY 2: Thank you for your suggestion. We agree that we could have put more emphasis on the uncertainty of our predictions.

Originally, we aimed to calculate a continuous probability of overheating by sampling from a simulated normal distribution of CTmax and its uncertainty (as suggested). However, this

has proven to be difficult computationally and we partially circumvented this problem by providing more conservative estimates, where overheating events were recorded only when operative body temperatures overlapped 50% of 95% of the predicted distribution of CT_{max}. We have now reconsidered propagating the uncertainty in the number of overheating events and managed to overcome computational challenges by applying an analytical approach to calculating overheating probabilities and their uncertainty (using the *dnorm* function rather than simulating with the *rnorm* function in R). In fact, we have calculated the probability that daily operative body temperatures exceeded the predicted distribution of CT_{max} by extracting the proportion of the distribution of CT_{max} overlapping with the maximum hourly body temperature. We then averaged this probability for each species' local occurrence and multiplied it by the number of days surveyed (warmest quarters of 2006-2015; 910 days) to estimate a predicted number of overheating events and their associated standard error from a binomial distribution. Because our CT_{max} predictions are associated with large errors, as expected from imputation approaches, we restricted the standard deviation of simulated distributions to one, generating values within ~3°C of the mean. This is because large uncertainty can lead to spurious patterns and inflate overheating probabilities for estimates with large uncertainty (see e.g., Fig. S8). Indeed, while the mean of imputed estimates is unbiased (Austin & van Buuren 2022, BMC Med. Res. Methodol.; Madley-Dowd et al., 2019, J. Clin. Epidemiol), the uncertainty of the estimates tends to overestimate true biological variation. Similar approaches to restricting estimate uncertainty have also been used in other data imputation studies (e.g., Callaghan et al., 2021. PNAS). We also provided an alternative calculation, where the standard deviation was capped to the “*biological range*” of CT_{max}, that is, the standard deviation of all CT_{max} estimates across species (s.e. range: 1.84 - 2.17). We present these results in Fig. S8 (d,e,f; see below), which show that the number of overheating events is likely overestimated, particularly in aquatic microhabitats.

We clarified this new approach in the methods:

“Second, we calculated the number of days the maximum daily operative body temperature exceeded CT_{max} across the warmest quarters of 2006-2015, i.e., the number of overheating events. To propagate the uncertainty, we calculated the mean probability that daily operative body temperatures exceeded the predicted distribution of CT_{max} (using the *dnorm* function). Note that the standard error (standard deviation of estimates) of simulated CT_{max} distributions were restricted to one (i.e., simulating distributions within ~3°C of the mean) to avoid inflating overheating probabilities due to large imputation uncertainty (*cf*⁷²; see also *Sensitivity analyses*; Fig. S8). We then multiplied the mean overheating probability by the total number of simulated days (910) to estimate the number of overheating events and their associated standard error using properties of the binomial distribution. Third, we calculated the binary probability (0/1) that species overheat for at least one day across the 910 days surveyed (warmest quarters of 2006-2015).” (lines 760-771)

As well as in supplementary methods:

“To estimate overheating probabilities, we calculated the mean daily probability that operative body temperatures exceeded the predicted distribution of CT_{max} and restricted the standard deviation of simulated distributions to one (i.e., within ~3°C of the mean) to avoid inflating overheating probability for observations with large uncertainty. We also provided alternative results (Fig. S8) where the standard deviation of CT_{max} was restricted to the “*biological range*”, i.e., the standard deviation of the distribution of all CT_{max} estimates across species (range = 1.84 – 2.17). We also provide a sensitivity analysis where overheating risk was positive only when the 95% confidence intervals of predicted overheating days did not overlap with zero (Fig. S8).” (lines 64-72)

Fig. S8 | Latitudinal variation in the number of overheating events using regular (a,b,c), uncertain (d,e,f), or conservative estimates (g,h,i) in terrestrial (a,d,g), aquatic (b,e,h) and arboreal (c,f,i) microhabitats. The number of overheating events (days) were calculated based on the mean probability that daily maximum temperatures exceeded CT_{max} during the warmest quarters of 2006-2015 for each species in each grid cell. Uncertain estimates are those where daily overheating probabilities were calculated based on broad predicted distributions of CT_{max} (i.e., simulated over the whole “*biological range*”), likely inflating overheating probabilities for observations with large uncertainty. Conservative estimates are those when overheating risk was considered only when the 95% confidence intervals of the predicted number of overheating events did not overlap with zero (e,f). Blue points depict the number of overheating events in historical microclimates, while orange and pink points depict the number of overheating events assuming 2°C and 4°C of global warming above pre-industrial levels, respectively. For clarity, only the species predicted to experience overheating events across latitudes are depicted.

Our results and main conclusions remain largely unchanged to incorporating the uncertainty in overheating probabilities. However, we note that the predicted number of overheating events is higher than before (up to 207.18 predicted overheating events in the high warming scenario), though the number of species/populations/communities predicted to experience overheating events is slightly lower than before (up to 391 species predicted to overheat in the high warming scenario, which represents ~7.5% of species). We also found that 11 species are now predicted to experience overheating events in aquatic environments, but only under the high warming scenario (4°C of global warming; see Fig. 4). We have updated our results and figures throughout the main text, extended data, and supplement. We believe propagating the uncertainty in our estimates made our conclusions more robust and reliable, and we thank you for this important suggestion.

To further visualise the uncertainty in the estimates in the figures, we have now changed the point size according to the precision (inverse of standard error) of TSM estimates in Fig. 3 (see below), Extended Data Fig. 3, and Fig. S4. We agree that this was a necessary change, and we hope the readers will appreciate that our estimates have different levels of uncertainty.

Fig. 3 | Assemblage-level patterns in thermal safety margin for amphibians in terrestrial (a), aquatic (b) or arboreal (c) microhabitats. Thermal safety margins (TSM) were calculated as the weighted mean difference between CT_{max} and the predicted operative body temperature in full shade during the warmest quarters of 2006-2015 in each assemblage (1-degree grid cell). Black colour depicts areas with no data. The right panel depicts latitudinal patterns in TSM in current climates (blue) or assuming 4°C of global warming above pre-industrial levels (pink), as predicted from generalised additive mixed models. Point estimates are scaled by precision (1/s.e.). Dashed lines represent the equator and tropics.

Second, I appreciate the authors' perspective that validation of their predictions writ large would constitute a whole other study, and I am mostly satisfied that the authors have instead added more nuance to their imputed CT_{max} values and climate vulnerability estimates as hypothetical. However, the authors have added a figure that shows the range of hourly operative body temperature from their model for several frog species at two sites compared to field body temperatures measured (Fig. S12). Although the authors presented this figure as evidence for validation of their estimated hourly operative body temperatures, I did not find this figure to be supportive, for the reason that the range of modelled operative temperatures was so broad (grey box covering ~14-28°C) with no information about central tendency or the source of variation within that range (e.g. I presume there were diurnal patterns, plus possibly some seasonal variation between the months considered). This range of possible body temperatures is so broad it could arguably encompass the field body temperatures of any frog anywhere, and therefore I do not find this to be a useful validation. Was there any information about time of day of the field body temperature estimates? Is there any information about the spatial scale of the operative body temperature estimate vs. the field measures that can explain and possibly connect the high variation in both? I feel some refinement is needed to increase the precision of the prediction for this figure to support the claim.

REPLY 3: Thank you for this important comment.

We agree that the range of body temperatures predicted from biophysical models was large. This is because little information about the days of sampling was provided in the study we used for the validation, which prevented us from restricting our predictions to match field data more closely. We did provide a measure of central tendency (black horizontal line) in the previous version of the manuscript, but this central tendency is arguably dependent on the range of dates used for predictions. Without further details on the dates, it is unfortunately impossible for us to refine our predictions further. However, we have realised that the field body temperatures were measured between 6 pm and 0:30 am in this study. Therefore, we restricted our predictions between 6 pm and 1 am, which made our body temperature predictions narrower (Fig. S12). We have also displayed the mean (black horizontal line), standard deviation (dark grey box) and range (light grey box) of predicted body temperatures on the figures to display the variation in predicted values more accurately. We can see that restricting the temperature predictions to the right hourly window has significantly increased our predictive power, and we believe it now makes this validation robust and relevant (see Fig. S12 below).

Fig. S12 | Validation of operative body temperature estimations. Terrestrial operative body temperatures estimated from biophysical models were compared to field body temperatures recorded around Tepic (21.48° N, -104.85° W; panel a) and El Cuarenteño (21.45° N, -105.03° W; panel b) between June and October of 2013/2015, for 11 species of frogs⁵. The mean hourly operative body temperatures predicted from our models for the same dates and time windows (18:00 – 01:00) are represented by the black horizontal line, along with their standard deviation (dark grey box), and range (light grey box). The mean (point) and range (bars) of field body temperatures recorded for each species are presented in colour. Note that our analyses were based on the maximum daily temperature recorded at each site during the warmest quarters of 2006-2015, which may not match the times and dates at which field body temperatures were recorded. Nevertheless, congruence between night-time predicted and field body temperatures suggests our models are likely to capture true biological variation in operative body temperatures throughout the day.

As an aside, given the broad range of estimated operative temperatures shown, which I assume is due to some combination of seasonal and daily variation, this figure also made me question which operative body temperature estimate was used for the estimation of climate vulnerability. I found in the methods again that it was the maximum daily, which I believe is sensible, but if this figure is refined and used, I would recommend that some reference to the daily maxima is also made, to keep this clear with readers.

REPLY 4: Thank for your recommendation. We have now emphasised that we are using daily maximum hourly temperatures for our analyses in Fig. S12 (see above) and clarified this in the main text.

“We then integrated predicted thermal limits with daily maximum operative body temperature fluctuations estimated from biophysical models to evaluate the sensitivity of amphibians to extreme heat events in terrestrial, aquatic, and arboreal microhabitats (Extended Data Fig. 1; Methods).” (lines 106-109)

“We estimated the vulnerability of amphibians by estimating daily differences between predicted thermal limits and maximum hourly operative body temperatures (Extended Data Fig. 1; Methods).” (lines 121-123)

These issues are surmountable and I commend the authors on a very thorough study.

REPLY 5: Thank you so much for your positive feedback and excellent suggestions!

Referee #2 (Remarks to the Author):

when I reviewed this manuscript, I thought combining data imputation and mechanistic physiological models was a valuable way to make progress toward predicting a large number of species while applying biological principles. However, I thought some statements were oversold and had some questions about methods. The authors have largely addressed all of my concerns. A study of this size and complexity will have many gaps, and the authors cannot fill them all. Given the computational resources on hand and the many sensitivity analyses in the supplement, I think they have done enough for publication.

REPLY 6: Thank you so much for your positive feedback on our study. Indeed, many gaps remain, but we hope this study will catalyse new research efforts and complement our understanding of the vulnerability of amphibians to climate warming.

One small note - on L 640, the authors still note that they weight heat tolerance estimates by their variance. They suggested in their response that MCMCglmm accepts a weight of the variance. But my understanding is that the model then applies the inverse to it. Whatever the case, one would never weight an estimate by variance, essentially assigning more weight to the least certain estimates. The authors did make this mistake in another place and corrected it based on my comments, so they should also check here/correct the text.

REPLY 7: Thank you for your attention to details. MCMCglmm does indeed take the measurement error variance, but the model processes the variance internally so that our estimates are weighted appropriately by the inverse of the sampling variance.

We have now mentioned that estimates were weighted based on the inverse of their sampling variance to avoid confusion.

“We also weighted heat tolerance estimates based on the inverse of their sampling variance, accounted for phylogenetic non-independence using a correlation matrix of phylogenetic relatedness, and fitted random intercepts for species-specific effects and phylogenetic effects, as well as their correlation with acclimation temperatures (i.e., random slopes).” (lines 616-620)

“In these models, we weighted estimates based on the inverse of their sampling variance, species identity was fitted as a random effect, and we accounted for phylogenetic non-independence using a variance-covariance matrix of phylogenetic relatedness (calculated from the consensus tree of ²⁶).” (lines 803-806)

The paper will provide an important means to extend mechanistic models to more species and generate more accurate predictions of which species and regions are most threatened by climate change.

REPLY 8: Thank you so much!

Referee #3 (Remarks to the Author):

The manuscript by Pottier et al. 2024 is a novel contribution to the global change literature that examines the vulnerability of amphibians to climate warming. I was impressed by the sheer scope of the work presented in that it links a broad range of disciplines together – phylogenetics, biophysical ecology, and macroecology – to tackle a key challenge of understanding the vulnerability of amphibian taxa to climate warming.

REPLY 9: Thank you so much!

I was asked to review specifically the use of climate data in this work. One of the core strengths of this work is the use of microclimatic data to model amphibian vulnerability to warming. The authors rely on the NicheMapR package to downscale gridded global climate data both in time and in spatial resolution while accounting for a range of important biophysical processes such as terrain effects and canopy shading. The NicheMapR package and its applications have been vetted extensively elsewhere along with the relevant biophysics underpinning the downscaling routines. The authors clearly state in the main text the assumptions that are inherent in this downscaling work related to canopy shading. They also note that their approach doesn't account for uncertainty in water availability and changing hydrological conditions. Given the scope and scale of analysis, I think the assumptions employed in the work are reasonable and clearly articulated.

REPLY 10: Thank you for your accurate summary and excellent feedback.

I have one question/concern about the use of climate data as described in the methods. Based on what I can ascertain from the methods, the authors use NCEP reanalysis data for the historic reference period and downscale these data using NicheMapR. The authors then use TerraClimate +2 and +4C projections for estimating future conditions and similarly downscale these to hourly and finer spatial resolutions. Its not clear whether the authors bias correct between the NCEP reference dataset and the Terraclimate futures? The methods state: ‘We ran all microclimatic estimations between 2005 and 2015 to match the range of pseudo-years available for TerraClimate future climate projections.’ Is this a bias correction step? If yes, a simple addition to the paragraph stating explicitly that bias correction was done and how it was done, would improve the clarity of the methods. If bias correction

between these datasets was not done than it would be important to describe why and what implications that may have for the results.

REPLY 11: Thank you for this important comment. We did not need to apply a bias correction because TerraClimate scenario offsets are imposed onto the NCEP data. Because it is an offset from an already-bias-corrected dataset, a subsequent bias correction is not needed. We have added a sentence to clarify this.

“The *micro_ncep* function then downscales monthly TerraClimate inputs to hourly by imposing a diurnal cycle to the data and imposes TerraClimate offsets onto the climatic data from NCEP. Because the TerraClimate data is already bias-corrected, adding future climate projections onto the NCEP data did not require further bias correction.” (lines 671-674)

Beyond this, I found the manuscript to be well written, clear, impactful, and an important contribution to multiple fields. Thanks for the opportunity to review.

REPLY 12: Thank you so much!

Referee #3 (Remarks on code availability):

I briefly examined the code provided by the authors but did not review it in detail or entirety given its length and sophistication. What I can say is that the authors appear to make a concerted effort to provide the community with a valuable set of resources. Thoroughly replicating the analysis will prove to be difficult given the size of the dataset some of which are not available directly online without contacting the authors. Additionally, the use of sophisticated climate downscaling routines will further make the effort challenging.

REPLY 13: Thank you for reviewing the code. Indeed, we could not share the entirety of the materials because of the sheer size of the data and objects. However, we are inviting researchers to contact us to obtain these data and materials and are exploring options to store all materials in a public repository that would allow large storage at minimal cost. We are also happy to assist researchers to replicate our study, although, as you pointed out, it can be computationally difficult.

Dear Editor(s),

We are thankful for the opportunity to submit a revised version of our manuscript, “*Vulnerability of amphibians to global warming*”, for consideration in *Nature*.

We have now addressed the final comments from the referees. We appreciate their positive feedback and have addressed their concerns about one of our statements. We provide point-by-point replies to all referee comments below.

Yours sincerely,

Patrice Pottier, on behalf of all authors (17 December 2024).

Referees' comments:

Referee #2 (Remarks to the Author):

I previously reviewed this article and thought it had done a nice job of breaking the “n-biology barrier,” meaning that it can make predictions for many species but also uses biologically meaningful information by applying imputation techniques.

My remaining questions regarding the methods were answered and corrected in this version.

REPLY 1: Thank you. We are delighted to hear we have addressed your questions and appreciate your feedback and suggestions on the manuscript.

I also was asked to review responses from reviewer 1, who was unable to re-review the manuscript. These responses can be broken down into x points:

Point 1: Need to incorporate uncertainty from the imputation technique

In response, the authors now include this uncertainty as the standard error of the simulated CTmax distributions. They additionally demonstrate that alternative methods can inflate heat risks and thus are undesirable in supplemental analyses and figures. I think there are many ways to approach this, and this is still a developing field. I might have liked having seen a more robust sampling approach applied, but I understand the computational limitations imposed. The applied approach seems to be a good compromise between what is needed and what is practical.

REPLY 2: Thank you for summarising the points made by Referee #1. We agree we would have also liked to use a more robust approach to simulating the uncertainty in the estimates, but we are glad to hear you recognise our approach is a good compromise given the computational limitations. We also believe our approach would provide comparable conclusions to more complex simulations approaches (e.g., sampling from the predicted distribution of each CTmax estimates).

Point 2: Validation was not conclusive

In response, the authors have refined their analyses somewhat after restricting temperatures to those used during sampling. Although this refinement does demonstrate a closer

relationship between observations and predictions, the limitations of the data and few species evaluated do not make as strong of a case, at least in my mind, as is indicated in the main text: this validation “confirmed our imputation approach was accurate and unbiased by demonstrating a strong congruence between experimental and imputed data.” I recommend toning down this statement quite a bit. I’m glad that some validation was attempted, but I think it’s too limited for over-confidence.

REPLY 3: Thank you for your thoughtful comment. We agree that our validation is not comprehensive and mostly focuses on a sample of species. However, the statement you have highlighted (“*We confirmed our imputation approach was accurate and unbiased by demonstrating a strong congruence between experimental and imputed data in cross-validations (experimental mean \pm standard deviation = 36.19 ± 2.67 ; imputed mean = 35.93 ± 2.54 ; $n = 375$; $r = 0.86$; Extended Data Fig. 2a,b), though, as expected, the uncertainty in imputed predictions was higher in understudied clades (Extended Data Fig. 2c).*”; Lines 87-91) refers to our statistical cross-validation approach between experimental and imputed CTmax estimates (Extended Data Fig. 2); not the validation of the field body temperatures (previously presented in Fig. S12).

The statements on the validation of field body temperature estimates (previously presented in Fig. S12) in the main text are:

“This modelling system has been extensively validated with field observations^{91–93} (see also Fig. S12).” (Lines 599-600)

“We also confirmed that predicted operative body temperatures were comparable to field body temperatures measured in some wild frogs (see Supplementary methods; Fig. S12).” (Lines 786-789).

We have not toned down the latest statement to invite additional validations. Note that Fig. S12 has now been moved to Extended Data Fig. 10.

“We confirmed that predicted operative body temperatures were comparable to field body temperatures measured in some wild frogs (Extended Data Fig. 10), and we invite additional validations with other species in different geographical areas.” (Lines 786-789).

We have also toned down that the referee pointed out by adding “likely”, to avoid confusion:

“We confirmed our imputation approach was likely accurate and unbiased by demonstrating a strong congruence between experimental and imputed data in cross-validations (experimental mean \pm standard deviation = 36.19 ± 2.67 ; imputed mean = 35.93 ± 2.54 ; $n = 375$; $r = 0.86$; Extended Data Fig. 2a,b), though, as expected, the uncertainty in imputed predictions was higher in understudied clades (Extended Data Fig. 2c).” (Lines 87-91)

Point 3: Emphasize that daily maximum hourly temperatures are used
This point has been fully addressed.

REPLY 4: Thank you!

Overall, except for some toning down of the confidence of statements, I find that the authors made a substantial effort to address these legitimate comments.

REPLY 5: Thank you so much! We agree these were very important comments, and we are glad you think we have addressed them correctly.

Referee #2 (Remarks on code availability):

I don't have the time to review the code again, but did so earlier. Another reviewer also took a look this round.

REPLY 6: Thank you for checking the code in the previous version.

Referee #3 (Remarks to the Author):

I previously reviewed this manuscript focusing on the use of climate data in the analysis. The authors have clearly and succinctly responded to my questions/comments about bias correction between global climate datasets and have addressed my concerns. I have no further comments to add.

REPLY 7: Thank you so much! We are delighted to hear that we have addressed your concerns.